# Robust Reinforcement Learning with General Utility

**Ziyi Chen, Yan Wen, Zhengmian Hu, Heng Huang**
Department of Computer Science, Institute of Health Computing,
University of Maryland College Park
College Park, MA 20742, USA
{zc286,ywen1,zhu123,heng}@umd.edu

## Abstract

Reinforcement Learning (RL) problem with general utility is a powerful decision making framework that covers standard RL with cumulative cost, exploration problems, and demonstration learning. Existing works on RL with general utility do not consider the robustness under environmental perturbation, which is important to adapt RL system in the real-world environment that differs from the training environment. To train a robust policy, we propose a robust RL framework with general utility, which subsumes many existing RL frameworks including RL, robust RL, RL with general utility, constrained RL, robust constrained RL, pure exploration, robust entropy regularized RL, etc. Then we focus on popular convex utility functions, with which our proposed learning framework is a challenging nonconvex-nonconcave minimax optimization problem, and design a two-phase stochastic policy gradient type algorithm and obtain its sample complexity result for gradient convergence. Furthermore, for convex utility on a widely used polyhedral ambiguity set, we design an algorithm and obtain its convergence rate to a global optimal solution.

## 1 Introduction

Reinforcement learning (RL) is an important decision-making framework [41] aiming to find the optimal policy that minimizes accumulative cost, which is also a *linear utility function* of occupancy measure. Recent works have extended standard RL to more *general utility functions* to account for a variety of practical needs, including risk-sensitive applications [22, 8, 52], exploration maximization [18, 54, 51, 13, 6], and safety constraints [54, 51, 13]. There are provably convergent algorithms to solve RL with general utility [54, 55, 6]. However, these works study RL with general utility in a fixed environment, which may fail in many applications where the policy is trained in a simulation environment but implemented in a different real-world environment [37, 56].

To make the policy robust to such environmental change, robust RL has been proposed to find the optimal robust policy under the worst possible environment [36, 20, 48, 45, 14]. However, all the existing robust RL works restrict to linear utility function to our knowledge. Therefore, we aim to answer the following research question:

> *Q: Can we train a robust policy for RL with general utility and obtain convergence results?*

### 1.1 Our Contributions

We affirmatively answer this question by proposing robust RL with general utility, the first learning framework that obtains a robust policy for general utility in the worst possible environment. It is formulated as a minimax optimization problem $\min_{\theta \in \Theta} \max_{\xi \in \Xi} f(\lambda_{\theta,\xi})$ where $f$ is the utility function and $\lambda_{\theta,\xi}$ is the occupancy measure under the policy parameter $\theta \in \Theta$ and the environmental

38th Conference on Neural Information Processing Systems (NeurIPS 2024).

transition kernel parameter $\xi \in \Xi$. Our robust RL with general utility is a combination of its two important special cases, namely, RL with general utility [54] (formulated as $\min_{\theta \in \Theta} f(\lambda_{\theta,\xi})$ where the environmental parameter $\xi$ is fixed) and robust RL [36] where $f$ is restricted to linear utility function. This new learning framework also covers many other existing RL frameworks including constrained RL [2] and robust constrained RL [43] with safety critical applications such as healthcare and unmanned drones, entropy regularized RL [10] and its robust extension [32] which help agents learn from human demonstration, pure exploration [18] with application to explore an environment with sparse reward signals and its robust extension, etc. These examples use convex utility functions $f$, which is the focus of this paper. See Section 2.1 for details of these examples.

Then, we focus on designing provably convergent algorithms for our new proposed learning framework with the widely used convex utility function $f$. In this case, our objective $\min_\theta \max_\xi f(\lambda_{\theta,\xi})$ is still a highly challenging nonconvex-nonconcave minimax optimization problem. Hence, we have to utilize the structure and properties of $\lambda_{\theta,\xi}$ to design algorithms and obtain convergence results. To elaborate, we design a projected stochastic gradient descent ascent algorithm with two phases. Interestingly, the first phase targeted at the objective function $f$ obtains a stationary point of a different envelope function. Hence, we add a second phase targeted at a corrected objective $\widetilde{f}$ to converge to a near-stationary solution of the original objective $f$. The convergence analysis is non-trivial with two novel techniques. First, we have proved a projected gradient dominance property (Proposition 4) that is much stronger than the existing one on convex utility, with less assumptions, no bias term and applicability to more general parameterized policy. Second, in the convergence analysis of the second phase, we obtain convergence to a global Nash equilibrium (thus a stationary point) of $\widetilde{f}$ by Proposition 4, which is close to a stationary point of $f$ by proving that $\nabla_\xi \widetilde{f}(\lambda_{\theta,\xi}) \approx \nabla_\xi f(\lambda_{\theta,\xi})$.

Furthermore, with convex utility function $f$ and the widely used $s$-rectangular polyhedral ambiguity set $\Xi$ (including the popular $L^1$ and $L^\infty$ ambiguity sets), we design an alternative algorithm which **converges to a global optimal solution of this nonconvex-nonconcave optimization problem** at a sublinear rate $\mathcal{O}(1/K)$ (Theorem 3). This is much more challenging than global convergence for convex RL (that is, RL with convex utility function and fixed $\xi$) [54, 51, 6] and for robust RL with linear utility satisfying Bellman equation [36, 20, 45, 15, 25], so we need novel algorithm design and novel techniques. First, we prove that $\arg\max_\xi f(\lambda_{\theta,\xi})$ can be obtained in the finite set of vertices $V(\Xi)$ (Proposition 6). This is intuitive if $f(\lambda_{\theta,.})$ is a convex function but in many applications, only $f(\lambda)$ is convex. To solve this challenge, we prove a novel local invertibility property of $\lambda_{\theta,.}$ (Proposition 5) by checking Bellman equation of $\lambda_{\theta,\xi}$ state by state in two cases. Then we prove Proposition 6 using a novel state-by-state extension from an optimal non-vertex $\xi$ to an optimal vertex $\xi$. Second, the major difficulty to design our algorithm is to find a descent direction of $\Gamma(\theta) := \max_\xi f(\lambda_{\theta,\xi})$. We select the near-optimal vertices $\xi \in \Xi_k \subset V(\Xi)$ that may affect the optimization progress $\Gamma(\theta_{k+1}) - \Gamma(\theta_k)$, and find the descent direction with provably large descent for all the corresponding functions $\{f(\lambda_{\theta_k,\xi})\}_{\xi \in \Xi_k}$ (Proposition 7) via convex optimization. Third, by Proposition 6, the global convergence measure $\Delta_k := \Gamma(\theta_k) - \min_\theta \Gamma(\theta)$ at each iteration $k$ either is $\mathcal{O}(1/k)$-close to optimal ($\Gamma(\theta_k) \leq \mathcal{O}(1/k)$) or enjoys large descent (Eq. (26)), so we prove the convergence in 3 cases: $\mathcal{O}(1/K)$-optimal final $\theta_K$, iterate Eq. (26) from $\theta_0$ or from a $\mathcal{O}(1/k)$-optimal intermediate $\theta_k$.

## 1.2 Related Works

**RL with General Utility.** Standard RL aims to optimize over the accumulated reward/cost [21, 41]. Some early operation research works focus on other non-linear objectives such as variance-penalized MDPs [12], risk-sensitive objectives [22, 8, 52], entropy exploration [18], constrained RL [2, 1, 35] and learning from demonstration [39, 3].

[54] proposes RL with general utilities to cover the above applications and applies variational policy gradient method that provably converges to the global optimal solution for convex utility. [55] proposes a variance reduced policy gradient algorithm which requires $\widetilde{\mathcal{O}}(\epsilon^{-3})$ samples to achieve an $\epsilon$-stationary policy for general utility and $\widetilde{\mathcal{O}}(\epsilon^{-2})$ samples to achieve an $\epsilon$-global optimal policy for convex utility and overparameterized policy. [51] provides a meta-algorithm to solve the convex MDP problem as a min-max game between a policy player and a cost player who produces rewards that the policy player must maximize. They further show that any method-solving problems under the standard RL settings can be used to solve the more general convex MDP problem. [27] obtains policy

gradient theorem for RL with general utilities. [6] proposes a simpler single-loop parameter-free normalized policy gradient algorithm with recursive momentum variance reduction. This algorithm requires $\widetilde{\mathcal{O}}(\epsilon^{-3})$ samples to achieve $\epsilon$-stationary policy in general and $\widetilde{\mathcal{O}}(\epsilon^{-2})$ samples to achieve $\epsilon$-global optimal policy for convex utility. For large finite state action spaces, it requires $\widetilde{\mathcal{O}}(\epsilon^{-4})$ samples to achieve $\epsilon$-stationary policy via linear function approximation of the occupancy measure. [53] proposes decentralized multi-agent RL with general utilities. [13] shows that convex RL is a subclass of multi-agent mean-field games.

**Robust RL.** Robust RL is designed to learn a policy that is robust to perturbation of environmental factors. Usually robust RL is NP-hard [45], but becomes tractable for ambiguity sets that is $(s, a)$-rectangular [36, 20, 45, 44, 29, 56] or $s$-rectangular [45, 42, 23, 26]. Methods to solve robust RL include value iteration [36, 20, 45, 15, 25], policy iteration [20, 4, 24] and policy gradient [29, 44, 56, 42, 26, 17, 28].

## 2 Robust Reinforcement Learning with General Utility

**Notations.** The space of probability distribution on a space $\mathcal{X}$ is denoted as $\Delta^{\mathcal{X}}$. If $\mathcal{X}$ is finite, we denote its cardinality as $|\mathcal{X}|$. $\|\cdot\|_p$ denotes $p$-norm of vectors and $\|\cdot\| = \|\cdot\|_2$ by default.

**Reinforcement Learning with General Utility.** Reinforcement Learning (RL) with general utility is an emerging learning framework [54, 55, 6], specified by a tuple $\langle \mathcal{S}, \mathcal{A}, p_\xi, f, \rho, \gamma \rangle$, with finite state space $\mathcal{S}$, finite action space $\mathcal{A}$, transition kernel $p_\xi \in (\Delta^{\mathcal{S}})^{\mathcal{S} \times \mathcal{A}}$ parameterized by $\xi \in \Xi$ ($\Xi \subset \mathbb{R}^{d_\Xi}$ is convex and compact), discount factor $\gamma \in (0, 1)$, general utility function $f : \Delta^{\mathcal{S} \times \mathcal{A}} \to \mathbb{R}$ and the distribution $\rho \in \Delta^{\mathcal{S}}$ of the initial state $s_0$. At time $t$, given the environmental state $s_t$, the agent takes action $a_t \sim \pi_\theta(\cdot|s_t)$ based on a policy $\pi_\theta \in (\Delta^{\mathcal{A}})^{\mathcal{S}}$ parameterized by $\theta \in \Theta$ ($\Theta \subset \mathbb{R}^{d_\Theta}$ is convex). Then the environment transitions to state $s_{t+1} \sim p_\xi(\cdot|s_t, a_t)$. The occupancy measure $\lambda_{\theta,\xi} \in \Delta^{\mathcal{S} \times \mathcal{A}}$ at $(s, a) \in \mathcal{S} \times \mathcal{A}$ is defined below.

$$\lambda_{\theta,\xi}(s, a) \stackrel{\text{def}}{=} (1 - \gamma) \sum_{t=0}^{+\infty} \gamma^t \mathbb{P}_{\pi_\theta, p_\xi}(s_t = s, a_t = a | s_0 \sim \rho), \tag{1}$$

where $\mathbb{P}_{\pi_\theta, p_\xi}$ denotes the probability measure of the Markov chain $\{s_t, a_t\}_{t \geq 0}$ induced by policy $\pi_\theta$, transition kernel $p_\xi$ and the initial distribution $\rho$. The aim of the agent is to find the optimal policy $\pi_\theta$ that solves $\min_{\theta \in \Theta} f(\lambda_{\theta,\xi})$ given fixed transition kernel $p_\xi$. Here, $f(\lambda_{\theta,\xi})$ can be seen as the overall cost of selecting policy $\pi_\theta$ in the environment $p_\xi$, and there are many examples of the utility function $f$ covering a variety of applications (See Section 2.1). However, existing works on RL with general utility assume a fixed environmental transition kernel $p_\xi$, which may fail in many applications where the policy is deployed in a real-world environment different from the simulation environment for training. To obtain a policy that is robust to such environmental change, we propose a new learning framework called *robust RL with General Utility* as follows.

**Our Proposed Robust RL with General Utility.** The goal of our proposed *robust RL with general utility* is to find an optimal robust policy under the worst possible environmental parameter $\xi$ from an ambiguity set $\Xi$, as formulated by the following minimax optimization problem with general utility function $f$.

$$\min_{\theta \in \Theta} \max_{\xi \in \Xi} f(\lambda_{\theta,\xi}), \tag{2}$$

In practice, the distance between the real-world environment (for deployment) and simulation environment (for training) is assumed to be bounded. Therefore, $\Xi$ is usually set as a neighborhood around the nominal kernel $\widehat{\xi}$ estimated from the simulation environment, i.e. $\Xi = \{\xi \in \mathbb{R}^{d_\Theta} : d(\xi, \widehat{\xi}) \leq d_0\}$ with distance measure $d$ and the distance upper bound $d_0 \geq 0$.

### 2.1 Examples of Our Robust RL with General Utility

**Example 1: RL with General Utility.**
When $\Xi = \{\widehat{\xi}\}$ for a fixed environmental parameter $\widehat{\xi}$, our proposed *robust RL with general utility* (2) reduces to (non-robust) *RL with general utility* $\min_{\theta \in \Theta} f(\lambda_{\theta,\widehat{\xi}})$, as introduced above.

**Example 2: Robust Constrained RL and Its Special Cases.**
Robust constrained RL [38, 43, 40] is an emerging learning framework where an agent should obey safety conditions in all possible real-world environments, which is important in safety critical applications such as healthcare and unmanned aerial vehicle [43]. For math formulation, denote $c^{(0)}, c^{(1)}, \ldots, c^{(K)}$ as cost functions $\mathcal{S} \times \mathcal{A} \to \mathbb{R}$. At time $t$, the agent receives performance-related cost $c^{(0)}(s_t, a_t)$ and safety-related costs $\{c^{(k)}(s_t, a_t)\}_{k=1}^K$. Define value functions $V_{\theta,\xi}^{(k)}$ and robust value functions $V_\theta^{(k)}$ as follows.

$$V_{\theta,\xi}^{(k)} \stackrel{\text{def}}{=} \langle c^{(k)}, \lambda_{\theta,\xi} \rangle = \sum_{s,a} c^{(k)}(s,a) \lambda_{\theta,\xi}(s,a), \quad V_\theta^{(k)} \stackrel{\text{def}}{=} \max_{\xi \in \Xi} V_{\theta,\xi}^{(k)}, \ k = 0, 1, \ldots, K. \quad (3)$$

Then robust constrained RL is formulated as the following constrained policy optimization problem.

$$\min_{\theta \in \Theta} V_\theta^{(0)}, \text{ s.t. } V_\theta^{(k)} \le \tau_k \text{ for all } k = 1, \ldots, K, \quad (4)$$

where $\tau_k \in \mathbb{R}$ is the safety threshold, and $V_\theta^{(k)} \le \tau_k$ means that the safety constraints $V_{\theta,\xi}^{(k)} \le \tau_k$ holds for any environmental parameter $\xi \in \Xi$.

**Proposition 1.** *The robust constrained RL problem* (4) *is a special case of our proposed robust RL with general utility* (2) *using the following convex utility function* $f$.

$$f(\lambda) = \begin{cases} \langle c^{(0)}, \lambda \rangle, & \text{if } \langle c^{(k)}, \lambda \rangle \le \tau_k \text{ for all } k = 1, \ldots, K \\ +\infty, & \text{otherwise} \end{cases}. \quad (5)$$

After removing the safety constraints, *robust constrained RL* reduces to an important special case called *robust RL* (formulated as $\min_{\theta \in \Theta} \max_{\xi \in \Xi} \langle c^{(0)}, \lambda_{\theta,\xi} \rangle$ with linear utility function $f(\lambda) = \langle c^{(0)}, \lambda \rangle$) [36]. Furthermore, when $\Xi = \{\widehat{\xi}\}$ for fixed $\widehat{\xi}$, *robust constrained RL* and *robust RL* reduce to *constrained RL* [2] and *RL* [41] respectively. All these examples are important special cases of our proposed *robust RL with general utility* based on Proposition 1.

**Example 3: Robust Entropy Regularized RL and Its Special Cases.**
Robust entropy regularized RL is also an important RL framework with application to imitation learning and inverse reinforcement learning which help agents learn from human experts' demonstration [32, 33], and is formulated as the following minimax optimization problem.

$$\min_{\theta \in \Theta} \max_{\xi \in \Xi} \sum_{s,a} \left[ \lambda_{\theta,\xi}(s,a) c(s,a) \right] - \mu \sum_s \left[ \lambda_{\theta,\xi}(s) \mathcal{H}[\pi_\theta(\cdot|s)] \right], \quad (6)$$

where $c$ is a cost function, $\lambda_{\theta,\xi}(s) = \sum_a \lambda_{\theta,\xi}(s,a)$ is the state occupancy measure, and $\mathcal{H}[\pi_\theta(\cdot|s)] = -\sum_a \pi_\theta(a|s) \log \pi_\theta(a|s)$ is the entropy regularizer (with coefficient $\mu \ge 0$) which encourages the agent to explore more states and actions and helps to prevent early convergence to sub-optimal policies.

**Proposition 2.** *The robust entropy regularized RL problem* (6) *is a special case of our proposed robust RL with general utility* (2) *using the following convex utility function* $f$.

$$f(\lambda) = \sum_{s,a} \lambda(s,a) \left[ c(s,a) + \mu \log \frac{\lambda(s,a)}{\sum_{a'} \lambda(s,a')} \right]. \quad (7)$$

When $\mu = 0$, *robust entropy regularized RL* (6) reduces to *robust RL* [36]. When $c \equiv 0$ but $\mu > 0$, *robust entropy regularized RL* reduces to *robust pure exploration*. Furthermore, when $\Xi = \{\widehat{\xi}\}$, *robust entropy regularized RL*, *robust RL* and *robust pure exploration* reduce to *entropy regularized RL* [10], *RL* [41] and *pure exploration* [18] respectively. All these examples are important special cases of our proposed *robust RL with general utility* based on Proposition 2.

## 2.2 Gradients for Our Robust RL with General Utility

**Theorem 1.** *The gradients of the objective function* (2) *for our proposed robust RL with general utility can be computed as follows.*

$$\nabla_\theta f(\lambda_{\theta,\xi}) = \mathbb{E}_{\pi_\theta, p_\xi} \left[ \sum_{t=0}^{+\infty} \gamma^t \frac{\partial f(\lambda_{\theta,\xi})}{\partial \lambda_{\theta,\xi}(s_t, a_t)} \sum_{h=0}^t \nabla_\theta \log \pi_\theta(a_h|s_h) \bigg| s_0 \sim \rho \right], \quad (8)$$

$$\nabla_\xi f(\lambda_{\theta,\xi}) = \mathbb{E}_{\pi_\theta, p_\xi} \left[ \sum_{t=0}^{+\infty} \gamma^t \frac{\partial f(\lambda_{\theta,\xi})}{\partial \lambda_{\theta,\xi}(s_t, a_t)} \sum_{h=0}^{t} \nabla_\xi \log p_\xi(s_{h+1}|s_h, a_h) \middle| s_0 \sim \rho \right]. \tag{9}$$

We make the following standard assumptions which are also used in RL with general utility [55, 6].

**Assumption 1.** *There exist constants $l_{\pi_\theta}, L_{\pi_\theta}, l_{p_\xi}, L_{p_\xi} > 0$ such that for all $s, s' \in \mathcal{S}, a \in \mathcal{A}$, $\theta, \theta' \in \Theta$ and $\xi, \xi' \in \Xi$, we have*

$$\|\nabla_\theta \log \pi_\theta(a|s)\| \le \ell_{\pi_\theta}, \quad \|\nabla_\theta \log \pi_{\theta'}(a|s) - \nabla_\theta \log \pi_\theta(a|s)\| \le L_{\pi_\theta} \|\theta' - \theta\|,$$

$$\|\nabla_\xi \log p_\xi(s'|s, a)\| \le \ell_{p_\xi}, \quad \|\nabla_\xi \log p_{\xi'}(s'|s, a) - \nabla_\xi \log p_\xi(s'|s, a)\| \le L_{p_\xi} \|\xi' - \xi\|.$$

**Assumption 2.** *There exist constants $l_\lambda, L_\lambda > 0$ such that for all $\lambda, \lambda' \in \Delta^{\mathcal{S} \times \mathcal{A}}$, $\|\nabla_\lambda f(\lambda)\| \le l_\lambda$ and $\|\nabla_\lambda f(\lambda') - \nabla_\lambda f(\lambda)\| \le L_\lambda \|\lambda' - \lambda\|$.*

**Proposition 3.** *Under Assumptions 1 and 2, the gradients (8) and (9) satisfy the following bounds for any $\theta, \theta' \in \Theta$ and $\xi, \xi' \in \Xi$.*

$$\|\nabla_\theta f(\lambda_{\theta,\xi})\| \le \ell_\theta := \frac{\ell_{\pi_\theta}}{(1-\gamma)^2}, \quad \|\nabla_\xi f(\lambda_{\theta,\xi})\| \le \ell_\xi := \frac{\ell_{p_\xi}}{(1-\gamma)^2}, \tag{10}$$

$$\|\nabla_\theta f(\lambda_{\theta',\xi'}) - \nabla_\theta f(\lambda_{\theta,\xi})\| \le L_{\theta,\theta} \|\theta' - \theta\| + L_{\theta,\xi} \|\xi' - \xi\|, \tag{11}$$

$$\|\nabla_\xi f(\lambda_{\theta',\xi'}) - \nabla_\xi f(\lambda_{\theta,\xi})\| \le L_{\xi,\theta} \|\theta' - \theta\| + L_{\xi,\xi} \|\xi' - \xi\|, \tag{12}$$

*where* $L_{\theta,\theta} := \frac{\ell_{\pi_\theta}^2 \sqrt{|\mathcal{A}|}(L_\lambda + \ell_\lambda \sqrt{|\mathcal{S}||\mathcal{A}|})}{(1-\gamma)^3} + \frac{L_{\pi_\theta} \ell_\lambda}{(1-\gamma)^2}$, $L_{\theta,\xi} := \frac{\gamma \ell_{\pi_\theta} \ell_{p_\xi} \sqrt{|\mathcal{S}|}}{(1-\gamma)^3}(L_\lambda + 2\ell_\lambda \sqrt{|\mathcal{S}||\mathcal{A}|})$, $L_{\xi,\theta} := \frac{\ell_{\pi_\theta} \ell_{p_\xi} \sqrt{|\mathcal{A}|}(L_\lambda + \ell_\lambda \sqrt{|\mathcal{S}||\mathcal{A}|})}{(1-\gamma)^3}$, $L_{\xi,\xi} := \frac{\gamma \ell_{p_\xi}^2 \sqrt{|\mathcal{S}|}(L_\lambda + 2\ell_\lambda \sqrt{|\mathcal{S}||\mathcal{A}|})}{(1-\gamma)^3} + \frac{\ell_\lambda(L_{p_\xi} + \ell_{p_\xi}^2 |\mathcal{S}|)}{(1-\gamma)^2}$.

In practice, the exact gradients (8) and (9) are unavailable and can only be estimated via stochastic samples. We refer the details to Appendix C as those largely follow [6].

Define the following projected gradients with stepsizes $b, a > 0$, which have been used to measure convergence of algorithms to stationary points of optimization [30, 5, 47] and RL problems [49, 46, 34].

$$G_b^{(\theta)}(\theta, \xi) := \frac{1}{b}\left[\theta - \text{proj}_\Theta\left(\theta - b\nabla_\theta f(\lambda_{\theta,\xi})\right)\right], \quad G_a^{(\xi)}(\theta, \xi) := \frac{1}{a}\left[\text{proj}_\Xi\left(\xi + a\nabla_\xi f(\lambda_{\theta,\xi})\right) - \xi\right] \tag{13}$$

## 3  Gradient Convergence for Convex Utility

**Assumption 3.** *The utility function $f(\lambda)$ is convex.*

Robust RL with convex utility functions $f$ subsumes many important special cases, including robust constrained RL, robust entropy regularized RL, constrained RL, robust RL, RL, pure exploration, etc., as shown in Examples 2 and 3 in Section 2.1.

Partially inspired by the gradient descent ascent (GDA) algorithm [31] for nonconvex-concave minimax optimization, we design the projected stochastic GDA algorithm (Algorithm 1) with two phases to solve robust RL with convex utility. The first phase (called *original phase*) can be seen as projected stochastic GDA algorithm on the *original* objective function $f$. Specifically, in the $k$-th the outer loop with fixed $\xi_k$, the inner loop applies $T$ projected stochastic gradient descent steps (14) to obtain $\theta_k$ which converges to the global solution of $\Phi(\xi_k) := \min_{\theta \in \Theta} f(\lambda_{\theta,\xi_k})$ as $f$ is convex. Then, we update $\xi_k$ using the projected stochastic gradient ascent step (15). However, the output $\widetilde{\xi}$ of the first phase only converges to a stationary point of the following the envelope function $\widetilde{\Phi}$ [1].

$$\widetilde{\Phi}(\xi) := \max_{\xi' \in \Xi} \left[\Phi(\xi') - L_{\xi,\xi}\|\xi' - \xi\|^2\right]. \tag{18}$$

To converge to a stationary point of $f$, we add the second phase (called *corrected phase*) which applies projected stochastic GDA to the following corrected objective.

$$\min_{\theta \in \Theta} \max_{\xi \in \Xi} \widetilde{f}(\theta, \xi) := f(\lambda_{\theta,\xi}) - L_{\xi,\xi}\|\xi - \widetilde{\xi}\|^2. \tag{19}$$

---

[1]The convergence rate of $\|\nabla\widetilde{\Phi}(\widetilde{\xi})\|$ is proved in [31] when $f(\lambda_{\theta,\xi})$ is a convex function of $\theta$, and will be proved in Appendix N.2 for our Theorem 2 when $f(\lambda)$ is convex.

---

**Algorithm 1** Projected Stochastic Gradient Descent Ascent Algorithm For Convex Utility

---

1: **Hyperparameters:** $K, T, K', T', \alpha, \beta, a, b, L_{\xi,\xi}, \{m_\lambda^{(k)}, H_\lambda^{(k)}, m_\theta^{(k)}, H_\theta^{(k)}\}_{k=1}^4$.
2: **Initialize:** $\xi_0 \in \Xi, \theta_0, \theta_K \in \Theta$.
   *# Begin original phase to solve the original optimization problem* (2)
3: **for** Iterations $k = 0, 1, \ldots, K - 1$ **do**
4:     Initialize $\theta_{k,0} \leftarrow \theta_0$.
5:     **for** Inner steps $t = 0, 1, \ldots, T - 1$ **do**
6:         Obtain $g_{k,t}^{(\theta)} \approx \nabla_\theta f(\lambda_{\theta_{k,t},\xi_k})$ by Algorithm 3 with hyperparameters $m_\lambda^{(1)}, H_\lambda^{(1)}, m_\theta^{(1)}, H_\theta^{(1)}$.
7:         Apply the following projected stochastic policy gradient descent step.
$$\theta_{k,t+1} = \text{proj}_\Theta\big(\theta_{k,t} - \alpha g_{k,t}^{(\theta)}\big). \tag{14}$$

8:     **end for**
9:     Assign $\theta_k \leftarrow \theta_{k,T}$.
10:    Obtain $g_k^{(\xi)} \approx \nabla_\xi f(\lambda_{\theta_k,\xi_k})$ by Algorithm 3 with hyperparameters $m_\lambda^{(2)}, H_\lambda^{(2)}, m_\xi^{(2)}, H_\xi^{(2)}$.
11:    Apply the following projected stochastic gradient descent step.
$$\xi_{k+1} = \text{proj}_\Xi\big(\xi_k + \beta g_k^{(\xi)}\big). \tag{15}$$

12: **end for**
13: Obtain $\widetilde{\xi}$ from $\{\xi_k\}_{k=0}^{K-1}$ uniformly at random.
   *# Begin corrected phase to solve the corrected optimization problem* (19)
14: **for** Iterations $k = K, K + 1, \ldots, K + K' - 1$ **do**
15:    Initialize $\xi_{k,0} \leftarrow \widetilde{\xi}$.
16:    **for** Inner steps $t = 0, 1, \ldots, T' - 1$ **do**
17:        Obtain $g_{k,t}^{(\xi)} \approx \nabla_\xi f(\lambda_{\theta_k,\xi_{k,t}})$ by Algorithm 3 with hyperparameters $m_\lambda^{(3)}, H_\lambda^{(3)}, m_\xi^{(3)}, H_\xi^{(3)}$.
18:        Apply the following projected stochastic gradient ascent step.
$$\xi_{k,t+1} = \text{proj}_\Xi\big[\xi_{k,t} + a\big(g_{k,t}^{(\xi)} - 2L_{\xi,\xi}(\xi_{k,t} - \widetilde{\xi})\big)\big]. \tag{16}$$

19:    **end for**
20:    Assign $\xi_k \leftarrow \xi_{k,T'}$.
21:    Obtain $g_k^{(\theta)} \approx \nabla_\theta f(\lambda_{\theta_k,\xi_k})$ by Algorithm 3 with hyperparameters $m_\lambda^{(4)}, H_\lambda^{(4)}, m_\theta^{(4)}, H_\theta^{(4)}$.
22:    Apply the following projected stochastic gradient descent step.
$$\theta_{k+1} = \text{proj}_\Theta\big(\theta_k - b g_k^{(\theta)}\big). \tag{17}$$

23: **end for**
24: **Output:** $(\theta_{\widetilde{k}}, \xi_{\widetilde{k}})$ where $\widetilde{k}$ is obtained from $\{K, K + 1, \ldots, K + K' - 1\}$ uniformly at random.

---

The convergence analysis of Algorithm 1 is challenging largely because $f(\lambda_{\theta,\xi})$ is only a convex function of $\lambda_{\theta,\xi}$ not of $\theta$. To tackle this challenge for non-robust convex RL with fixed $\xi$, [54] assumed that a global Lipschitz continuous inverse mapping from $\lambda_{\theta,\xi}$ to $\theta$ exists. [55, 6] relaxed this assumption to the following assumption of local inverse mapping, which covers the popular direct policy parameterization $\pi_\theta(a|s) = \theta_{s,a}$ [6] and softmax policy parameterization $\pi_\theta(a|s) = \frac{\exp(\theta_{s,a})}{\sum_{a'} \exp(\theta_{s,a'})}$ (see Proposition 8 for the proof).

**Assumption 4** (Local Invertibility of $\lambda_{\cdot,\xi}$). *There exists constants $\ell_{\lambda^{-1}} > 0$ and $\overline{\delta} \in (0, 1)$ such that for any fixed $\theta \in \Theta$ and $\xi \in \Xi$, the occupancy measure (1) satisfies:*
*1. There exists sets $\mathcal{U}_{\theta,\xi} \subset \Theta$ and $\mathcal{V}_{\lambda_{\theta,\xi}} \subset \Delta^{\mathcal{S} \times \mathcal{A}}$ that contain $\theta$ and $\lambda_{\theta,\xi}$ respectively, such that $\lambda_{\theta,\xi} : \mathcal{U}_{\theta,\xi} \to \mathcal{V}_{\lambda_{\theta,\xi}}$ is a bijection. Its inverse denoted as $\lambda_{\theta,\xi}^{-1}$ is $\ell_{\lambda^{-1}}$-Lipscthiz.*
*2. There exists at least one optimal policy $\theta^*(\xi) \in \arg\min_{\theta' \in \Theta} f(\lambda_{\theta',\xi})$ such that for any $\delta \in [0, \overline{\delta}]$, $(1 - \delta)\lambda_{\theta,\xi} + \delta\lambda_{\theta^*(\xi),\xi} \in \mathcal{V}_{\lambda_{\theta,\xi}}$.*

**Proposition 4** (Projected Gradient Dominance for Convex Utility). *Under Assumptions 1-4, the utility function $f$ satisfies the following gradient dominance property for any $\theta \in \Theta$ and $\xi \in \Xi$.*

$$f(\lambda_{\theta,\xi}) - \min_{\theta' \in \Theta} f(\lambda_{\theta',\xi}) \leq \big[\sqrt{2}\ell_{\lambda^{-1}}\big(\beta L_{\theta,\theta} + 1\big) + \beta\ell_\theta\big]\|G_\beta^{(\theta)}(\theta,\xi)\|. \tag{20}$$

**Remark:** Proposition 4 indicates that the function $f(\lambda_{\cdot,\xi})$ is projected gradient dominant for convex utility function $f$, which is important in the convergence analysis of Algorithm 1. Our Proposition 4 is stronger than Lemma F.7 of [6], a similar gradient dominance property for convex RL which requires assumption of positive definite Fisher information matrix, involves bias in the error term and focuses on unconstrained optimization with softmax parameterized policy (a special of our general parameterized policy with constrained variable $\theta \in \Theta$).

**Technical Novelty.** In our proof, to tackle the constraint $\theta \in \Theta$ which is more challenging than the unconstrained case $\Theta = \mathbb{R}^{|\mathcal{S}||\mathcal{A}|}$ in [55, 6], we apply Assumption 4 to $\theta' := \theta - \beta G_\beta^{(\theta)}(\theta, \xi)$ not to the obvious choice $\theta$, which yields $\theta_\delta \in \Theta$ for any $\delta \in [0, \bar{\delta}]$ such that $\lambda_{\theta_\delta,\xi} = (1-\delta)\lambda_{\theta',\xi} + \delta\lambda_{\theta^*(\xi),\xi} \in \mathcal{V}_{\lambda_{\theta,\xi}}$. Then

$$\nabla_\theta f(\lambda_{\theta',\xi})^\top(\theta_\delta - \theta') \overset{(i)}{\geq} \left[\nabla_\theta f(\lambda_{\theta',\xi}) - \nabla_\theta f(\lambda_{\theta,\xi}) + G_\beta^{(\theta)}(\theta, \xi)\right]^\top(\theta_\delta - \theta') \overset{(ii)}{\geq} -\mathcal{O}[\delta\|G_\beta^{(\theta)}(\theta, \xi)\|],$$

where (i) uses the projection property $(\theta_\delta - \theta')^\top[G_\beta^{(\theta)}(\theta, \xi) - \nabla_\theta f(\lambda_{\theta,\xi})] \leq 0$ and (ii) uses $\|\theta_\delta - \theta'\| \leq \mathcal{O}(\delta)$. The above bound implies Eq. (20) since $f$ is convex and $\ell_\theta$-Lipscthiz.

**Assumption 5.** $\Xi$ *is convex and compact with diameter* $D_\Xi := \max_{\xi,\xi'\in\Xi} \|\xi' - \xi\| > 0$.

Assumption 5 holds for the commonly used direct kernel parameterization $p_\xi(s'|s, a) = \xi(s, a, s')$ (for all $s, s' \in \mathcal{S}$ and $a \in \mathcal{A}$)[42, 28, 26, 17] and $\Xi$ defined a compact neighborhood around a nominal transition kernel parameter $\widehat{\xi}$.

We show the gradient convergence result of Algorithm 1 by the following theorem and demonstrate the gradient convergence by the experiments in Appendix A.

**Theorem 2** (Gradient Convergence for Convex Utility)**.** *Suppose Assumptions 1-5 hold. For any precision* $0 < \epsilon \leq \frac{48L_{\xi,\xi}}{\widetilde{L}}\left[\sqrt{2}\ell_{\lambda^{-1}}\left(L_{\theta,\theta} + 4\widetilde{L}\right) + \ell_\theta\right]$, *we can always find proper hyperparameter values of Algorithm 1 (see Eqs. (127)-(150) in Appendix N.6 for these hyperparamter values) such that the algorithm output* $(\theta_{\widetilde{k}}, \xi_{\widetilde{k}})$ *is an* $\epsilon$*-close to a stationary point, that is,* $\mathbb{E}[\|G_b^{(\theta)}(\theta_{\widetilde{k}}, \xi_{\widetilde{k}})\|^2] \leq \epsilon^2$ *and* $\mathbb{E}[\|G_a^{(\xi)}(\theta_{\widetilde{k}}, \xi_{\widetilde{k}})\|^2] \leq \epsilon^2$ *with projected gradients* $G_b^{(\theta)}$ *and* $G_a^{(\xi)}$ *defined in Eq. (13). The number of required stochastic samples is* $\mathcal{O}\left[\frac{\log^2[(1-\gamma)^{-1}\epsilon^{-1}]}{(1-\gamma)^{25}\epsilon^{10}}\right]$.

**Proof Sketch of Theorem 2 and Technical Novelty.** Inspired by Appendix D of [31], $\widetilde{\xi}$ from the first phase satisfies $\mathbb{E}\|\nabla\widetilde{\Phi}(\widetilde{\xi})\|^2 \to 0$ (see Appendix N.2). Then, $\xi_k := \xi_{k,T'}$ from the inner update (16) of the second phase converges to the unique maximizer (denoted as $\xi_k^*$) of the $L_{\xi,\xi}$-concave function $\widetilde{f}(\theta_k, \cdot)$ as $T' \to +\infty$ (see Appendix N.3). This means the update step (17) is approximately projected gradient descent for $\min_{\theta\in\Theta} \widetilde{\Psi}(\theta)$, which yields the convergence rate of $\mathbb{E}\left[\|G_b^{(\theta)}(\theta_{\widetilde{k}}, \xi_{\widetilde{k}})\|^2\right]$ (see Appendix N.4).

However, the biggest challenge is to obtain the convergence rate of $\mathbb{E}\left[\|G_a^{(\xi)}(\theta_{\widetilde{k}}, \xi_{\widetilde{k}})\|^2\right]$ (see Appendix N.5), which corresponds to $\nabla_\xi f$ while the second *corrected phase* aims at the corrected objective $\widetilde{f}$. To show that $\nabla_\xi\widetilde{f}(\theta_k, \xi_k) \approx \nabla_\xi f(\theta_k, \xi_k)$, note that $\nabla_\xi\widetilde{f}(\theta_k, \xi_k) - \nabla_\xi f(\theta_k, \xi_k) = -2L_{\xi,\xi}(\xi_k - \widetilde{\xi})$ and that $\nabla\widetilde{\Phi}(\widetilde{\xi}) = 2L_{\xi,\xi}[\xi^*(\widetilde{\xi}) - \widetilde{\xi}] \approx 0$ (already proved) where $\xi^*(\widetilde{\xi})$ is the unique maximizer of $\Phi(\xi') - L_{\xi,\xi}\|\xi' - \widetilde{\xi}\|^2$, a strongly concave function of $\xi'$ in Eq. (18). Hence, it suffices to show $\xi_k \approx \xi^*(\widetilde{\xi})$. Note that $(\theta_k, \xi_k)$ is an **approximate Nash equilibrium** of $\widetilde{f}$, i.e., $\xi_k \approx \xi_k^* := \arg\max_{\xi\in\Xi}\widetilde{f}(\theta_k, \xi)$ (proved above) and $\theta_k \approx \arg\min_{\theta\in\Theta}\widetilde{f}(\theta, \xi_k)$ (derived below).

$$\mathbb{E}[\widetilde{f}(\theta_k, \xi_k) - \min_{\theta'\in\Theta}\widetilde{f}(\theta', \xi_k)] = \mathbb{E}[f(\lambda_{\theta_k,\xi_k}) - \min_{\theta'\in\Theta}f(\lambda_{\theta',\xi_k})] \overset{(i)}{\leq} \mathcal{O}(\mathbb{E}\|G_\beta^{(\theta)}(\theta, \xi)\|) \leq \mathcal{O}(\epsilon),$$

where (i) uses Proposition 4. Hence, based on the **property of Nash equilibrium**, we have $\xi_k \approx \arg\max_\xi \psi(\xi) = \xi^*(\widetilde{\xi})$ where $\psi(\xi) := \min_{\theta\in\Theta}\widetilde{f}(\theta, \xi) = \Phi(\xi) - L_{\xi,\xi}\|\xi - \widetilde{\xi}\|^2$.

## 4 Global Convergence on Polyhedral Ambiguity Set

This section aims to obtain a global optimal policy $\theta^*$ that minimizes the robust utility $\Gamma(\theta) \overset{\text{def}}{=} \max_{\xi\in\Xi} f(\lambda_{\theta,\xi})$. This maximization is challenging for convex utility $f$. In contrast, global conver-

gence results have been obtained without such challenge in some important special cases, including convex RL with fixed $\xi$ [54, 51, 6] and robust RL where linear utility $f$ is amenable to both $\min_{\theta \in \Theta}$ and $\max_{\xi \in \Xi}$ [36, 45, 42, 23, 26]. Fortunately, we will show that by using the popular $s$-rectangular *polyhedral* ambiguity set $\Xi$, $\arg\max_{\xi \in \Xi} f(\lambda_{\theta,\xi})$ always exists among the finitely many vertices of $\Xi$.

## 4.1 S-rectangular Polyhedral Ambiguity Set

In this subsection, we will introduce the popular $s$-rectangular polyhedral ambiguity set, and derive its important propositions for designing globally converged algorithm.

Fhe global convergence is generally NP-hard, even for the important special case called *robust RL with linear utility*, [45]. A common practice to make the problem tractable is to use direct kernel parameterization $p_\xi(s'|s,a) = \xi(s,a,s')$ [42, 28, 26, 17] and assume the ambiguity set $\Xi$ to satisfy some certain rectangularity conditions, such as *s-rectangularity* defined below [45, 42, 23, 26].

**Assumption 6.** *We use direct kernel parameterization and assume that $\Xi$ is $s$-rectangular, i.e.,* $\Xi = \times_{s \in \mathcal{S}} \Xi_s := \{\xi \in (\Delta^{\mathcal{S}})^{\mathcal{S} \times \mathcal{A}} : \xi(s, \cdot, \cdot) \in \Xi_s, \forall s \in \mathcal{S}\}$, *a Cartesian product of $\Xi_s \subset (\Delta^{\mathcal{S}})^{\mathcal{A}}$.*

**Proposition 5** (Local Invertibility of $\lambda_{\theta,\cdot}$). *Suppose Assumption 6 holds and $\Xi$ is a convex set. For any $\theta \in \Theta$, $\xi_0, \xi_1 \in \Xi$ and $\delta \in [0,1]$, define the following kernel parameters $\xi_\delta \in (\Delta^{\mathcal{S}})^{\mathcal{S} \times \mathcal{A}}$.*

$$\xi_\delta(s,a,s') = \begin{cases} \text{arbitrary as long as } \xi_\delta(s,a,\cdot) \in \Delta^{\mathcal{S}}, & \text{if } \lambda_{\theta,\xi_0}(s) = \lambda_{\theta,\xi_1}(s) = 0 \\ \dfrac{\delta\lambda_{\theta,\xi_1}(s)\xi_1(s,a,s') + (1-\delta)\lambda_{\theta,\xi_0}(s)\xi_0(s,a,s')}{\delta\lambda_{\theta,\xi_1}(s) + (1-\delta)\lambda_{\theta,\xi_0}(s)}, \text{otherwise} \end{cases}, \quad (21)$$

*where $\lambda_{\theta,\xi}(s) := \sum_{a \in \mathcal{A}} \lambda_{\theta,\xi}(s,a)$ for any $s \in \mathcal{S}$, $\theta \in \Theta$ and $\xi \in \Xi$. Then $\xi_\delta \in \Xi$ and its corresponding occupancy measure is $\lambda_{\theta,\xi_\delta} = \delta\lambda_{\theta,\xi_1} + (1-\delta)\lambda_{\theta,\xi_0}$.*

**Remark:** Proposition 5 indicates that the mapping from $\xi$ to $\lambda_{\theta,\xi}$ is locally invertible for $s$-rectangular set $\Xi$, which is important to solve the aforementioned challenge that convex utility is not amenable for $\max_{\xi \in \Xi} f(\lambda_{\theta,\xi})$. This role is similar to that played by the local invertibility assumption (Assumption 4) for policy $\theta$. To our knowledge, Proposition 5 has never been obtained in the existing literature.

**Assumption 7.** *Under Assumption 6, for every $s \in \mathcal{S}$, $\Xi_s$ is a polyhedron spanned by a finite set of vertices $V(\Xi_s) := \{\xi_m^{(s)}\}_{m=1}^{M_s} \subset \Xi_s$, i.e., $\Xi_s = \big\{ \sum_{m=1}^{M_s} \nu_m \xi_m^{(s)} : \nu_m \geq 0, \sum_{m=1}^{M_s} \nu_m = 1 \big\}$.*

Polyhedral ambiguity set defined by Assumption 7 includes the widely used $s$-rectangular $L_1$ and $L_\infty$ ambiguity sets, defined as $\Xi = \{\xi \in (\Delta^{\mathcal{S}})^{\mathcal{S} \times \mathcal{A}} : \|\xi(s,:,:) - \widehat{\xi}(s,:,:)\|_p \leq \alpha_s, \forall s \in \mathcal{S}\}$ for $p \in \{1, \infty\}$ respectively [7, 19, 16], where $\widehat{\xi} \in \Xi$ is the nominal transition kernel usually obtained via empirical estimation. On polyhedral ambiguity set, the optimal kernels $\arg\max_{\xi \in \Xi} f(\lambda_{\theta,\xi})$ can always be obtained at the vertices of $\Xi$, as shown below.

**Proposition 6.** *Under Assumptions 3, 6 and 7, for any $\theta \in \Theta$, we have $\max_{\xi \in \Xi} f(\lambda_{\theta,\xi}) = \max_{\xi \in V(\Xi)} f(\lambda_{\theta,\xi})$, where $V(\Xi) = \times_{s \in \mathcal{S}} V(\Xi_s)$ is the vertex set.*

**Technical Novelty.** Suppose a non-vertex kernel $\xi^* \in \arg\max_{\xi \in \Xi} f(\lambda_{\theta,\xi}) / V(\Xi)$ is optimal. Since $f$ is convex, if $\lambda_{\theta,\xi^*}$ is a convex combination of $\lambda_{\theta,\xi_1^*}$ and $\lambda_{\theta,\xi^{(\epsilon)}}$ for some $\xi_1, \xi_0 \in \Xi$ (corresponding to $\xi_1^*, \xi^{(\epsilon)}$ respectively in the proof in Appendix I), then $\xi_1, \xi_0$ are also optimal. Ideally, if $\xi_1 \in V(\Xi)$ or $\xi_0 \in V(\Xi)$, the proof is done. However, this is not guaranteed since in Proposition 5 and Assumption 6, the convex combination coefficients differ among the states $s \in \mathcal{S}$. To solve this challenge, it suffices to find such optimal $\xi_1$ that differs from $\xi^*$ at only one state $s$ such that the non-vertex $\xi^*(s) \notin V(\Xi_s)$ is replaced with vertex $\xi_1(s) \in V(\Xi_s)$. Then we can conduct such change from non-vertex to vertex for **only one state $s$ at a time** until the kernel becomes vertex at every state, while keeping the optimality all the way. To find such $\xi_1(s)$, note that on polyhedral set $\Xi_s$, there always exist $\xi_1(s) \in V(\Xi_s)$ and $\xi_0(s) \in \Xi_s$ such that the non-vertex point $\xi^*(s)$ is a convex combination of $\xi_1(s)$ and $\xi_0(s)$, while $\xi^*(s') = \xi_1(s') = \xi_0(s')$ for any $s' \neq s$. Hence, there exists $\delta \in [0,1]$ such that $\xi^* = \xi_\delta$ defined by Proposition 5, which implies that $\lambda_{\theta,\xi^*}$ is a convex combination of $\lambda_{\theta,\xi_1^*}$ and $\lambda_{\theta,\xi^{(\epsilon)}}$.

## 4.2 Globally Converged Algorithm

The original objective (2) is equivalent to the minimization problem $\min_{\theta \in \Theta} \Gamma(\theta)$, where $\Gamma(\theta) := \max_{\xi \in V(\Xi)} f(\lambda_{\theta,\xi})$ with finite vertex set $V(\Xi)$ based on Proposition 6. A natural choice to solve this minimization problem is the following policy update rule (for simplicity we consider the unconstrained policy space $\Theta = \mathbb{R}^{d_\Theta}$ as in [55, 6]).

$$\theta_{k+1} = \theta_k - \beta_k d_k, \quad (22)$$

where $\beta_k > 0$ is the stepsize and $d_k$ is a unit descent direction of $\Gamma(\theta_k)$. Subgradient descent method seems an obvi-

**Algorithm 2** Globally Converged Algorithm for Convex Utility on Polyhedral Ambiguity Set

1: **Hyperparameters:** $K$, $\{\sigma_k, \epsilon_k, \beta_k\}_{k=0}^{K-1}$.
2: **Initialize:** $\theta_0 \in \Theta$.
3: **for** Iterations $k = 0, 1, \ldots, K-1$ **do**
4:     Calculate $\lambda_{\theta_k,\xi}$, $f(\lambda_{\theta_k,\xi})$ and $\nabla_\theta f(\lambda_{\theta_k,\xi})$ for all $\xi \in V(\Xi)$.
5:     Select near-optimal vertices $\Xi_k := \{\xi \in V(\Xi) : f(\lambda_{\theta_k,\xi}) \geq \max_{\xi' \in V(\Xi)} f(\lambda_{\theta_k,\xi'}) - \sigma_k\}$.
6:     Find $d'_k \in B_1 := \{d \in \mathbb{R}^{d_\Theta} : \|d\| \leq 1\}$ such that
        $A_k(d'_k) \leq \min_{d \in B_1} A_k(d) + \epsilon_k$.
    ($A_k$ *is defined in Eq.* (23). *One way to solve* $\min_{d \in B_1} A_k(d)$ *is to apply projected subgradient method* (28) *and obtain* $d'_k \in \arg\max_{d \in \{d_{k,t}: 0 \leq t \leq T\}} A_k(d)$.)
7:     Let $d_k := d'_k / \|d'_k\|$ and obtain $\theta_{k+1}$ by Eq. (22).
8: **end for**
9: **Output:** $(\theta_K, \xi_K)$ where $\xi_K \in \arg\max_{\xi \in V(\Xi)} f(\lambda_{\theta_K,\xi})$.

ous choice for $d_k$ which aligns with the direction of a subgradient $\nabla_\theta f(\lambda_{\theta_k,\xi_k})$ where $\xi_k \in \arg\max_{\xi \in V(\Xi)} f(\lambda_{\theta_k,\xi})$. However, the convergence analysis of subgradient descent method [11] requires the convexity of $f(\lambda_{\cdot,\xi_k})$ which does not hold in our setting, and the function value is not monotonically decreasing. To solve these challenges, we design Algorithm 2 which selects near-optimal vertices $\Xi_k := \{\xi \in V(\Xi) : f(\lambda_{\theta_k,\xi}) \geq \max_{\xi' \in V(\Xi)} f(\lambda_{\theta_k,\xi'}) - \sigma_k\}$ with a certain threshold $\sigma_k > 0$ and obtains $d_k$ by solving the convex optimization problem $\min_{d \in B_1} A_k(d)$ up to precision $\epsilon_k > 0$, where $A_k(d)$ below denotes effective descent of $\Gamma(\theta_k)$ along the direction $d$.

$$A_k(d) := \max_{\xi \in \Xi_k} \left[ \nabla_\theta f(\lambda_{\theta_k,\xi})^\top d \right], d \in B_1 := \{d' \in \mathbb{R}^{d_\Theta} : \|d'\| \leq 1\}. \quad (23)$$

Here we only care about the near-optimal vertices in $\Xi_k \subset V(\Xi)$ because for any worse vertices $\xi \in V(\Xi)/\Xi_k$, $f(\theta_k, \xi) < \max_{\xi' \in V(\Xi)} f(\lambda_{\theta_k,\xi'}) - \sigma_k$ implies $f(\theta_{k+1}, \xi) < \max_{\xi' \in V(\Xi)} f(\lambda_{\theta_{k+1},\xi'})$ for appropriate $\sigma_k > 0$. This means such worse $\xi$ can not affect the optimization progress $\Gamma(\theta_k) - \Gamma(\theta_{k+1})$. Hence, by solving $\min_{d \in B_1} A_k(d)$, we can obtain a direction $d_k$ in which all the potentially effective function values $\{f(\lambda_{\theta_k,\xi})\}_{\xi \in \Xi_k}$ have uniformly large amount of descent $-\nabla f(\lambda_{\theta_k,\xi})^\top d_k$.

To analyze the global convergence of Algorithm 2, we want to guarantee sufficient descent $\Gamma(\theta_k) - \Gamma(\theta_{k+1})$ whenever $\theta_k$ is not close to optimal. It suffices to slightly alter Assumption 4 as follows.

**Assumption 8.** *A variant of Assumption 4 holds which replaces the non-robust optimal policy* $\theta^*(\xi)$ *with a robust optimal policy* $\theta^* \in \arg\min_{\theta \in \Theta} \Gamma(\theta)$ *and shrinks the range from* $\xi \in \Xi$ *to* $\xi \in V(\Xi)$.

**Remark:** Assumption 8 is no stronger than Assumption 4 and also covers the popular direct policy parameterization. Also, Assumption 8 guarantees that from any policy $\theta \in \Theta$, there exists a partial curve $\{\theta_\delta : \delta \in [0, \bar{\delta}]\}$ towards a robust optimal policy $\theta^*$ such that $\lambda_{\theta_\delta,\xi} = (1-\delta)\lambda_{\theta,\xi} + \delta\lambda_{\theta^*,\xi}$, so we can utilize convexity of $f$ and obtain the following important sufficient descent property.

**Proposition 7** (Sufficient Descent on Polyhedral Ambiguity Set). *Under Assumptions 1-3 and 8, at any* $\theta \in \Theta := \mathbb{R}^{d_\Theta}$, *there exists a unit descent direction* $d$ *(*$\|d\| = 1$*) such that*

$$f(\lambda_{\theta,\xi}) - f(\lambda_{\theta^*,\xi}) \leq \left[ -\sqrt{2}\ell_{\lambda^{-1}} \nabla_\theta f(\lambda_{\theta,\xi})^\top d \right]_+, \forall \xi \in \Xi \quad (24)$$

*where* $\theta^* \in \arg\min_{\theta' \in \Theta} \Gamma(\theta)$ *is given by Assumption 8 and* $x_+ := \max(x, 0)$ *for any* $x \in \mathbb{R}$.

**Remark:** $d$ in Proposition 7 is a good descent direction since whenever the function value gap $f(\lambda_{\theta,\xi}) - f(\lambda_{\theta^*,\xi}) > 0$, it is dominated by the gradient descent amount $-\nabla_\theta f(\lambda_{\theta,\xi})^\top d > 0$. Unlike existing gradient dominance properties for robust RL [42, 26, 17], $f(\lambda_{\theta,\xi}) - f(\lambda_{\theta^*,\xi}) \leq 0$ is possible so we use $[\cdot]_+$ to cover all cases. This brings challenge and thus novel techniques to obtain the first global convergence result of our robust RL with general convex utility as follows.

**Theorem 3** (Global Convergence for Convex Utility on Polyhedral Ambiguity Set). *Implement Algorithm 2 with* $\beta_k = \frac{2\sqrt{2}\ell_\lambda}{k+2}$, $\sigma_k = \frac{4\sqrt{2}\ell_\theta \ell_\lambda}{k+2}$ *and any* $\epsilon_k > 0$. *Then under Assumptions 1-3, 6-8, the*

*algorithm output $\theta_K$ has the following global convergence rate.*

$$\Gamma(\theta_K) - \min_{\theta' \in \Theta} \Gamma(\theta') \leq \sqrt{2}\ell_{\lambda^{-1}} \max_{1 \leq k \leq K} \epsilon_k + \frac{4\ell_{\lambda^{-1}}}{K+1}(\ell_{\lambda^{-1}} L_{\theta,\theta} + 2\sqrt{2}\ell_\theta). \tag{25}$$

**Remark:** The convergence rate $\mathcal{O}(1/K)$ matches the state-of-the-art of policy gradient type methods for robust RL [26], while the error term $\epsilon_k$ results from solving the convex optimization problem $\min_{d \in B_1} A_k(d)$ in line 6 of Algorithm 2.

**Technical Novelty.** Applying Proposition 7 to Algorithm 2 with $\sigma_k = 2\beta_k\ell_\theta$, we have

$$\Delta_k := \Gamma(\theta_k) - \min_{\theta' \in \Theta} \Gamma(\theta') \leq \left[\sqrt{2}\ell_{\lambda^{-1}}[\epsilon_k - A_k(d'_k)]\right]_+ + 2\beta_k\ell_\theta. \tag{26}$$

To overcome the main difficulty caused by $[\cdot]_+$ above, we analyze each $k$-th iteration in **2 cases** $A_k(d'_k) \geq 0$ and $A_k(d'_k) < 0$. If $A_k(d'_k) \geq 0$, then $\Delta_k \leq \sqrt{2}\ell_{\lambda^{-1}}\epsilon_k + 2\beta_k\ell_\theta$ and thus $\Delta_{k+1} \leq \sqrt{2}\ell_{\lambda^{-1}}\epsilon_k + 3\beta_k\ell_\theta$; If $A_k(d'_k) < 0$, then in Eq. (26) we replace $A_k(d'_k)$ with $A_k(d_k) \leq A_k(d'_k) < 0$ and remove $[\cdot]_+$. This along with $\Gamma(\theta_{k+1}) - \Gamma(\theta_k) \leq \beta_k A_k(d_k) + \frac{L_{\theta,\theta}}{2}\beta_k^2$ (by smoothness) implies

$$\Delta_{k+1} \leq \frac{k}{k+2}\Delta_k + \mathcal{O}\left[\frac{\epsilon_k}{k+2} + \frac{1}{(k+2)^2}\right]. \tag{27}$$

Then we obtain the rate (25) in **3 cases**: If $A_k(d'_k) < 0$ for all $k = 0, 1, \ldots, K-1$, iterate Eq. (27) from $\Delta_0$; If $A_{K-1}(d_{K-1}) \geq 0$, $\Delta_K \leq \sqrt{2}\ell_{\lambda^{-1}}\epsilon_K + \sigma_k$; If $A_{K'-1}(d_{K'-1}) \geq 0$ while $A_k(d_k) < 0$ for all $k = K', \ldots, K-1$, iterate Eq. (27) from $\Delta_{K'} \leq \sqrt{2}\ell_{\lambda^{-1}}\epsilon_{K'-1} + 3\beta_{K'-1}\ell_\theta$.

Algorithm 2 involves convex optimization problems $\min_{d \in B_1} A_k(d)$, which can be solved via the following projected subgradient method for $t = 0, 1, \ldots, T-1$.

$$d_{k,t+1} \leftarrow \text{proj}_{B_1}[d_{k,t} - \alpha\nabla_\theta f(\lambda_{\theta_k,\xi_{k,t}})], \text{ where } \xi_{k,t} \in \arg\max_{\xi \in \Xi_k}\nabla_\theta f(\lambda_{\theta_k,\xi})^\top d_{k,t}. \tag{28}$$

The best direction $d'_k \in \arg\max_{d \in \{d_{k,t}: 0 \leq t \leq T\}} A_k(d)$ from the above subgradient method achieves $\epsilon_k$ accuracy within $T = \mathcal{O}(\epsilon_k^{-2})$ steps [11], which yields the following complexity result.

**Corollary 1.** *Under the conditions of Theorem 3, for any $\epsilon > 0$, implement Algorithm 2 with $K = 8\ell_{\lambda^{-1}}\epsilon^{-1}(\ell_{\lambda^{-1}} L_{\theta,\theta} + 2\sqrt{2}\ell_\theta)$ iterations and $T = 36\ell_{\lambda^{-1}}^2\ell_\theta^2\epsilon^{-2}$ subgradient descent updates* (28) *with stepsize $\alpha = \frac{\epsilon}{3\ell_{\lambda^{-1}}\ell_\theta^2}$ to obtain $d'_k$. Then the output $\theta_K$ achieves $\Gamma(\theta_K) - \min_{\theta' \in \Theta}\Gamma(\theta') \leq \epsilon$.*

Finally, we can prove that all these Assumptions 1-8 required by our convergence results (Theorems 2 and 3) can be satisfied by the following examples.

**Proposition 8.** *Assumptions 1-8 are all satisfied if we use the following choices:*
• *Softmax policy parameterization $\pi_\theta(a|s) = \frac{\exp(\theta_{s,a})}{\sum_{a'}\exp(\theta_{s,a'})}$, where $\theta \in \Theta = [-R, R]^{|\mathcal{S}| \times |\mathcal{A}|}$ for some constant $R > 0$ to prevent $\pi_\theta(a|s)$ from approaching 0.*
• *Direct kernel parameterization $p_\xi(s'|s,a) = \xi_{s,a,s'}$ with s-rectangular $L_1$ or $L_\infty$ ambiguity sets defined as $\Xi = \{\xi \in (\Delta^{\mathcal{S}})^{\mathcal{S} \times \mathcal{A}} : \|\xi(s,:,:) - \widehat{\xi}(s,:,:)\|_p \leq \alpha_s, \forall s \in \mathcal{S}\}$ for $p \in \{1, \infty\}$ respectively, where the fixed nominal kernel $\widehat{\xi}$ satisfies $\widehat{\xi}(s,a,s') > \alpha_s, \forall s,a,s'$ to prevent $p_\xi(s'|s,a)$ from approaching 0.*
• *The utility function $f(\lambda)$ defined in Eq. (7) for robust entropy regularized RL and its special cases, within the range $\lambda \in \Lambda = \{\lambda_{\theta,\xi} : \theta \in \Theta, \xi \in \Xi\}$ for the domains $\Theta$ and $\Xi$ selected above.*

## 5 Conclusion

In this work, we propose robust RL with general utility, the first learning framework that obtains a robust policy for RL with general utility. We propose a stochastic policy gradient type algorithm for convex utilities and obtains its sample complexity result for gradient convergence. Furthermore, for convex utility on polyhedral ambiguity set, we propose an alternative policy gradient type algorithm and obtain its global convergence rate. Note that this globally converged algorithm requires enumeration among many vertices, and thus it is an important future direction to reduce enumeration by utilizing structural properties. In addition, to extend the results to large or continuous state-action space is also an interesting direction.

## Acknowledgments

This work was partially supported by NSF IIS 2347592, 2347604, 2348159, 2348169, DBI 2405416, CCF 2348306, CNS 2347617.

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

# Appendix

## Table of Contents

## A  Experiments

In this section, we present simulation results of Algorithm 1 for convex utility.

**Simulation Setting.** We choose $\mathcal{S} = \{1, 2, \cdots, S\}$ with $S = 10$ states and $\mathcal{A} = \{1, 2, \cdots, A\}$ with $A = 5$ actions. The discount factor is $\gamma = 0.95$ and we select uniform distribution as the

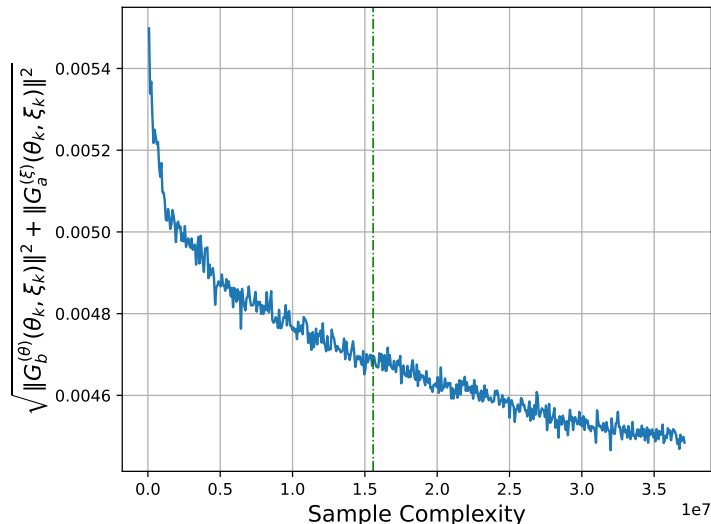

Figure 1: Numerical Experimental Result (the green vertical line denotes the transition from Phase I to Phase II of Algorithm 1).

initial state distribution $\rho$. To optimize the objective function (2), we apply direct parameterization to policy parameter $\theta_{s,a} = \pi(a|s) \in \Theta = (\Delta^{\mathcal{A}})^{\mathcal{S}}$ and transition kernel parameter $\xi_{s,a,s'} = p(s'|s,a) \in (\Delta^{\mathcal{S}})^{\mathcal{S} \times \mathcal{A}}$. In order to preserve $\overline{\xi}(:,:,s') \in \Delta^{\mathcal{S}}$, We select nominal kernel $\overline{\xi}(\cdot,\cdot,s')$ as $\frac{|10+\varepsilon_{s'}|}{\sum_{s'}|10+\varepsilon_{s'}|}$, where $\varepsilon_{s'} \overset{\text{i.i.d}}{\sim} \mathcal{N}(0,1)$ for each $s' \in \mathcal{S}$. Then we select sufficiently small radius $r = 0.01 < \min_{s,a,s'} \overline{\xi}_{s,a,s'}$ and use the $L^2$ ambiguity set $\Xi := \{\xi : \|\xi(s,:,:) - \overline{\xi}(s,:,:)\| \le r\}$ (for transition kernel) such that all $\xi \in \Xi$ have all positive entries. As for the general utility function $f$, we use the following convex entropy function with application to exploration (Example 2.2 of [54]).

$$\min_{\theta \in \Theta} \max_{\xi \in \Xi} f(\lambda_{\theta,\xi}) := -\sum_s \lambda_{\theta,\xi}(s) \log \lambda_{\theta,\xi}(s) \tag{29}$$

where $\lambda_{\theta,\xi}(s) := \sum_{a \in \mathcal{A}} \lambda_{\theta,\xi}(s,a)$ denotes the state visitation measure for any $s \in \mathcal{S}$, $\theta \in \Theta$ and $\xi \in \Xi$.

**Hyperparameters.** For Algorithm 1, we use the following hyperparameters obtained from fine-tuning but not from Theorem 2: $K = 200$, $T = 25$, $K' = 300$, $T' = 25$, $\alpha = 0.002$, $\beta = 0.001$, $a = 0.002$, $b = 0.002$, $L_{\xi,\xi} = 20$, $m_\lambda^{(1)} = 15$, $H_\lambda^{(1)} = 100$, $m_\theta^{(1)} = 15$, $H_\theta^{(1)} = 100$, $m_\lambda^{(2)} = 15$, $H_\lambda^{(2)} = 100$, $m_\xi^{(2)} = 15$, $H_\xi^{(2)} = 100$, $m_\lambda^{(3)} = 10$, $H_\lambda^{(3)} = 100$, $m_\xi^{(3)} = 10$, $H_\xi^{(3)} = 100$, $m_\lambda^{(4)} = 10$, $H_\lambda^{(4)} = 100$, $m_\theta^{(4)} = 10$, $H_\theta^{(4)} = 100$.

**Environment.** The experiment is implemented on Python 3.8 on AMD EPYC-7313 CPU with 3.00GHz, which costs about 1.5 hours in total.

**Results.** The numerical result of Algorithm 1 is shown in Figure 1. Here the y-axis is the norm of the true projected gradient $\sqrt{\|G_b^{(\theta)}(\theta_k, \xi_k)\|^2 + \|G_a^{(\xi)}(\theta_k, \xi_k)\|^2}$ at each outer iteration $k$ of both phases of Algorithm 1 (separated by the green vertical dashed line), and the x-axis is the sample complexity (i.e., the total number of generated samples up to iteration $k$). Figure 1 shows that the projected gradient decays and converges to a small value, which matches Theorem 2.

## B   Supporting Lemmas

**Lemma 1.** *Under Assumption 1, for any $s, s' \in \mathcal{S}$, $a \in \mathcal{A}$, $\theta, \theta' \in \Theta$, $\xi, \xi' \in \Xi$, we have*

$$|\pi_{\theta'}(a|s) - \pi_\theta(a|s)| \le \ell_{\pi_\theta} \|\theta' - \theta\|, \quad |p_{\xi'}(s'|s,a) - p_\xi(s'|s,a)| \le \ell_{p_\xi} \|\xi' - \xi\|, \tag{30}$$

$$\left\| \nabla_\xi p_{\xi'}(s'|s,a) - \nabla_\xi p_\xi(s'|s,a) \right\| \le \left[ L_{p_\xi} p_{\xi'}(s'|s,a) + \ell_{p_\xi}^2 \right] \|\xi' - \xi\|. \tag{31}$$

*Proof.* Based on Assumption 1, the following inequalities holds for all $s, s' \in \mathcal{S}$, $a \in \mathcal{A}$, $\theta \in \Theta$, $\xi \in \Xi$, which by Lagrange mean value theorem directly proves Eq. (30)

$$\|\nabla_\theta \pi_\theta(a|s)\| \leq \|\nabla_\theta \log \pi_\theta(a|s)\| \leq \ell_{\pi_\theta}, \quad \|\nabla_\xi p_\xi(s'|s,a)\| \leq \|\nabla_\xi \log p_\xi(s'|s,a)\| \leq \ell_{p_\xi}.$$

Then we prove Eq. (31) as follows.

$$\begin{aligned}
&\left\|\nabla_\xi p_{\xi'}(s'|s,a) - \nabla_\xi p_\xi(s'|s,a)\right\| \\
=&\left\|p_{\xi'}(s'|s,a)\nabla_\xi \log p_{\xi'}(s'|s,a) - p_\xi(s'|s,a)\nabla_\xi \log p_\xi(s'|s,a)\right\| \\
\leq& p_{\xi'}(s'|s,a)\left\|\nabla_\xi \log p_{\xi'}(s'|s,a) - \nabla_\xi \log p_\xi(s'|s,a)\right\| \\
&+ |p_{\xi'}(s'|s,a) - p_\xi(s'|s,a)|\left\|\nabla_\xi \log p_\xi(s'|s,a)\right\| \\
\overset{(i)}{\leq}& p_{\xi'}(s'|s,a)L_{p_\xi}\|\xi' - \xi\| + \ell_{p_\xi} \cdot \ell_{p_\xi}\|\xi' - \xi\| \\
\leq& \left[L_{p_\xi}p_{\xi'}(s'|s,a) + \ell_{p_\xi}^2\right]\|\xi' - \xi\|,
\end{aligned}$$

where (i) uses Eq. (30) and Assumption 1. $\qquad\square$

**Lemma 2.** *For any $\theta \in \Theta$ and $\xi \in \Xi$, the occupancy measure $\lambda_{\theta,\xi}$ defined by Eq. (1) is the unique solution to the following Bellman equation of $\lambda \in \mathbb{R}^{|\mathcal{S}| \times |\mathcal{A}|}$.*

$$\lambda(s',a') = \left[(1-\gamma)\rho(s') + \gamma\sum_{s,a}\lambda(s,a)p_\xi(s'|s,a)\right]\pi_\theta(a'|s'), \quad s' \in \mathcal{S}, a' \in \mathcal{A}. \tag{32}$$

*Therefore, the state occupancy measure $\lambda_{\theta,\xi}(s) := \sum_{a\in\mathcal{A}}\lambda_{\theta,\xi}(s,a)$ satisfies*

$$\lambda_{\theta,\xi}(s,a) = \lambda_{\theta,\xi}(s)\pi_\theta(a|s). \tag{33}$$

*Proof.* First, we can prove that $\lambda_{\theta,\xi}$ satisfies Eq. (32) as follows.

$$\begin{aligned}
\lambda_{\theta,\xi}(s',a') &= (1-\gamma)\sum_{t=0}^{+\infty}\gamma^t \mathbb{P}_{\pi_\theta,p_\xi}(s_t = s', a_t = a'|s_0 \sim \rho) \\
&= \pi_\theta(a'|s')(1-\gamma)\sum_{t=0}^{+\infty}\gamma^t \mathbb{P}_{\pi_\theta,p_\xi}(s_t = s'|s_0 \sim \rho) \\
&= \pi_\theta(a'|s')(1-\gamma)\left[\mathbb{P}_{\pi_\theta,p_\xi}(s_0 = s'|s_0 \sim \rho) + \gamma\sum_{t=0}^{+\infty}\gamma^t \mathbb{P}_{\pi_\theta,p_\xi}(s_{t+1} = s'|s_0 \sim \rho)\right] \\
&= \pi_\theta(a'|s')(1-\gamma)\left[\rho(s') + \gamma\sum_{t=0}^{+\infty}\gamma^t\sum_{s,a}\mathbb{P}_{\pi_\theta,p_\xi}(s_t = s, a_t = a|s_0 \sim \rho)p_\xi(s'|s,a)\right] \\
&= \pi_\theta(a'|s')\left[(1-\gamma)\rho(s') + \gamma\sum_{s,a}p_\xi(s'|s,a)\right. \\
&\quad \left.\left((1-\gamma)\sum_{t=0}^{+\infty}\gamma^t \mathbb{P}_{\pi_\theta,p_\xi}(s_t = s, a_t = a|s_0 \sim \rho)\right)\right] \\
&= \left[(1-\gamma)\rho(s') + \gamma\sum_{s,a}\lambda_{\theta,\xi}(s,a)p_\xi(s'|s,a)\right]\pi_\theta(a'|s').
\end{aligned}$$

Next, we prove the uniqueness. Suppose $\lambda_1, \lambda_2 \in \mathbb{R}^{|\mathcal{S}| \times |\mathcal{A}|}$ satisfies Eq. (32). Then we have

$$\begin{aligned}
\sum_{s',a'}|\lambda_2(s',a') - \lambda_1(s',a')| &= \sum_{s',a'}\gamma\pi_\theta(a'|s')\left|\sum_{s,a}[\lambda_2(s,a) - \lambda_1(s,a)]p_\xi(s'|s,a)\right| \\
&\leq \gamma\sum_{s'}\sum_{s,a}|\lambda_2(s,a) - \lambda_1(s,a)|p_\xi(s'|s,a) \\
&\leq \gamma\sum_{s,a}|\lambda_2(s,a) - \lambda_1(s,a)|,
\end{aligned}$$

which implies that $(1 - \gamma) \sum_{s,a} |\lambda_2(s, a) - \lambda_1(s, a)| \leq 0$ and thus $\lambda_2 = \lambda_1$, i.e., the solution to Eq. (32) is unique.

Finally, we will prove Eq. (33). Note that

$$\lambda_{\theta,\xi}(s) = \sum_{a \in \mathcal{A}} \lambda_{\theta,\xi}(s, a) \overset{(i)}{=} (1 - \gamma)\rho(s') + \gamma \sum_{s,a} \lambda(s, a)p_\xi(s'|s, a),$$

where (i) uses Eq. (32). Then Eq. (33) can be proved by substituting the above equality into Eq. (32). $\qquad\square$

**Lemma 3.** *Under Assumption 1, the occupancy measure* (1) *satisfies the following Lipschitz properties for any $\theta, \theta' \in \Theta$ and $\xi, \xi' \in \Xi$.*

$$\|\lambda_{\theta',\xi'} - \lambda_{\theta,\xi}\| \leq \frac{\gamma\|p_{\xi'} - p_\xi\| + \|\pi_{\theta'} - \pi_\theta\|}{1 - \gamma} \leq \frac{\gamma\ell_{p_\xi}\sqrt{|\mathcal{S}|}\|\xi' - \xi\| + \ell_{\pi_\theta}\sqrt{|\mathcal{A}|}\|\theta' - \theta\|}{1 - \gamma}. \tag{34}$$

*Proof.* For any $\theta, \theta' \in \Theta$ and $\xi, \xi' \in \Xi$, we have

$$\|\lambda_{\theta',\xi'} - \lambda_{\theta,\xi}\|$$

$$= \sqrt{\sum_{s',a'} |\lambda_{\theta',\xi'}(s', a') - \lambda_{\theta,\xi}(s', a')|^2}$$

$$\overset{(i)}{=} \Bigg[ \sum_{s',a'} \Bigg| \gamma\pi_{\theta'}(a'|s') \sum_{s,a} \big[\lambda_{\theta',\xi'}(s, a)p_{\xi'}(s'|s, a) - \lambda_{\theta,\xi}(s, a)p_\xi(s'|s, a)\big]$$

$$+ \Big((1 - \gamma)\rho(s') + \gamma \sum_{s,a} \lambda_{\theta,\xi}(s, a)p_\xi(s'|s, a)\Big)\big[\pi_{\theta'}(a'|s') - \pi_\theta(a'|s')\big] \Bigg|^2 \Bigg]^{1/2}$$

$$\overset{(ii)}{\leq} \sqrt{\sum_{s',a'} \Big| \gamma\pi_{\theta'}(a'|s') \sum_{s,a} \lambda_{\theta',\xi'}(s, a)[p_{\xi'}(s'|s, a) - p_\xi(s'|s, a)] \Big|^2}$$

$$+ \sqrt{\sum_{s',a'} \Big| \gamma\pi_{\theta'}(a'|s') \sum_{s,a} p_\xi(s'|s, a)[\lambda_{\theta',\xi'}(s, a) - \lambda_{\theta,\xi}(s, a)] \Big|^2}$$

$$\sqrt{\sum_{s',a'} \Big| \Big((1 - \gamma)\rho(s') + \gamma \sum_{s,a} \lambda_{\theta,\xi}(s, a)p_\xi(s'|s, a)\Big)\big[\pi_{\theta'}(a'|s') - \pi_\theta(a'|s')\big]\Big|^2}$$

$$\overset{(iii)}{\leq} \gamma\sqrt{\sum_{s'} (\ell_{p_\xi}\|\xi' - \xi\|)^2}$$

$$+ \gamma\sqrt{\sum_{s'} \Big| \sum_{s,a} p_\xi(s'|s, a)[\lambda_{\theta',\xi'}(s, a) - \lambda_{\theta,\xi}(s, a)] \Big|^2} + \sqrt{\sum_{a'} (\ell_{\pi_\theta}\|\theta' - \theta\|)^2}$$

$$\overset{(iv)}{\leq} \gamma\ell_{p_\xi}\sqrt{|\mathcal{S}|}\|\xi' - \xi\| + \gamma\sqrt{\sum_{s'} \sum_{s,a} p_\xi(s'|s, a)|\lambda_{\theta',\xi'}(s, a) - \lambda_{\theta,\xi}(s, a)|^2} + \ell_{\pi_\theta}\sqrt{|\mathcal{A}|}\|\theta' - \theta\|$$

$$\leq \gamma\ell_{p_\xi}\sqrt{|\mathcal{S}|}\|\xi' - \xi\| + \gamma\|\lambda_{\theta',\xi'} - \lambda_{\theta,\xi}\| + \ell_{\pi_\theta}\sqrt{|\mathcal{A}|}\|\theta' - \theta\|,$$

where (i) uses Eq. (32), (ii) uses triangular inequality, (iii) uses Lemma 1, $\sum_{a'} \pi_{\theta'}(a'|s')^2 \leq 1$ and $\sum_{s'}[(1 - \gamma)\rho(s') + \gamma \sum_{s,a} \lambda_{\theta,\xi}(s, a)p_\xi(s'|s, a)] = 1$ and (iv) uses Jensen's inequality. Then Eq. (34) can be proved by rearranging the above inequality. $\qquad\square$

**Lemma 4.** *The distance between any pair of probability vectors $x, y \in \Delta^{\mathcal{X}}$ on finite space $\mathcal{X}$ has the upper bound that $\|x - y\| \leq \sqrt{2}$.*

*Proof.* Denote $d = |\mathcal{X}|$ and $x_j, y_j$ as the $j$-th entry of $x, y$ respectively. Then

$$\|x - y\|^2 \leq \sum_{j=1}^d x_j^2 + y_j^2 - 2x_j y_j \leq \sum_{j=1}^d x_j + y_j = 2.$$

$\square$

**Lemma 5.** *Under Assumptions 1-2, the projected gradients in* (13) *have the following properties.*

$$\|G_\beta^{(\theta)}(\theta,\xi)\| \le \|\nabla_\theta f(\lambda_{\theta,\xi})\| \le \ell_\theta, \quad \|G_\beta^{(\xi)}(\theta,\xi)\| \le \|\nabla_\xi f(\lambda_{\theta,\xi})\| \le \ell_\xi, \tag{35}$$

$$\|G_\beta^{(\theta)}(\theta',\xi') - G_\beta^{(\theta)}(\theta,\xi)\| \le \left(\frac{1}{\beta} + L_{\theta,\theta}\right)\|\theta' - \theta\| + L_{\theta,\xi}\|\xi' - \xi\|, \tag{36}$$

$$\|G_\alpha^{(\xi)}(\theta',\xi') - G_\alpha^{(\xi)}(\theta,\xi)\| \le L_{\xi,\theta}\|\theta' - \theta\| + \left(\frac{1}{\alpha} + L_{\xi,\xi}\right)\|\xi' - \xi\|. \tag{37}$$

*Proof.* The proof for $\|G_\beta^{(\theta)}(\theta,\xi)\|$ in Eq. (35) simply follows from the contraction property of projection as follows.

$$\|G_\beta^{(\theta)}(\theta,\xi)\| := \frac{1}{\beta}\|\theta - \text{proj}_\Theta(\theta - \beta\nabla_\theta f(\lambda_{\theta,\xi})\| \le \frac{1}{\beta}\|\theta - (\theta - \beta\nabla_\theta f(\lambda_{\theta,\xi})\| = \|\nabla_\theta f(\lambda_{\theta,\xi})\|.$$

Then, $\|\nabla_\theta f(\lambda_{\theta,\xi})\| \le \ell_\theta$ by Proposition 3. The proof logic for $\|G_\beta^{(\xi)}(\theta,\xi)\|$ is the same.

Next, we prove Eq. (36) as follows and the proof of Eq. (37) follows the same logic.

$$\begin{aligned}\|G_\beta^{(\theta)}(\theta',\xi') - G_\beta^{(\theta)}(\theta,\xi)\| =& \frac{1}{\beta}\|\text{proj}_\Theta[\theta' - \beta\nabla_\theta f(\lambda_{\theta',\xi'})] - \text{proj}_\Theta[\theta - \beta\nabla_\theta f(\lambda_{\theta,\xi})]\| \\ \le& \frac{1}{\beta}\|\theta' - \theta\| + \|\nabla_\theta f(\lambda_{\theta,\xi}) - \nabla_\theta f(\lambda_{\theta',\xi'})\| \\ \overset{(i)}{\le}& \left(\frac{1}{\beta} + L_{\theta,\theta}\right)\|\theta' - \theta\| + L_{\theta,\xi}\|\xi' - \xi\|,\end{aligned} \tag{38}$$

where (i) uses Proposition 3. $\square$

**Lemma 6.** *Suppose* $\mathcal{X} \subset \mathbb{R}^d$ *is a closed convex set. For any* $x \in \mathbb{R}^d$ *and* $x' \in \mathcal{X}$*, we have*

$$[x' - \text{proj}_\mathcal{X}(x)]^\top [x - \text{proj}_\mathcal{X}(x)] \le 0 \tag{39}$$

*Proof.* For any $\delta \in (0,1]$, $x_\delta := \delta x' + (1-\delta)\text{proj}_\mathcal{X}(x)$ belongs to the convex set $\mathcal{X}$ since $x', \text{proj}_\mathcal{X}(x) \in \mathcal{X}$. Then based on the definition of projection we have

$$\begin{aligned}0 \le& \|x_\delta - x\|^2 - \|\text{proj}_\mathcal{X}(x) - x\|^2 \\ \le& \|x_\delta - \text{proj}_\mathcal{X}(x)\|^2 - 2[x_\delta - \text{proj}_\mathcal{X}(x)]^\top [x - \text{proj}_\mathcal{X}(x)] \\ =& \delta^2\|x' - \text{proj}_\mathcal{X}(x)\|^2 - 2\delta[x' - \text{proj}_\mathcal{X}(x)]^\top [x - \text{proj}_\mathcal{X}(x)].\end{aligned}$$

The above inequality can be rearranged as follows

$$[x' - \text{proj}_\mathcal{X}(x)]^\top [x - \text{proj}_\mathcal{X}(x)] \le \frac{\delta}{2}\|x' - \text{proj}_\mathcal{X}(x)\|^2,$$

which proves Eq. (39) as $\delta \to +0$. $\square$

## C Stochastic Gradients

To get the stochastic estimation of the gradients (8) and (9), we first estimate the occupancy measure (1) as follows.

$$\widehat{\lambda}(\tau^{(\lambda)}; s, a) := \frac{1 - \gamma}{m_\lambda} \sum_{i=1}^{m_\lambda} \sum_{h=0}^{H_\lambda - 1} \gamma^h \mathbb{1}\{s_{i,h}^{(\lambda)} = s, a_{i,h}^{(\lambda)} = a\}, \tag{40}$$

where $\mathbb{1}\{\cdot\}$ is an indicator function and $\tau^{(\lambda)} := \{\tau_i^{(\lambda)}\}_{i=1}^{m_\lambda}$ contains $m_\lambda$ independent trajectories $\tau_i^{(\lambda)} := \{s_{i,h}^{(\lambda)}, a_{i,h}^{(\lambda)}\}_{h=0}^{H_\lambda - 1}$ $(i = 1, \ldots, m_\lambda)$ of length $H_\lambda$ generated from the policy $\pi_\theta$ and transition kernel $p_\xi$. Then the estimated cost function is $\widehat{c} := \nabla_\lambda f[\widehat{\lambda}(\tau^{(\lambda)})]$.

---

**Algorithm 3** Obtain Stochastic Gradients at $(\theta, \xi)$

---

1: **Input:** $z := (\theta, \xi) \in \mathcal{Z} := \Theta \times \mathcal{Z}$.
2: **Hyperparameters:** $m_\lambda, H_\lambda, m_\theta, H_\theta, m_\xi, H_\xi$.
3: Generate independent trajectories $\tau_i^{(\lambda)} := \{s_{i,h}^{(\lambda)}, a_{i,h}^{(\lambda)}\}_{h=0}^{H_\lambda - 1}$ $(i = 1, \ldots, m_\lambda)$ from $\pi_\theta, p_\xi$.
4: Obtain $\widehat{\lambda}(\tau^{(\lambda)}; s, a)$ for every $s, a \in \mathcal{S} \times \mathcal{A}$ by Eq. (40) with $\tau^{(\lambda)} := \{\tau_i^{(\lambda)}\}_{i=1}^{m_\lambda}$.
5: Obtain $\widehat{c} := \nabla_\lambda f[\widehat{\lambda}(\tau^{(\lambda)})]$.
6: Generate independent trajectories $\tau_i^{(\theta)} := \{s_{i,h}^{(\theta)}, a_{i,h}^{(\theta)}\}_{h=0}^{H_\theta - 1}$ $(i = 1, \ldots, m_\theta)$ from $\pi_\theta, p_\xi$.
7: Obtain $g^{(\theta)}(\tau^{(\theta)}, \theta, \xi, \widehat{c})$ by Eq. (41) with $\tau^{(\theta)} := \{\tau_i^{(\theta)}\}_{i=1}^{m_\theta}$.
8: Generate independent trajectories $\tau_i^{(\xi)} := \{s_{i,h}^{(\xi)}, a_{i,h}^{(\xi)}\}_{h=0}^{H_\xi - 1}$ $(i = 1, \ldots, m_\xi)$ from $\pi_\theta, p_\xi$.
9: Obtain $g^{(\xi)}(\tau^{(\xi)}, \theta, \xi, \widehat{c})$ by Eq. (42) with $\tau^{(\xi)} := \{\tau_i^{(\xi)}\}_{i=1}^{m_\xi}$.
10: **Output:** $g^{(\theta)}(\tau^{(\theta)}, \theta, \xi, \widehat{c}) \approx \nabla_\theta f(\lambda_{\theta,\xi})$, $g^{(\xi)}(\tau^{(\xi)}, \theta, \xi, \widehat{c}) \approx \nabla_\xi f(\lambda_{\theta,\xi})$.

---

The stochastic gradients (8) and (9) can be approximated respectively by the following stochastic sample averaged values known as GPOMDP [50].

$$g^{(\theta)}(\tau^{(\theta)}, \theta, \xi, \widehat{c}) = \frac{1}{m_\theta} \sum_{i=1}^{m_\theta} \left[ \sum_{t=0}^{H_\theta - 1} \gamma^t \widehat{c}(s_{i,t}^{(\theta)}, a_{i,t}^{(\theta)}) \sum_{h=0}^{t} \nabla_\theta \log \pi_\theta(a_{i,h}^{(\theta)} \mid s_{i,h}^{(\theta)}) \right], \tag{41}$$

$$g^{(\xi)}(\tau^{(\xi)}, \theta, \xi, \widehat{c}) = \frac{1}{m_\lambda} \sum_{i=1}^{m_\lambda} \left[ \sum_{t=0}^{H_\lambda - 1} \gamma^t \widehat{c}(s_{i,t}^{(\xi)}, a_{i,t}^{(\xi)}) \sum_{h=0}^{t} \nabla_\xi \log p_\xi(s_{i,h+1}^{(\xi)} \mid s_{i,h}^{(\xi)}, a_{i,h}^{(\xi)}) \right]. \tag{42}$$

where $\tau^{(\theta)} := \{\tau_i^{(\theta)}\}_{i=1}^{m_\theta}$ and $\tau^{(\xi)} := \{\tau_i^{(\xi)}\}_{i=1}^{m_\xi}$ contain $m_\theta$ independent trajectories $\tau_i^{(\theta)} := \{s_{i,h}^{(\theta)}, a_{i,h}^{(\theta)}\}_{h=0}^{H_\theta - 1}$ $(i = 1, \ldots, m_\theta)$ and $m_\xi$ independent trajectories $\tau_i^{(\xi)} := \{s_{i,h}^{(\xi)}, a_{i,h}^{(\xi)}\}_{h=0}^{H_\xi - 1} \cup \{s_{i,H_\xi}^{(\xi)}\}$ $(i = 1, \ldots, m_\xi)$ respectively, both generated from the policy $\pi_\theta$ and transition kernel $p_\xi$.

We summarize the procedure of obtaining the stochastic gradients (8) and (9) in Algorithm 3. These stochastic gradients approximate the true gradients with the following error bounds.

**Proposition 9.** *Under Assumptions 1 and 2, the stochastic gradients* (41) *and* (42) *have the following error bounds.*

$$\mathbb{E}_{\pi_\theta, p_\xi} \|g^{(\theta)}(\tau^{(\theta)}, \theta, \xi, \widehat{c}) - \nabla_\theta f(\lambda_{\theta,\xi})\|^2$$
$$\leq \frac{3\ell_{\pi_\theta}^2}{(1-\gamma)^4} \left[ L_\lambda^2 |\mathcal{S}||\mathcal{A}| \left( \frac{1}{m_\lambda} + \gamma^{2H_\lambda} \right) + \frac{\ell_\lambda^2}{m_\theta} + \ell_\lambda^2 [1 + H_\theta(1-\gamma)]^2 \gamma^{2H_\theta} \right], \tag{43}$$

$$\mathbb{E}_{\pi_\theta, p_\xi} \|g^{(\xi)}(\tau^{(\xi)}, \theta, \xi, \widehat{c}) - \nabla_\xi f(\lambda_{\theta,\xi})\|^2$$
$$\leq \frac{3\ell_{p_\xi}^2}{(1-\gamma)^4} \left[ L_\lambda^2 |\mathcal{S}||\mathcal{A}| \left( \frac{1}{m_\lambda} + \gamma^{2H_\lambda} \right) + \frac{\ell_\lambda^2}{m_\xi} + \ell_\lambda^2 [1 + H_\xi(1-\gamma)]^2 \gamma^{2H_\xi} \right]. \tag{44}$$

## D Proof of Proposition 1

As follows, we slightly rewrite the utility function $f$ defined in Eq. (5), by replacing $\lambda$ with $\lambda_{\theta,\xi}$.

$$f(\lambda_{\theta,\xi}) = \begin{cases} \langle c^{(0)}, \lambda_{\theta,\xi} \rangle, & \text{if } \langle c^{(k)}, \lambda_{\theta,\xi} \rangle \leq \tau_k \text{ for all } k = 1, \ldots, K \\ +\infty, & \text{otherwise} \end{cases}.$$

Therefore, for any $\theta \in \Theta$, we have

$$\max_{\xi \in \Xi} f(\lambda_{\theta,\xi}) = \begin{cases} \max_{\xi \in \Xi} \langle c^{(0)}, \lambda_{\theta,\xi} \rangle, & \text{if } \langle c^{(k)}, \lambda_{\theta,\xi} \rangle \leq \tau_k \text{ for all } \xi \in \Xi \text{ and } k = 1, \ldots, K \\ +\infty, & \text{otherwise} \end{cases}.$$

Recalling the definition of the robust value function (3), i.e., $V_\theta^{(k)} \stackrel{\text{def}}{=} \max_{\xi \in \Xi} \langle c^{(k)}, \lambda_{\theta,\xi} \rangle$, the equation above can be rewritten as follows.

$$\max_{\xi \in \Xi} f(\lambda_{\theta,\xi}) = \begin{cases} V_\theta^{(0)}, & \text{if } V_\theta^{(k)} \leq \tau_k \text{ for all } k = 1, \ldots, K \\ +\infty, & \text{otherwise} \end{cases}.$$

Therefore, $\theta \in \Theta$ minimizes $\max_{\xi \in \Xi} f(\lambda_{\theta,\xi})$ if and only if $\theta$ solves the constrained robust RL problem (4), as repeated below.

$$\min_{\theta \in \Theta} V_\theta^{(0)}, \text{ s.t. } V_\theta^{(k)} \le \tau_k \text{ for all } k = 1, \dots, K.$$

Finally, we will prove that $f(\lambda)$ defined in Eq. (5) is a convex function. Note that $A_k = \{\lambda \in \Delta^{\mathcal{S} \times \mathcal{A}} : \langle c^{(k)}, \lambda \rangle \le \tau_k\}$ is a convex set, so $A = \cap_{k=1}^K A_k$ is also a convex set. Then for any $\lambda_1, \lambda_0 \in \Delta^{\mathcal{S} \times \mathcal{A}}$ and $\alpha \in [0, 1]$, we aim to prove that

$$f[\alpha \lambda_1 + (1 - \alpha)\lambda_0] \le \alpha f(\lambda_1) + (1 - \alpha)f(\lambda_0). \tag{45}$$

If either $\lambda_1 \notin A$ or $\lambda_0 \notin A$, then Eq. (45) obviously holds as the right side equals $+\infty$. Otherwise, if $\lambda_1, \lambda_0 \in A$, then $\delta \lambda_1 + (1 - \delta)\lambda_0 \in A$ as $A$ is a convex set, and thus Eq. (45) holds with equality as proved below.

$$\begin{aligned} f[\delta \lambda_1 + (1 - \delta)\lambda_0] &= \langle c^{(0)}, \delta \lambda_1 + (1 - \delta)\lambda_0 \rangle \\ &= \delta \langle c^{(0)}, \lambda_1 \rangle + (1 - \delta)\langle c^{(0)}, \lambda_0 \rangle \\ &= \delta f(\lambda_1) + (1 - \delta)f(\lambda_0). \end{aligned}$$

# E    Proof of Proposition 2

The utility function $f$ in Eq. (7) satisfies

$$\begin{aligned} f(\lambda_{\theta,\xi}) &= \sum_{s,a} \lambda_{\theta,\xi}(s, a) \Big[ c(s, a) + \mu \log \frac{\lambda_{\theta,\xi}(s, a)}{\sum_{a'} \lambda_{\theta,\xi}(s, a')} \Big] \\ &\overset{(i)}{=} \sum_{s,a} \big[ \lambda_{\theta,\xi}(s, a)c(s, a) \big] + \mu \sum_{s,a} \lambda_{\theta,\xi}(s)\pi_\theta(a|s) \log \pi_\theta(a|s) \\ &\overset{(ii)}{=} \sum_{s,a} \big[ \lambda_{\theta,\xi}(s, a)c(s, a) \big] - \mu \sum_s \big[ \lambda_{\theta,\xi}(s)\mathcal{H}[\pi_\theta(\cdot|s)] \big]. \end{aligned}$$

where (i) uses $\lambda_{\theta,\xi}(s) = \sum_a \lambda_{\theta,\xi}(s, a)$ and Eq. (33) that $\lambda_{\theta,\xi}(s, a) = \lambda_{\theta,\xi}(s)\pi_\theta(a|s)$, and (ii) denotes the entropy function that $\mathcal{H}[\pi_\theta(\cdot|s)] = -\sum_a \pi_\theta(a|s) \log \pi_\theta(a|s)$. The above function is exactly the minimax objective function (6) of the robust entropy regularized RL.

Finally, we will prove that $f(\lambda)$ defined in Eq. (7) is a convex function. For any $\lambda_0, \lambda_1 \in \Delta^{\mathcal{S} \times \mathcal{A}}$ and $\delta \in [0, 1]$, denote $\lambda_\delta = \delta \lambda_1 + (1 - \delta)\lambda_0$, $\lambda_\delta(s) = \sum_a \lambda_\delta(s, a)$ and policy $\pi_\delta(a|s) = \frac{\lambda_\delta(s,a)}{\lambda_\delta(s)}$. Then, the convexity of $f$ can be proved as follows.

$$\begin{aligned} &\delta f(\lambda_1) + (1 - \delta)f(\lambda_0) - f(\lambda_\delta) \\ =&\mu \sum_{s,a} \Big[ \delta \lambda_1(s, a) \log \frac{\lambda_1(s,a)}{\lambda_1(s)} + (1 - \delta)\lambda_0(s, a) \log \frac{\lambda_0(s,a)}{\lambda_0(s)} - \lambda_\delta(s, a) \log \frac{\lambda_\delta(s,a)}{\lambda_\delta(s)} \Big] \\ =&\mu \sum_{s,a} \Big[ \delta \lambda_1(s, a) \log \pi_1(a|s) + (1 - \delta)\lambda_0(s, a) \log \pi_0(a|s) \\ &\quad - [\delta \lambda_1(s, a) + (1 - \delta)\lambda_0(s, a)] \log \pi_\delta(a|s) \Big] \\ =&\mu \sum_{s,a} \Big[ \delta \lambda_1(s)\pi_1(a|s) \log \frac{\pi_1(a|s)}{\pi_\delta(a|s)} + (1 - \delta)\lambda_0(s)\pi_0(a|s) \log \frac{\pi_0(a|s)}{\pi_\delta(a|s)} \Big] \\ =&\mu \sum_s \Big[ \delta \lambda_1(s)\text{KL}[\pi_1(\cdot|s)\|\pi_\delta(\cdot|s)] + (1 - \delta)\lambda_0(s)\text{KL}[\pi_0(\cdot|s)\|\pi_\delta(\cdot|s)] \Big] \ge 0. \end{aligned}$$

# F    Proof of Proposition 3

The first formula of Eq. (10) can be proved as follows and the second formula can be proved in the same way.

$$\|\nabla_\theta f(\lambda_{\theta,\xi})\| \overset{(i)}{\le} \mathbb{E}_{\pi_\theta, p_\xi} \Big[ \sum_{t=0}^{+\infty} \gamma^t |c(s_t, a_t)| \sum_{h=0}^t \|\nabla_\theta \log \pi_\theta(a_h|s_h)\| \Big]$$

$$\overset{(ii)}{\leq} \mathbb{E}_{\pi_\theta, p_\xi} \left[ \sum_{t=0}^{+\infty} \gamma^t \sum_{h=0}^{t} \ell_{\pi_\theta} \right]$$

$$= \ell_{\pi_\theta} \sum_{t=0}^{+\infty} \gamma^t (t+1) = \frac{\ell_{\pi_\theta}}{(1-\gamma)^2},$$

where (i) uses Eq. (41) and (ii) uses $c(s_t, a_t) \in [0,1]$ and Assumption 1.

Define the following V function.

$$V_{\theta,\xi}(c) := \mathbb{E}_{\pi_\theta, p_\xi} \left[ \sum_{t=0}^{\infty} \gamma^t c(s_t, a_t) \Big| s_0 \sim \rho \right]. \tag{46}$$

For any fixed cost function $c : \mathcal{S} \times \mathcal{A} \to \mathbb{R}$, the gradient $\nabla_\theta V_{\theta,\xi}(c)$ can be rewritten as follows.

$$\nabla_\theta V_{\theta,\xi}(c) \overset{(i)}{=} \mathbb{E}_{\pi_\theta, p_\xi} \left[ \sum_{t=0}^{+\infty} \gamma^t c(s_t, a_t) \sum_{h=0}^{t} \nabla_\theta \log \pi_\theta(a_h | s_h) \right]$$

$$= \mathbb{E}_{\pi_\theta, p_\xi} \left[ \sum_{h=0}^{+\infty} \gamma^h \nabla_\theta \log \pi_\theta(a_h | s_h) \sum_{t=h}^{+\infty} \gamma^{t-h} c(s_t, a_t) \right]$$

$$= \sum_{h=0}^{+\infty} \gamma^h \sum_{s,a} \mathbb{P}_{\pi_\theta, p_\xi}(s_h = s, a_h = a | s_0 \sim \rho) \nabla_\theta \log \pi_\theta(a|s)$$

$$\mathbb{E}_{\pi_\theta, p_\xi} \left( \sum_{t=h}^{+\infty} \gamma^{t-h} c(s_t, a_t) \Big| s_h = s, a_h = a \right)$$

$$= \sum_{s,a} \sum_{h=0}^{+\infty} \gamma^h \mathbb{P}_{\pi_\theta, p_\xi}(s_h = s, a_h = a | s_0 \sim \rho) \nabla_\theta \log \pi_\theta(a|s)$$

$$\mathbb{E}_{\pi_\theta, p_\xi} \left( \sum_{t=0}^{+\infty} \gamma^t c(s_t, a_t) \Big| s_0 = s, a_0 = a \right)$$

$$\overset{(ii)}{=} \frac{1}{1-\gamma} \sum_{s,a} \lambda_{\theta,\xi}(s,a) \nabla_\theta \log \pi_\theta(a|s) Q_{\theta,\xi}(s,a;c), \tag{47}$$

where (i) uses Eq. (5) of [6] and (ii) uses the occupancy measure (1) and defines the following Q function.

$$Q_{\theta,\xi}(s,a;c) := \mathbb{E}_{\pi_\theta, p_\xi} \left( \sum_{t=0}^{+\infty} \gamma^t c(s_t, a_t) \Big| s_0 = s, a_0 = a \right). \tag{48}$$

The above Q function has the following upper bound

$$|Q_{\theta,\xi}(s,a;c)| \leq \frac{c_{\max}}{1-\gamma}, \tag{49}$$

where $c_{\max} := \max_{s,a} |c(s,a)|$ and also satisfies the following Bellman equation.

$$Q_{\theta,\xi}(s,a;c) = c(s,a) + \gamma \sum_{s',a'} p_\xi(s'|s,a) \pi_\theta(a'|s') Q_{\theta,\xi}(s',a';c). \tag{50}$$

Therefore, for any $\theta, \theta' \in \Theta, \xi, \xi' \in \Xi$ and fixed cost function $c$, we have

$$\max_{s,a} |Q_{\theta',\xi'}(s,a;c) - Q_{\theta,\xi}(s,a;c)|$$

$$\leq \gamma \max_{s,a} \sum_{s',a'} |p_{\xi'}(s'|s,a) - p_\xi(s'|s,a)| \pi_{\theta'}(a'|s') |Q_{\theta',\xi'}(s',a';c)|$$

$$+ \gamma \max_{s,a} \sum_{s',a'} p_\xi(s'|s,a) |\pi_{\theta'}(a'|s') - \pi_\theta(a'|s')| |Q_{\theta',\xi'}(s',a';c)|$$

$$+ \gamma \max_{s,a} \sum_{s',a'} p_\xi(s'|s,a)\pi_\theta(a'|s')|Q_{\theta',\xi'}(s',a';c) - Q_{\theta,\xi}(s',a';c)|$$

$$\overset{(i)}{\leq} \frac{\gamma c_{\max}}{1-\gamma}\Big(\max_{s,a}\|p_{\xi'}(\cdot|s,a) - p_\xi(\cdot|s,a)\|_1 + \max_s \|\pi_{\theta'}(\cdot|s) - \pi_\theta(\cdot|s)\|_1\Big)$$
$$+ \gamma \max_{s,a}|Q_{\theta',\xi'}(s,a;c) - Q_{\theta,\xi}(s,a;c)|,$$

where (i) uses Eq. (49). Rearranging the above inequality yields that

$$\max_{s,a}|Q_{\theta',\xi'}(s,a;c) - Q_{\theta,\xi}(s,a;c)|$$

$$\leq \frac{\gamma c_{\max}}{(1-\gamma)^2}\Big(\max_{s,a}\|p_{\xi'}(\cdot|s,a) - p_\xi(\cdot|s,a)\|_1 + \max_s \|\pi_{\theta'}(\cdot|s) - \pi_\theta(\cdot|s)\|_1\Big)$$

$$\overset{(i)}{\leq} \frac{\gamma c_{\max}}{(1-\gamma)^2}\big(\ell_{p_\xi}|\mathcal{S}|\|\xi' - \xi\| + \ell_{\pi_\theta}|\mathcal{A}|\|\theta' - \theta\|\big). \tag{51}$$

where (i) uses Lemma 1. For any $\theta \in \Theta$, $\xi \in \Xi$ and fixed cost functions $c, c'$, we have

$$\max_{s,a}|Q_{\theta,\xi}(s,a;c) - Q_{\theta,\xi}(s,a;c')| \overset{(i)}{\leq} \mathbb{E}_{\pi_\theta,p_\xi}\left(\sum_{t=0}^{+\infty}\gamma^t|c'(s_t,a_t) - c(s_t,a_t)|\Big|s_0 = s, a_0 = a\right)$$

$$\leq \sum_{t=0}^{+\infty}\gamma^t\|c' - c\|_\infty = \frac{\|c' - c\|_\infty}{1-\gamma}. \tag{52}$$

where (i) uses Eq. (48).

Therefore, Eq. (11) can be proved as follows.

$$\|\nabla_\theta f(\lambda_{\theta',\xi'}) - \nabla_\theta f(\lambda_{\theta,\xi})\|$$
$$= \|\nabla_\theta V_{\theta',\xi'}[\nabla_\lambda f(\lambda_{\theta',\xi'})] - \nabla_\theta V_{\theta,\xi}[\nabla_\lambda f(\lambda_{\theta,\xi})]\|$$

$$\overset{(i)}{\leq} \frac{1}{1-\gamma}\sum_{s,a}\big|\lambda_{\theta',\xi'}(s,a) - \lambda_{\theta,\xi}(s,a)\big|\big\|\nabla_\theta \log \pi_{\theta'}(a|s)\big\|\big|Q_{\theta',\xi'}[s,a;\nabla_\lambda f(\lambda_{\theta',\xi'})]\big|$$

$$+ \frac{1}{1-\gamma}\sum_{s,a}\lambda_{\theta,\xi}(s,a)\big\|\nabla_\theta \log \pi_{\theta'}(a|s) - \nabla_\theta \log \pi_\theta(a|s)\big\|\big|Q_{\theta',\xi'}[s,a;\nabla_\lambda f(\lambda_{\theta',\xi'})]\big|$$

$$+ \frac{1}{1-\gamma}\sum_{s,a}\lambda_{\theta,\xi}(s,a)\big\|\nabla_\theta \log \pi_\theta(a|s)\big\|\big|Q_{\theta',\xi'}[s,a;\nabla_\lambda f(\lambda_{\theta',\xi'})] - Q_{\theta,\xi}[s,a;\nabla_\lambda f(\lambda_{\theta',\xi'})]\big|$$

$$+ \frac{1}{1-\gamma}\sum_{s,a}\lambda_{\theta,\xi}(s,a)\big\|\nabla_\theta \log \pi_\theta(a|s)\big\|\big|Q_{\theta,\xi}[s,a;\nabla_\lambda f(\lambda_{\theta',\xi'})] - Q_{\theta,\xi}[s,a;\nabla_\lambda f(\lambda_{\theta,\xi})]\big|$$

$$\overset{(ii)}{\leq} \frac{\ell_{\pi_\theta}\ell_\lambda}{(1-\gamma)^2}\sum_{s,a}\big|\lambda_{\theta',\xi'}(s,a) - \lambda_{\theta,\xi}(s,a)\big| + \frac{L_{\pi_\theta}\ell_\lambda}{(1-\gamma)^2}\sum_{s,a}\lambda_{\theta,\xi}(s,a)\|\theta' - \theta\|$$

$$+ \frac{\ell_{\pi_\theta}}{1-\gamma}\sum_{s,a}\lambda_{\theta,\xi}(s,a)\cdot\frac{\gamma\ell_\lambda}{(1-\gamma)^2}\big(\ell_{p_\xi}|\mathcal{S}|\|\xi' - \xi\| + \ell_{\pi_\theta}|\mathcal{A}|\|\theta' - \theta\|\big)$$

$$+ \frac{\ell_{\pi_\theta}}{(1-\gamma)^2}\sum_{s,a}\lambda_{\theta,\xi}(s,a)\big\|\nabla_\lambda f(\lambda_{\theta',\xi'}) - \nabla_\lambda f(\lambda_{\theta,\xi})\big\|_\infty$$

$$\overset{(iii)}{\leq} \frac{\ell_{\pi_\theta}\ell_\lambda\sqrt{|\mathcal{S}||\mathcal{A}|}}{(1-\gamma)^2}\big\|\lambda_{\theta',\xi'} - \lambda_{\theta,\xi}\big\| + \frac{L_{\pi_\theta}\ell_\lambda}{(1-\gamma)^2}\|\theta' - \theta\|$$

$$+ \frac{\gamma\ell_\lambda\ell_{\pi_\theta}}{(1-\gamma)^3}\big(\ell_{p_\xi}|\mathcal{S}|\|\xi' - \xi\| + \ell_{\pi_\theta}|\mathcal{A}|\|\theta' - \theta\|\big) + \frac{\ell_{\pi_\theta}L_\lambda}{(1-\gamma)^2}\big\|\lambda_{\theta',\xi'} - \lambda_{\theta,\xi}\big\|$$

$$\overset{(iv)}{\leq} \frac{\ell_{\pi_\theta}(L_\lambda + \ell_\lambda\sqrt{|\mathcal{S}||\mathcal{A}|})}{(1-\gamma)^2}\cdot\frac{\gamma\ell_{p_\xi}\sqrt{|\mathcal{S}|}\|\xi' - \xi\| + \ell_{\pi_\theta}\sqrt{|\mathcal{A}|}\|\theta' - \theta\|}{1-\gamma} + \frac{L_{\pi_\theta}\ell_\lambda}{(1-\gamma)^2}\|\theta' - \theta\|$$

$$+ \frac{\gamma\ell_\lambda\ell_{\pi_\theta}}{(1-\gamma)^3}\big(\ell_{p_\xi}|\mathcal{S}|\|\xi' - \xi\| + \ell_{\pi_\theta}|\mathcal{A}|\|\theta' - \theta\|\big)$$

$$\overset{(v)}{\leq} \frac{\gamma \ell_{\pi_\theta} \ell_{p_\xi} \sqrt{|\mathcal{S}|}}{(1-\gamma)^3}(L_\lambda + 2\ell_\lambda \sqrt{|\mathcal{S}||\mathcal{A}|})\|\xi' - \xi\| + \Big( \frac{\ell_{\pi_\theta}^2 \sqrt{|\mathcal{A}|}(L_\lambda + \ell_\lambda \sqrt{|\mathcal{S}||\mathcal{A}|})}{(1-\gamma)^3} + \frac{L_{\pi_\theta} \ell_\lambda}{(1-\gamma)^2} \Big) \|\theta' - \theta\|,$$

where (i) uses the gradient (47), (ii) uses Assumptions 1-2 and Eqs. (49), (51) and (52) with $c_{\max} = \|\nabla_\lambda f(\lambda_{\theta',\xi'})\|_\infty$ replaced by its upper bound $\ell_\lambda$, (iii) uses Assumption 2, (iv) uses Lemma 3 and (v) uses Assumption 1.

The proof of Eq. (12) follows the same logic. To elaborate, $\nabla_\xi V_{\theta,\xi}(c)$ can be derived as follows in a similar way to the derivation of Eq. (47)

$$\nabla_\xi V_{\theta,\xi}(c) := \mathbb{E}_{\pi_\theta, p_\xi} \Big[ \sum_{t=0}^{+\infty} \gamma^t c(s_t, a_t) \sum_{h=0}^{t} \nabla_\xi \log p_\xi(s_{h+1}|s_h, a_h) \Big]$$

$$= \sum_{h=0}^{+\infty} \gamma^h \sum_{s,a,s'} \mathbb{P}_{\pi_\theta, p_\xi}(s_h = s, a_h = a, s_{h+1} = s'|s_0 \sim \rho) \nabla_\xi \log p_\xi(s'|s, a)$$

$$\mathbb{E}_{\pi_\theta, p_\xi} \Big( \sum_{t=h}^{+\infty} \gamma^{t-h} c(s_t, a_t) \Big| s_h = s, a_h = a, s_{h+1} = s' \Big)$$

$$= \sum_{h=0}^{+\infty} \gamma^h \sum_{s,a,s'} \mathbb{P}_{\pi_\theta, p_\xi}(s_h = s, a_h = a|s_0 \sim \rho) p_\xi(s'|s, a) \nabla_\xi \log p_\xi(s'|s, a)$$

$$\mathbb{E}_{\pi_\theta, p_\xi} \Big( c(s, a) + \sum_{t=1}^{+\infty} \gamma^t c(s_t, a_t) \Big| s_0 = s, a_0 = a, s_1 = s' \Big)$$

$$\overset{(i)}{=} \frac{1}{1-\gamma} \sum_{s,a,s'} \lambda_{\theta,\xi}(s, a) \nabla_\xi p_\xi(s'|s, a)[c(s, a) + \gamma V_{\theta,\xi}(s'; c)], \tag{53}$$

where (i) uses the occupancy measure (1) and defines the following V function.

$$V_{\theta,\xi}(s'; c) := \mathbb{E}_{\pi_\theta, p_\xi} \Big[ \sum_{t=0}^{\infty} \gamma^t c(s_t, a_t) \Big| s_0 = s' \Big]. \tag{54}$$

The above V function has the following upper bound

$$|V_{\theta,\xi}(s, a; c)| \leq \frac{c_{\max}}{1-\gamma}, \tag{55}$$

where $c_{\max} := \max_{s,a} |c(s, a)|$ and also satisfies the following Bellman equation.

$$V_{\theta,\xi}(s; c) = \sum_a \pi_\theta(a|s) \Big[ c(s, a) + \gamma \sum_{s'} p_\xi(s'|s, a) V_{\theta,\xi}(s'; c) \Big]. \tag{56}$$

As a result,

$$\max_s |V_{\theta',\xi'}(s; c) - V_{\theta,\xi}(s; c)|$$

$$\leq \max_s \sum_a |\pi_{\theta'}(a|s) - \pi_\theta(a|s)| \Big[ |c(s, a)| + \gamma \sum_{s'} p_{\xi'}(s'|s, a) |V_{\theta',\xi'}(s'; c)| \Big]$$

$$+ \gamma \max_s \sum_a \pi_\theta(a|s) \sum_{s'} |p_{\xi'}(s'|s, a) - p_\xi(s'|s, a)| |V_{\theta',\xi'}(s'; c)|$$

$$+ \gamma \max_s \sum_a \pi_\theta(a|s) \sum_{s'} p_\xi(s'|s, a) |V_{\theta',\xi'}(s'; c) - V_{\theta,\xi}(s'; c)| \Big]$$

$$\overset{(i)}{\leq} \frac{c_{\max}}{1-\gamma} \big( \ell_{\pi_\theta} |\mathcal{A}| \|\theta' - \theta\| + \gamma \ell_{p_\xi} |\mathcal{S}| \|\xi' - \xi\| \big) + \gamma \max_s |V_{\theta',\xi'}(s; c) - V_{\theta,\xi}(s; c)|,$$

where (i) uses Eq. (55) and Lemma 1. Rearranging the above inequality yields that

$$\max_s |V_{\theta',\xi'}(s; c) - V_{\theta,\xi}(s; c)| \leq \frac{c_{\max}}{(1-\gamma)^2} \big( \ell_{\pi_\theta} |\mathcal{A}| \|\theta' - \theta\| + \gamma \ell_{p_\xi} |\mathcal{S}| \|\xi' - \xi\| \big). \tag{57}$$

Similar to Eq. (52), we have

$$\max_s |V_{\theta,\xi}(s; c') - V_{\theta,\xi}(s; c)| \le \frac{\|c' - c\|_\infty}{1 - \gamma}. \tag{58}$$

Therefore, we can prove Eq. (12) as follows.

$$
\begin{aligned}
&\|\nabla_\xi f(\lambda_{\theta',\xi'}) - \nabla_\xi f(\lambda_{\theta,\xi})\| \\
=& \|\nabla_\xi V_{\theta',\xi'}[\nabla_\lambda f(\lambda_{\theta',\xi'})] - \nabla_\xi V_{\theta,\xi}[\nabla_\lambda f(\lambda_{\theta,\xi})]\| \\
\overset{(i)}{\le} & \frac{1}{1-\gamma} \sum_{s,a,s'} |\lambda_{\theta',\xi'}(s,a) - \lambda_{\theta,\xi}(s,a)| \|\nabla_\xi p_{\xi'}(s'|s,a)\| \\
& |\nabla_\lambda f(\lambda_{\theta',\xi'})(s,a) + \gamma V_{\theta',\xi'}[s'; \nabla_\lambda f(\lambda_{\theta',\xi'})]| \\
& + \frac{1}{1-\gamma} \sum_{s,a,s'} \lambda_{\theta,\xi}(s,a) \|\nabla_\xi p_{\xi'}(s'|s,a) - \nabla_\xi p_\xi(s'|s,a)\| \\
& |\nabla_\lambda f(\lambda_{\theta',\xi'})(s,a) + \gamma V_{\theta',\xi'}[s'; \nabla_\lambda f(\lambda_{\theta',\xi'})]| \\
& + \frac{\gamma}{1-\gamma} \sum_{s,a,s'} \lambda_{\theta,\xi}(s,a) \|\nabla_\xi p_\xi(s'|s,a)\| |V_{\theta',\xi'}[s; \nabla_\lambda f(\lambda_{\theta',\xi'})] - V_{\theta,\xi}[s; \nabla_\lambda f(\lambda_{\theta',\xi'})]| \\
& + \frac{1}{1-\gamma} \sum_{s,a,s'} \lambda_{\theta,\xi}(s,a) \|\nabla_\xi p_\xi(s'|s,a)\| \\
& (\gamma |V_{\theta,\xi}[s; \nabla_\lambda f(\lambda_{\theta',\xi'})] - V_{\theta,\xi}[s; \nabla_\lambda f(\lambda_{\theta,\xi})]| + |\nabla_\lambda f(\lambda_{\theta',\xi'})(s,a) - \nabla_\lambda f(\lambda_{\theta,\xi})(s,a)|) \\
\overset{(ii)}{\le} & \frac{\ell_{p_\xi}}{1-\gamma} \left(\ell_\lambda + \frac{\gamma \ell_\lambda}{1-\gamma}\right) \sum_{s,a,s'} p_{\xi'}(s'|s,a) |\lambda_{\theta',\xi'}(s,a) - \lambda_{\theta,\xi}(s,a)| \\
& + \frac{1}{1-\gamma} \left(\ell_\lambda + \frac{\gamma \ell_\lambda}{1-\gamma}\right) \sum_{s,a,s'} \lambda_{\theta,\xi}(s,a) [L_{p_\xi} p_{\xi'}(s'|s,a) + \ell_{p_\xi}^2] \|\xi' - \xi\| \\
& + \frac{\gamma \ell_{p_\xi}}{1-\gamma} \sum_{s,a,s'} \lambda_{\theta,\xi}(s,a) p_\xi(s'|s,a) \cdot \frac{\ell_\lambda}{(1-\gamma)^2} (\ell_{\pi_\theta} |\mathcal{A}| \|\theta' - \theta\| + \gamma \ell_{p_\xi} |\mathcal{S}| \|\xi' - \xi\|) \\
& + \frac{\ell_{p_\xi}}{1-\gamma} \sum_{s,a,s'} \lambda_{\theta,\xi}(s,a) p_{\xi'}(s'|s,a) \left[\frac{\gamma \|\nabla_\lambda f(\lambda_{\theta',\xi'}) - \nabla_\lambda f(\lambda_{\theta,\xi})\|_\infty}{1-\gamma} + \|\nabla_\lambda f(\lambda_{\theta',\xi'}) - \nabla_\lambda f(\lambda_{\theta,\xi})\|_\infty\right] \\
\overset{(iii)}{\le} & \frac{\ell_{p_\xi} \ell_\lambda \sqrt{|\mathcal{S}||\mathcal{A}|}}{(1-\gamma)^2} \|\lambda_{\theta',\xi'} - \lambda_{\theta,\xi}\| + \frac{\ell_\lambda (L_{p_\xi} + \ell_{p_\xi}^2 |\mathcal{S}|)}{(1-\gamma)^2} \|\xi' - \xi\| \\
& + \frac{\gamma \ell_\lambda \ell_{p_\xi}}{(1-\gamma)^3} (\ell_{\pi_\theta} |\mathcal{A}| \|\theta' - \theta\| + \gamma \ell_{p_\xi} |\mathcal{S}| \|\xi' - \xi\|) + \frac{\ell_{p_\xi} L_\lambda}{(1-\gamma)^2} \|\lambda_{\theta',\xi'} - \lambda_{\theta,\xi}\| \\
\overset{(iv)}{\le} & \frac{\ell_{p_\xi} (L_\lambda + \ell_\lambda \sqrt{|\mathcal{S}||\mathcal{A}|})}{(1-\gamma)^2} \frac{\gamma \ell_{p_\xi} \sqrt{|\mathcal{S}|} \|\xi' - \xi\| + \ell_{\pi_\theta} \sqrt{|\mathcal{A}|} \|\theta' - \theta\|}{1-\gamma} + \frac{\ell_\lambda (L_{p_\xi} + \ell_{p_\xi}^2 |\mathcal{S}|)}{(1-\gamma)^2} \|\xi' - \xi\| \\
& + \frac{\gamma \ell_\lambda \ell_{p_\xi}}{(1-\gamma)^3} (\ell_{\pi_\theta} |\mathcal{A}| \|\theta' - \theta\| + \gamma \ell_{p_\xi} |\mathcal{S}| \|\xi' - \xi\|) \\
\le & \frac{\ell_{\pi_\theta} \ell_{p_\xi} \sqrt{|\mathcal{A}|} (L_\lambda + \ell_\lambda \sqrt{|\mathcal{S}||\mathcal{A}|})}{(1-\gamma)^3} \|\theta' - \theta\| \\
& + \left(\frac{\gamma \ell_{p_\xi}^2 \sqrt{|\mathcal{S}|} (L_\lambda + 2\ell_\lambda \sqrt{|\mathcal{S}||\mathcal{A}|})}{(1-\gamma)^3} + \frac{\ell_\lambda (L_{p_\xi} + \ell_{p_\xi}^2 |\mathcal{S}|)}{(1-\gamma)^2}\right) \|\xi' - \xi\|,
\end{aligned}
$$

where (i) uses the gradient (53) and denotes $\nabla_\lambda f(\lambda_{\theta',\xi'})(s,a) = \frac{\partial f(\lambda_{\theta',\xi'})}{\partial \lambda(s,a)}$ as the $(s,a)$-th element of $\nabla_\lambda f(\lambda_{\theta',\xi'})$, (ii) uses $\nabla_\xi p_{\xi'}(s'|s,a) = p_{\xi'}(s'|s,a) \nabla_\xi \log p_{\xi'}(s'|s,a)$, Assumptions 1-2 and Eqs. (31), (55), (57) and (58) with $c_{\max} = \|\nabla_\lambda f(\lambda_{\theta',\xi'})\|_\infty$ replaced by its upper bound $\ell_\lambda$, (iii) uses Assumptions 2 and (iv) uses Eq. (34).

# G    Proof of Proposition 4

Implement one projected gradient step from $\theta$ and obtain $\theta' = \text{proj}_{\Theta}\big(\theta - \beta\nabla_\theta f(\lambda_{\theta,\xi})\big) = \theta - \beta G_\beta^{(\theta)}(\theta,\xi)$ where the projected gradient $G_\beta^{(\theta)}(\theta,\xi)$ is defined by Eq. (13). Based on Assumption 4, for any $\delta \in [0,\overline{\delta}]$, there exists $\theta_\delta \in \Theta$ such that $\lambda_{\theta_\delta,\xi} = (1-\delta)\lambda_{\theta',\xi} + \delta\lambda_{\theta^*(\xi),\xi} \in \mathcal{V}_{\lambda_{\theta,\xi}}$. Based on Assumption 4, for any $\delta \in [0,\overline{\delta}]$, there exists $\theta_\delta \in \Theta$ such that $\lambda_{\theta_\delta,\xi} = (1-\delta)\lambda_{\theta',\xi} + \delta\lambda_{\theta^*(\xi),\xi} \in \mathcal{V}_{\lambda_{\theta',\xi}}$. Then we have,

$$\|\theta_\delta - \theta'\| \overset{(i)}{\leq} \ell_{\lambda^{-1}}\big\|\lambda_{\theta_\delta,\xi} - \lambda_{\theta,\xi}\big\| = \delta\ell_{\lambda^{-1}}\big\|\lambda_{\theta^*(\xi),\xi} - \lambda_{\theta,\xi}\big\| \overset{(ii)}{\leq} \sqrt{2}\delta\ell_{\lambda^{-1}}, \tag{59}$$

where (i) uses the $L_{\theta,\theta}$-smoothness of $f(\lambda_{\cdot,\xi})$ based on Proposition 3, (ii) uses Lemma 4.

By applying Lemma 6 to $\mathcal{X} = \Theta$, $x = \theta - \beta\nabla_\theta f(\lambda_{\theta,\xi})$, $x' = \theta_\delta \in \Theta$ (so $\text{proj}_{\mathcal{X}}(x) = \theta' = \theta - \beta G_\beta^{(\theta)}(\theta,\xi)$), we obtain that

$$(\theta_\delta - \theta')^\top[G_\beta^{(\theta)}(\theta,\xi) - \nabla_\theta f(\lambda_{\theta,\xi})] \leq 0. \tag{60}$$

Then on one hand, $f(\lambda_{\theta_\delta,\xi})$ has the following lower bound.

$$\begin{aligned}
f(\lambda_{\theta_\delta,\xi}) &\overset{(i)}{\geq} f(\lambda_{\theta',\xi}) + \nabla_\theta f(\lambda_{\theta',\xi})^\top(\theta_\delta - \theta') - \frac{L_{\theta,\theta}}{2}\|\theta_\delta - \theta'\|^2 \\
&\overset{(ii)}{\geq} f(\lambda_{\theta',\xi}) + \big[\nabla_\theta f(\lambda_{\theta',\xi}) - \nabla_\theta f(\lambda_{\theta,\xi}) + G_\beta^{(\theta)}(\theta,\xi)\big]^\top(\theta_\delta - \theta') - \frac{L_{\theta,\theta}}{2}\|\theta_\delta - \theta'\|^2 \\
&\overset{(iii)}{\geq} f(\lambda_{\theta',\xi}) - \big(L_{\theta,\theta}\|\theta' - \theta\| + \|G_\beta^{(\theta)}(\theta,\xi)\|\big)\|\theta_\delta - \theta'\| - \frac{L_{\theta,\theta}}{2}\|\theta_\delta - \theta'\|^2 \\
&\overset{(iv)}{\geq} f(\lambda_{\theta',\xi}) - \sqrt{2}\delta\ell_{\lambda^{-1}}\big(\beta L_{\theta,\theta} + 1\big)\|G_\beta^{(\theta)}(\theta,\xi)\| - L_{\theta,\theta}\delta^2\ell_{\lambda^{-1}}^2,
\end{aligned} \tag{61}$$

where (i) and (iii) use $L_{\theta,\theta}$-smoothness of $f(\lambda_{\cdot,\xi})$ based on Proposition 3, (ii) uses Eq. (60), and (iv) uses Lemma 5, $\|\theta_\delta - \theta'\| \leq \sqrt{2}\delta\ell_{\lambda^{-1}}$ (obtained in the same way as Eq. (59)) and $\theta' - \theta = -\beta G_\beta^{(\theta)}(\theta,\xi)$. On the other hand, $f(\lambda_{\theta_\delta,\xi})$ has the following upper bound since $f$ is convex.

$$f(\lambda_{\theta_\delta,\xi}) \leq (1-\delta)f(\lambda_{\theta',\xi}) + \delta f(\lambda_{\theta^*(\xi),\xi}) = (1-\delta)f(\lambda_{\theta',\xi}) + \delta \min_{\theta'' \in \Theta} f(\lambda_{\theta'',\xi}). \tag{62}$$

The above two inequalities (61) and (62) imply that

$$\begin{aligned}
f(\lambda_{\theta',\xi}) - \min_{\theta'' \in \Theta} f(\lambda_{\theta'',\xi}) &\leq \limsup_{\delta \to +0} \frac{1}{\delta}\big[f(\lambda_{\theta',\xi}) - f(\lambda_{\theta_\delta,\xi})\big] \\
&\leq \sqrt{2}\ell_{\lambda^{-1}}\big(\beta L_{\theta,\theta} + 1\big)\|G_\beta^{(\theta)}(\theta,\xi)\|.
\end{aligned} \tag{63}$$

Finally, we prove Eq. (20) as follows.

$$\begin{aligned}
f(\lambda_{\theta,\xi}) &\overset{(i)}{\leq} f(\lambda_{\theta',\xi}) + \ell_\theta\|\theta' - \theta\| \\
&\overset{(ii)}{\leq} \min_{\theta'' \in \Theta} f(\lambda_{\theta'',\xi}) + \big[\sqrt{2}\ell_{\lambda^{-1}}\big(\beta L_{\theta,\theta} + 1\big) + \beta\ell_\theta\big]\|G_\beta^{(\theta)}(\theta,\xi)\|.
\end{aligned}$$

where (i) uses Eq. (10) which implies that $f_{\lambda_{\cdot,\xi}}$ is $\ell_\theta$-Lipschitz, (ii) uses $\theta' - \theta = -\beta G_\beta^{(\theta)}(\theta,\xi)$ and Eq. (63).

# H    Proof of Proposition 5

We will first prove that $\xi_\delta \in \Xi$. Note that the $s$-rectangular ambiguity set $\Xi$ can be expressed as a Cartesian product of $\Xi_s$ for all $s \in \mathcal{S}$. Hence, as $\Xi$ is convex, $\Xi_s$ is convex for all $s \in \mathcal{S}$. Therefore, for any $s \in \mathcal{S}$, $\xi_\delta(s,\cdot,\cdot) \in \Xi_s$ since it is a convex combination of $\xi_0(s,\cdot,\cdot) \in \Xi_s$ and $\xi_1(s,\cdot,\cdot) \in \Xi_s$ defined by Eq. (21), so $\xi_\delta \in \Xi$.

Next, we will prove $\lambda_{\theta,\xi_\delta} = \delta\lambda_{\theta,\xi_1} + (1-\delta)\lambda_{\theta,\xi_0}$. Denote $\lambda_\delta := \delta\lambda_{\theta,\xi_1} + (1-\delta)\lambda_{\theta,\xi_0}$, so the aim becomes to prove $\lambda_{\theta,\xi_\delta} = \lambda_\delta$. Based on Lemma 2, it suffices to prove the following equation.

$$\lambda_\delta(s',a') = \Big[(1-\gamma)\rho(s') + \gamma\sum_{s,a}\lambda_\delta(s,a)\xi_\delta(s'|s,a)\Big]\pi_\theta(a'|s'), \quad s' \in \mathcal{S}, a' \in \mathcal{A}. \quad (64)$$

For each $s \in \mathcal{S}$, consider the following two cases.

**(Case 1):** $\lambda_{\theta,\xi_1}(s) > 0$ or $\lambda_{\theta,\xi_0}(s) > 0$.
Note that

$$\lambda_\delta(s,a)\xi_\delta(s,a,s')$$

$$\overset{(i)}{=}\big[\delta\lambda_{\theta,\xi_1}(s,a) + (1-\delta)\lambda_{\theta,\xi_0}(s,a)\big]\frac{\delta\lambda_{\theta,\xi_1}(s)\xi_1(s,a,s') + (1-\delta)\lambda_{\theta,\xi_0}(s)\xi_0(s,a,s')}{\delta\lambda_{\theta,\xi_1}(s) + (1-\delta)\lambda_{\theta,\xi_0}(s)}$$

$$\overset{(ii)}{=}\pi_\theta(a|s)\big[\delta\lambda_{\theta,\xi_1}(s) + (1-\delta)\lambda_{\theta,\xi_0}(s)\big]\frac{\delta\lambda_{\theta,\xi_1}(s)\xi_1(s,a,s') + (1-\delta)\lambda_{\theta,\xi_0}(s)\xi_0(s,a,s')}{\delta\lambda_{\theta,\xi_1}(s) + (1-\delta)\lambda_{\theta,\xi_0}(s)}$$

$$\overset{(iii)}{=}\delta\lambda_{\theta,\xi_1}(s,a)\xi_1(s,a,s') + (1-\delta)\lambda_{\theta,\xi_0}(s,a)\xi_0(s,a,s'), \quad (65)$$

where (i) uses Eq. (21) and $\lambda_\delta := \delta\lambda_{\theta,\xi_1} + (1-\delta)\lambda_{\theta,\xi_0}$, (ii) and (iii) use Eq. (33).

**(Case 2):** $\lambda_{\theta,\xi_0}(s) = \lambda_{\theta,\xi_1}(s) = 0$.
In this case, $\lambda_\delta(s) = \delta\lambda_{\theta,\xi_1}(s) + (1-\delta)\lambda_{\theta,\xi_0}(s) = 0$ and thus $\lambda_\delta(s,a) = \lambda_{\theta,\xi_1}(s,a) = \lambda_{\theta,\xi_0}(s,a) = 0$ for any $a \in \mathcal{A}$, so Eq. (65) also holds for any choice of $\xi_\delta(s,\cdot,\cdot)$.

Therefore, we can prove Eq. (64) as follows.

$$\Big[(1-\gamma)\rho(s') + \gamma\sum_{s,a}\lambda_\delta(s,a)\xi_\delta(s'|s,a)\Big]\pi_\theta(a'|s')$$

$$\overset{(i)}{=}\Big[(1-\gamma)\rho(s') + \gamma\sum_{s,a}[\delta\lambda_{\theta,\xi_1}(s,a)\xi_1(s,a,s') + (1-\delta)\lambda_{\theta,\xi_0}(s,a)\xi_0(s,a,s')]\Big]\pi_\theta(a'|s')$$

$$=\delta\Big[(1-\gamma)\rho(s') + \gamma\sum_{s,a}\lambda_{\theta,\xi_1}(s,a)\xi_1(s,a,s')\Big]\pi_\theta(a'|s')$$

$$+ (1-\delta)\Big[(1-\gamma)\rho(s') + \gamma\sum_{s,a}\lambda_{\theta,\xi_0}(s,a)\xi_0(s,a,s')\Big]\pi_\theta(a'|s')$$

$$\overset{(ii)}{=}\delta\lambda_{\theta,\xi_1}(s,a) + (1-\delta)\lambda_{\theta,\xi_0}(s,a) = \lambda_\delta(s,a),$$

where (i) uses Eq. (65) and $\lambda_\delta := \delta\lambda_{\theta,\xi_1} + (1-\delta)\lambda_{\theta,\xi_0}$ and (ii) applies Lemma 2 to $\lambda_{\theta,\xi_1}$ and $\lambda_{\theta,\xi_0}$.

# I   Proof of Proposition 6

Fix any $\theta \in \Theta$, and there exists at least one $\xi^* \in \arg\max_{\xi'\in\Xi}f(\lambda_{\theta,\xi'})$. If $\xi^* \in V(\Xi) := \times_{s\in\mathcal{S}}V(\Xi_s)$, then this proposition directly holds. Hence, we focus on the case where $\xi^* \notin V(\Xi)$, which means $\xi^*(s_0) \notin V(\Xi_{s_0})$ for at least one $s_0 \in \mathcal{S}$.

Based on Assumption 6-7, there exists a probability vector $\nu := [\nu_1,\dots,\nu_{M_{s_0}}]$ such that $\xi^*(s_0) = \sum_{m=1}^{M_{s_0}}\nu_m\xi_m^{(s_0)}$, where we denote $\xi'(s) := \xi'(s,\cdot,\cdot) \in \Xi_s$ for all $\xi' \in \Xi$. Without loss of generality, we assume $\nu_1 = \max_{1\leq m\leq M_s}\nu_m$ (Otherwise we can make this assumption hold by permutating the elements in each $\Xi_s$.).

For any $\epsilon > 0$, define $\xi_1^*, \xi^{(\epsilon)} \in (\Delta^{\mathcal{S}})^{\mathcal{S}\times\mathcal{A}}$ such that $\xi_1^*(s) = \xi^{(\epsilon)}(s) = \xi^*(s)$ for any $s \neq s_0$, while at $s = s_0$ we define $\xi_1^*(s_0) = \xi_1^{(s_0)} \in V(\Xi_{s_0})$ and

$$\xi^{(\epsilon)}(s_0) =\xi^*(s_0) + \epsilon[\xi^*(s_0) - \xi_1^*(s_0)] = [(1+\epsilon)\nu_1 - \epsilon]\xi_1^{(s_0)} + \sum_{m=2}^{M_{s_0}}(1+\epsilon)\nu_m\xi_m^{(s_0)},$$

which implies that

$$\xi^* = \frac{\epsilon\xi^{(\epsilon)} + \xi_1^*}{1+\epsilon}. \quad (66)$$

It is easily seen that $\xi_1^* \in \Xi$ by its definition. Since $\lim_{\epsilon \to +0}[(1+\epsilon)\nu_1(s) - \epsilon] = \nu_1(s) = \max_{1 \le m \le M_s} \nu_m(s) > 0$, there exists a sufficiently small $\epsilon > 0$ such that $[(1+\epsilon)\nu_1(s) - \epsilon, (1+\epsilon)\nu_2(s), \ldots, (1+\epsilon)\nu_{M_s}(s)] \in [0,1]^{|\Xi_s|}$ is a probability vector and thus $\xi^{(\epsilon)} \in \Xi$. Furthermore, select arbitrary $\delta \in [0,1]$ if $\lambda_{\theta,\xi^{(\epsilon)}}(s_0) = \lambda_{\theta,\xi_1^*}(s_0) = 0$ and the following $\delta$ otherwise.

$$\delta = \frac{\lambda_{\theta,\xi^{(\epsilon)}}(s_0)}{\epsilon\lambda_{\theta,\xi_1^*}(s_0) + \lambda_{\theta,\xi^{(\epsilon)}}(s_0)}, \tag{67}$$

where $\lambda_{\theta,\xi}(s) := \sum_{a \in \mathcal{A}} \lambda_{\theta,\xi}(s,a)$ is defined as the state occupancy measure for any $s \in \mathcal{S}$, $\theta \in \Theta$ and $\xi \in \Xi$. Then it can be directly verified that $\xi^*$ satisfies the following equality.

$$\xi^*(s,a,s') = \begin{cases} \text{arbitrary as long as } \xi^*(s,a,\cdot) \in \Delta^{\mathcal{S}}, & \text{if } \lambda_{\theta,\xi^{(\epsilon)}}(s) = \lambda_{\theta,\xi_1^*}(s) = 0 \\ \dfrac{\delta\lambda_{\theta,\xi_1^*}(s)\xi_1^*(s,a,s') + (1-\delta)\lambda_{\theta,\xi^{(\epsilon)}}(s)\xi^{(\epsilon)}(s,a,s')}{\delta\lambda_{\theta,\xi_1^*}(s) + (1-\delta)\lambda_{\theta,\xi^{(\epsilon)}}(s)}, \text{otherwise} \end{cases}.$$

Hence, based on Proposition 5, $\xi^*$ satisfies

$$\lambda_{\theta,\xi^*} = \delta\lambda_{\theta,\xi_1^*} + (1-\delta)\lambda_{\theta,\xi^{(\epsilon)}}. \tag{68}$$

On one hand, $f(\lambda_{\theta,\xi_1^*}) \le f(\lambda_{\theta,\xi^*})$ and $f(\lambda_{\theta,\xi^{(\epsilon)}}) \le f(\lambda_{\theta,\xi^*})$ since $\xi^*$ $\xi^* \in \arg\max_{\xi' \in \Xi} f(\lambda_{\theta,\xi'})$. On the other hand, the above Eq. (68) along with convexity of $f$ implies that

$$f(\lambda_{\theta,\xi^*}) \le \delta f(\lambda_{\theta,\xi_1^*}) + (1-\delta)f(\lambda_{\theta,\xi^{(\epsilon)}}).$$

Therefore, $f(\lambda_{\theta,\xi_1^*}) = f(\lambda_{\theta,\xi^{(\epsilon)}}) = f(\lambda_{\theta,\xi^*}) = \max_{\xi' \in \Xi} f(\lambda_{\theta,\xi'})$. If $\xi_1^* \in V(\Xi)$, then the proof is done. Otherwise, note that $\xi_0^*(s_0) \notin V(\Xi_{s_0})$ while $\xi_1^*(s_0) \in V(\Xi_{s_0})$, and $\xi_1^*(s) = \xi_0^*(s)$ for any $s \ne s_0$. Hence, in the same way, we can obtain the sequence $\xi_2^*, \xi_3^*, \ldots, \xi_N^*$ that satisfies the following conditions by changing non-vertex into vertex at one state each time until no non-vertex remains (i.e., until the condition 2 below holds):

1. For $1 \le k \le N-1$, $\xi_k^*(s_k) \notin V(\Xi_{s_k})$ while $\xi_{k+1}^*(s_k) \in V(\Xi_{s_k})$, and $\xi_k^*(s) = \xi_{k+1}^*(s)$ for any $s \ne s_k$.
2. $\xi_N^* \in V(\Xi)$.
3. For $1 \le k \le N$, $f(\lambda_{\theta,\xi_k^*}) = \max_{\xi' \in \Xi} f(\lambda_{\theta,\xi'})$.

As a result, we find the optimal vertex $\xi_N^* \in V(\Xi) \cap \arg\max_{\xi' \in \Xi} f(\lambda_{\theta,\xi'})$, which concludes the proof.

## J Proof of Proposition 7

Based on Assumption 8, for any $\theta \in \Theta$ and $\delta \in [0,\bar{\delta}]$, there exists $\theta_\delta \in \Theta$ such that $\lambda_{\theta_\delta,\xi} = (1-\delta)\lambda_{\theta,\xi} + \delta\lambda_{\theta^*,\xi}$.

$$\|\theta_\delta - \theta\| \overset{(i)}{\le} \ell_{\lambda^{-1}}\|\lambda_{\theta_\delta,\xi} - \lambda_{\theta,\xi}\| = \delta\ell_{\lambda^{-1}}\|\lambda_{\theta^*,\xi} - \lambda_{\theta,\xi}\| \overset{(ii)}{\le} \sqrt{2}\delta\ell_{\lambda^{-1}}, \tag{69}$$

where (i) uses the $L_{\theta,\theta}$-smoothness of $f(\lambda_{\cdot,\xi})$ based on Proposition 3, (ii) uses Lemma 4. Hence, on one hand, using $L_{\theta,\theta}$-smoothness of $f(\lambda_{\cdot,\xi})$ based on Proposition 3, $f(\lambda_{\theta_\delta,\xi})$ has the following lower bound.

$$f(\lambda_{\theta_\delta,\xi}) \ge f(\lambda_{\theta,\xi}) + \nabla_\theta f(\lambda_{\theta,\xi})^\top(\theta_\delta - \theta) - \frac{L_{\theta,\theta}}{2}\|\theta_\delta - \theta\|^2, \tag{70}$$

On the other hand, $f(\lambda_{\theta_\delta,\xi})$ has the following upper bound since $f$ is convex.

$$f(\lambda_{\theta_\delta,\xi}) \le (1-\delta)f(\lambda_{\theta,\xi}) + \delta f(\lambda_{\theta^*,\xi}) = (1-\delta)f(\lambda_{\theta,\xi}) + \delta \min_{\theta' \in \Theta} \max_{\xi' \in \Xi} f(\lambda_{\theta',\xi'}). \tag{71}$$

Combining the above two inequalities, we obtain that

$$-\nabla_\theta f(\lambda_{\theta,\xi})^\top \frac{\theta_\delta - \theta}{\|\theta_\delta - \theta\|} \ge \frac{f(\lambda_{\theta,\xi}) - f(\lambda_{\theta_\delta,\xi})}{\|\theta_\delta - \theta\|} - \frac{L_{\theta,\theta}}{2}\|\theta_\delta - \theta\|$$

$$\overset{(i)}{\geq} \frac{\delta[f(\lambda_{\theta,\xi}) - \min_{\theta' \in \Theta} \max_{\xi' \in \Xi} f(\lambda_{\theta',\xi'})]}{\|\theta_\delta - \theta\|} - \frac{L_{\theta,\theta}}{2}\|\theta_\delta - \theta\|$$

$$\overset{(ii)}{\geq} \frac{f(\lambda_{\theta,\xi}) - \min_{\theta' \in \Theta} \max_{\xi' \in \Xi} f(\lambda_{\theta',\xi'})}{\sqrt{2}\ell_{\lambda^{-1}}} - \frac{\sqrt{2}\delta\ell_{\lambda^{-1}}L_{\theta,\theta}}{2} \qquad (72)$$

where (i) uses Eq. (71), and (ii) uses Eq. (69) and assumes $f(\lambda_{\theta,\xi}) \geq \min_{\theta' \in \Theta} \max_{\xi' \in \Xi} f(\lambda_{\theta',\xi'})$ without loss of generality (otherwise, Eq. (24) trivially holds).

Based on the Bolzano–Weierstrass theorem, there exists a sequence $\delta_n \to +0$ such that $\frac{\theta_{\delta_n} - \theta}{\|\theta_{\delta_n} - \theta\|} \to d \in \mathbb{R}^{d_\Theta}$ as $n \to +\infty$. Hence, $\|d\| = 1$ and we can conclude the proof by letting $\delta = \delta_n$ and $n \to +\infty$ in the above inequality.

# K  Proof of Proposition 8

## K.1  Proof of Assumptions 1, 2 and 3

Proposition 2 indicates that the utility function $f$ defined by Eq. (7) is convex, which proves Assumption 3.

To prove Assumptions 1 and 2, it suffices to prove that the following functions have bounded first-order and second-order derivatives for any $(s, a, s') \in \mathcal{S} \times \mathcal{A} \times \mathcal{S}$.

1. $\log \pi_\theta(a|s) = \theta_{s,a} - \log \sum_{a'} \exp(\theta_{s,a'})$ as a function of $\theta \in \Theta = [-R, R]^{|\mathcal{S}| \times |\mathcal{A}|}$.

2. $\log p_\xi(s'|s, a) = \log \xi(s, a, s')$ as a function of $\xi \in \Xi = \{\xi' \in (\Delta^{\mathcal{S}})^{\mathcal{S} \times \mathcal{A}} : \|\xi'(s, :, :) - \widehat{\xi}(s, :, :)\|_p \leq \alpha_s, \forall s \in \mathcal{S}\}$ where $p \in \{1, \infty\}$ and $\widehat{\xi}(s, a, s') > \alpha_s, \forall s, a, s'$.

3. $f(\lambda) = \sum_{s,a} \lambda(s, a)\left[c(s, a) + \mu \log \frac{\lambda(s,a)}{\sum_{a'} \lambda(s,a')}\right]$ (repeat Eq. (7)) as a function of $\lambda \in \Lambda = \{\lambda_{\theta,\xi} : \theta \in \Theta, \xi \in \Xi\}$.

For any $\theta \in \Theta = [-R, R]^{|\mathcal{S}| \times |\mathcal{A}|}$, we have

$$\pi_\theta(a|s) = \frac{\exp(\theta_{s,a})}{\sum_{a'} \exp(\theta_{s,a'})} \geq \pi_{\min} \overset{\text{def}}{=} \frac{\exp(-R)}{\exp(-R) + (|\mathcal{A}| - 1)\exp(R)} > 0. \qquad (73)$$

When $p = 1$ or $p = \infty$, any $\xi \in \Xi$ satisfies

$$p_\xi(s'|s, a) = \xi(s, a, s') \geq \widehat{\xi}(s, a, s') - \|\xi(s, :, :) - \widehat{\xi}(s, :, :)\|_p$$
$$\geq \widehat{\xi}(s, a, s') - \alpha_s$$
$$\geq \xi_{\min} \overset{\text{def}}{=} \min_{s,a,s'}[\widehat{\xi}(s, a, s') - \alpha_s] > 0, \qquad (74)$$

where $\xi_{\min} > 0$ since it is minimum over finitely many positive numbers $\widehat{\xi}(s, a, s') - \alpha_s$. Then for any $\theta \in \Theta$ and $\xi \in \Xi$, we have

$$\lambda_{\theta,\xi}(s, a) \overset{\text{def}}{=} (1 - \gamma) \sum_{t=0}^{+\infty} \gamma^t \mathbb{P}_{\pi_\theta, p_\xi}(s_t = s, a_t = a|s_0 \sim \rho)$$
$$\geq \gamma(1 - \gamma)\mathbb{P}_{\pi_\theta, p_\xi}(s_1 = s, a_1 = a|s_0 \sim \rho)$$
$$= \gamma(1 - \gamma) \sum_{s',a'} \rho(s')\pi_\theta(a'|s')p_\xi(s|s', a')\pi_\theta(a|s)$$
$$\geq \gamma(1 - \gamma) \sum_{s',a'} \rho(s')\pi_\theta(a'|s')\xi_{\min}\pi_{\min}$$
$$= \lambda_{\min} \overset{\text{def}}{=} \xi_{\min}\pi_{\min}\gamma(1 - \gamma) > 0. \qquad (75)$$

Finally, for any $(s, a, s'), (s_1, a_1, s_1'), (s_2, a_2, s_2') \in \mathcal{S} \times \mathcal{A} \times \mathcal{S}$, we obtain all the derivative bounds as follows, where $\mathbb{1}\{\cdot\}$ is an indicator function.

$$\frac{\partial \log \pi_\theta(a|s)}{\partial \theta(s_1, a_1)} = \mathbb{1}\{s_1 = s\}\left[\mathbb{1}\{a_1 = a\} - \pi_\theta(a_1|s)\right] \in [-1, 1].$$

$$\frac{\partial^2 \log \pi_\theta(a|s)}{\partial\theta(s_1,a_1)\partial\theta(s_2,a_2)} = -\mathbb{1}\{s_1 = s\}\pi_\theta(a_1|s)\frac{\partial \log \pi_\theta(a|s)}{\partial\theta(s_2,a_2)} \in [-1,1].$$

$$\frac{\partial \log p_\xi(s'|s,a)}{\partial\xi(s_1,a_1,s_1')} = \frac{1}{\xi(s,a,s')}\mathbb{1}\{(s_1,a_1,s_1') = (s,a,s')\} \in [0,\xi_{\min}^{-1}].$$

$$\frac{\partial^2 \log p_\xi(s'|s,a)}{\partial\xi(s_1,a_1,s_1')\partial\xi(s_2,a_2,s_2')} = -\xi^{-2}(s,a,s')\mathbb{1}\{(s_1,a_1,s_1')=(s_2,a_2,s_2')=(s,a,s')\} \in [-\xi_{\min}^{-2},0].$$

$$\frac{\partial f(\lambda)}{\partial\lambda(s_1,a_1)} = c(s_1,a_1)+\log\frac{\lambda(s_1,a_1)}{\sum_{a'}\lambda(s_1,a')}+1-\frac{\lambda(s_1,a_1)}{\sum_{a'}\lambda(s_1,a')} \in \Big[c_{\min}-\log(|\mathcal{A}|\lambda_{\min}), c_{\max}+1\Big],$$

where $c_{\min} = \min_{s,a} c(s,a)$ and $c_{\max} = \max_{s,a} c(s,a)$.

$$\frac{\partial^2 f(\lambda)}{\partial\lambda(s_1,a_1)\partial\lambda(s_2,a_2)} = \mathbb{1}\{s_1 = s_2\}\Big[\frac{\mathbb{1}\{a_1 = a_2\}}{\lambda(s_1,a_1)} - \frac{1}{\sum_{a'}\lambda(s_1,a')} - \frac{\mathbb{1}\{a_1 = a_2\}-\lambda(s_1,a_1)}{[\sum_{a'}\lambda(s_1,a')]^2}\Big]$$

$$\in \Big[-\frac{1}{|\mathcal{A}|\lambda_{\min}} - \frac{1}{|\mathcal{A}|^2\lambda_{\min}^2}, \frac{1}{\lambda_{\min}} + \frac{1}{|\mathcal{A}|\lambda_{\min}}\Big].$$

### K.2  Proof of Assumptions 5, 6 and 7 about ambiguity set $\Xi$

It is straightforward to verify that the ambiguity set $\Xi = \{\xi \in (\Delta^{\mathcal{S}})^{\mathcal{S}\times\mathcal{A}} : \|\xi(s,:,:) - \widehat{\xi}(s,:,:)\|_p \le \alpha_s, \forall s \in \mathcal{S}\}$ ($p \in \{1,\infty\}$) is convex and compact, which proves Assumption 5.

Assumption 6 can be proved easily by letting $\Xi_s = \{\xi_s \in (\Delta^{\mathcal{S}})^{\mathcal{A}} : \|\xi_s - \widehat{\xi}(s,:,:)\|_p \le \alpha_s\}$ ($p \in \{1,\infty\}$).

Then we will prove Assumption 7, that is, $\Xi_s = \{\xi_s \in (\Delta^{\mathcal{S}})^{\mathcal{A}} : \|\xi_s - \widehat{\xi}(s,:,:)\|_p \le \alpha_s\}$ ($p \in \{1,\infty\}$) is a polyhedron. Based on Definition 1.1 and Theorem 1.26 of [9], it is equivalent to prove that $\Xi_s$ is bounded (already proved above) and is an intersection of finitely many closed half-planes (obvious based on the definitions of $\|\cdot\|_1$ and $\|\cdot\|_\infty$).

### K.3  Proof of Assumptions 4 and 8

We will only prove Assumption 8, since Assumption 4 can be proved in the same way.

Fix any $\xi \in \Xi$ and $\theta \in \Theta = [-R,R]^{|\mathcal{S}|\times|\mathcal{A}|}$. Then we select any $\theta^* = \theta^*(\theta) \in \arg\min_{\theta'\in\Theta_{\min}} \|\theta' - \theta\|_\infty$ where $\Theta_{\min} := \arg\min_{\theta'\in\Theta} \Gamma(\theta')$ is a compact set since $\Gamma$ is a continuous function.

Define the following notations.

- $\lambda_{\theta,\xi}^{(\delta)} = (1-\delta)\lambda_{\theta,\xi} + \delta\lambda_{\theta^*,\xi}$ for $\delta \in [0,1]$ (we select $\overline{\delta} = 1$).
- Policy $\pi_{\theta,\xi}^{(\delta)}$ defined as $\pi_{\theta,\xi}^{(\delta)}(a|s) = \frac{\lambda_{\theta,\xi}^{(\delta)}(s,a)}{\lambda_{\theta,\xi}^{(\delta)}(s)}$ where $\lambda_{\theta,\xi}^{(\delta)}(s) = \sum_{a'} \lambda_{\theta,\xi}^{(\delta)}(s,a')$

(Note that $\lambda_{\theta,\xi}^{(\delta)}(s) \ge \lambda_{\theta,\xi}^{(\delta)}(s,a) \ge \lambda_{\min} > 0$, so $\lambda_{\theta,\xi}^{(\delta)}(s)$ can be the denominator and $\pi_{\theta,\xi}^{(\delta)}(a|s) > 0$).

- $\theta_{\theta,\xi}^{(\delta)} \in \mathbb{R}^{|\mathcal{S}|\times\mathcal{A}}$ with each entry defined as follows.

$$(\theta_{\theta,\xi}^{(\delta)})_{s,a} = \log\Bigg[\frac{(1-\delta)\lambda_{\theta,\xi}(s,a) + \delta\lambda_{\theta^*,\xi}(s,a)}{(1-\delta)\lambda_{\theta,\xi}(s,a)\exp(-\theta_{s,a}) + \delta\lambda_{\theta^*,\xi}(s,a)\exp(-\theta_{s,a}^*)}\Bigg], \tag{76}$$

which is valid since $\pi_{\theta,\xi}^{(\delta)}(a|s) > 0$ and $\pi_\theta(a|s) \ge \pi_{\min} > 0$.

- $\mathcal{U}_{\theta,\xi} = \{\theta_{\theta,\xi}^{(\delta)} : \delta \in [0,1]\} \subset \mathbb{R}^{|\mathcal{S}|\times|\mathcal{A}|}$.
- $\mathcal{V}_{\theta,\xi} = \{\lambda_{\theta,\xi}^{(\delta)} : \delta \in [0,1]\} \subset \Delta^{\mathcal{S}\times\mathcal{A}}$.

Now, it remains to prove the following two statements.
(P1): $\theta_{\theta,\xi}^{(0)} = \theta$ and $\theta_{\theta,\xi}^{(1)} = \theta^*$.
(P2): $\mathcal{U}_{\theta,\xi} \subset \Theta = [-R,R]^{|\mathcal{S}|\times|\mathcal{A}|}$.

(P3): $\lambda_{\theta,\xi}^{(\delta)} = \lambda_{\theta_{\theta,\xi}^{(\delta)},\xi}$.

(P4): The mapping $\theta_{\theta,\xi}^{(\delta)} \to \lambda_{\theta,\xi}^{(\delta)}$ from $\mathcal{U}_{\theta,\xi}$ to $\mathcal{V}_{\theta,\xi}$ is a bijection and is Lipschitz continuous in both directions.

(P1) obviously follows from Eq. (76).

Note that $(\theta_{\theta,\xi}^{(\delta)})_{s,a}$ defined by Eq. (76) is a monotone function of $\delta \in [0,1]$, and (P1) implies that $(\theta_{\theta,\xi}^{(0)})_{s,a} = \theta_{s,a} \in [-R,R]$ and $\theta_{\theta,\xi}^{(1)} = \theta_{s,a}^* \in [-R,R]$. Therefore, $(\theta_{\theta,\xi}^{(\delta)})_{s,a} \in [-R,R]$ which proves (P2).

To prove (P3), rewrite Eq. (76) as follows.

$$
\begin{aligned}
&(\theta_{\theta,\xi}^{(\delta)})_{s,a} \\
&= \log[\lambda_{\theta,\xi}^{(\delta)}(s,a)] - \log\Big[(1-\delta)\lambda_{\theta,\xi}(s)\pi_\theta(a|s)\exp(-\theta_{s,a}) + \delta\lambda_{\theta^*,\xi}(s)\pi_{\theta^*}(a|s)\exp(-\theta_{s,a}^*)\Big] \\
&= \log[\lambda_{\theta,\xi}^{(\delta)}(s,a)] - \log\Big[\frac{(1-\delta)\lambda_{\theta,\xi}(s)}{\sum_{a'}\exp(\theta_{s,a'})} + \frac{\delta\lambda_{\theta^*,\xi}(s)}{\sum_{a'}\exp(\theta_{s,a'}^*)}\Big] \\
&= \log\Big[\lambda_{\theta,\xi}^{(\delta)}(s)\pi_{\theta,\xi}^{(\delta)}(a|s)\Big] - \log\Big[\frac{(1-\delta)\lambda_{\theta,\xi}(s)}{\sum_{a'}\exp(\theta_{s,a'})} + \frac{\delta\lambda_{\theta^*,\xi}(s)}{\sum_{a'}\exp(\theta_{s,a'}^*)}\Big] \\
&= \log\Big[\pi_{\theta,\xi}^{(\delta)}(a|s)\Big] + c_\delta(s), \quad\quad\quad\quad\quad\quad\quad\quad\quad\quad (77)
\end{aligned}
$$

where we denote $c_\delta(s) := \log\Big[\lambda_{\theta,\xi}^{(\delta)}(s)\Big] - \log\Big[\frac{(1-\delta)\lambda_{\theta,\xi}(s)}{\sum_{a'}\exp(\theta_{s,a'})} + \frac{\delta\lambda_{\theta^*,\xi}(s)}{\sum_{a'}\exp(\theta_{s,a'}^*)}\Big]$. Therefore,

$$
\pi_{\theta_{\theta,\xi}^{(\delta)}}(a|s) = \frac{\exp\big[(\theta_{\theta,\xi}^{(\delta)})_{s,a}\big]}{\sum_{a'}\exp\big[(\theta_{\theta,\xi}^{(\delta)})_{s,a'}\big]} = \pi_{\theta,\xi}^{(\delta)}(a|s) = \frac{\lambda_{\theta,\xi}^{(\delta)}(s,a)}{\lambda_{\theta,\xi}^{(\delta)}(s)}. \quad\quad (78)
$$

Note that for any $\theta' \in \mathbb{R}^{|\mathcal{S}| \times |\mathcal{A}|}$, we have

$$
\begin{aligned}
\lambda_{\theta',\xi}(s') &= \sum_{a'} \lambda_{\theta',\xi}(s',a') \\
&\overset{(i)}{=} \sum_{a'} \Big[(1-\gamma)\rho(s') + \gamma\sum_{s,a}\lambda_{\theta',\xi}(s,a)p_\xi(s'|s,a)\Big]\pi_{\theta'}(a'|s') \\
&= (1-\gamma)\rho(s') + \gamma\sum_{s,a}\lambda_{\theta',\xi}(s,a)p_\xi(s'|s,a) \quad\quad\quad (79)
\end{aligned}
$$

where (i) uses Lemma 2. Then we have

$$
\begin{aligned}
&\Big[(1-\gamma)\rho(s') + \gamma\sum_{s,a}\lambda_{\theta,\xi}^{(\delta)}(s,a)p_\xi(s'|s,a)\Big]\pi_{\theta_{\theta,\xi}^{(\delta)}}(a'|s') - \lambda_{\theta,\xi}^{(\delta)}(s',a') \\
&\overset{(i)}{=} \pi_{\theta_{\theta,\xi}^{(\delta)}}(a'|s')\Big[(1-\gamma)\rho(s') + \gamma\sum_{s,a}\lambda_{\theta,\xi}^{(\delta)}(s,a)p_\xi(s'|s,a) - \lambda_{\theta,\xi}^{(\delta)}(s')\Big] \\
&\overset{(ii)}{=} \delta\pi_{\theta_{\theta,\xi}^{(\delta)}}(a'|s')\Big[(1-\gamma)\rho(s') + \gamma\sum_{s,a}\lambda_{\theta,\xi}(s,a)p_\xi(s'|s,a) - \lambda_{\theta,\xi}(s')\Big] \\
&\quad + (1-\delta)\pi_{\theta_{\theta,\xi}^{(\delta)}}(a'|s')\Big[(1-\gamma)\rho(s') + \gamma\sum_{s,a}\lambda_{\theta^*,\xi}(s,a)p_\xi(s'|s,a) - \lambda_{\theta^*,\xi}(s')\Big] \\
&\overset{(iii)}{=} 0,
\end{aligned}
$$

where (i) uses Eq. (78), (ii) uses $\lambda_{\theta,\xi}^{(\delta)} = (1-\delta)\lambda_{\theta,\xi} + \delta\lambda_{\theta^*,\xi}$ and (iii) uses the Eq. (79) for $\theta' \in \{\theta^*,\theta\}$. Based on Lemma 2, the equality above implies (P3).

Next, we prove (P4). Note that the mapping from $\theta$ to $\lambda_{\theta,\xi}^{(\delta)} = \lambda_{\theta_{\theta,\xi}^{(\delta)},\xi}$ is Lipschitz continuous based on Lemma 3. Hence, we only need to consider its reverse mapping.

If $\lambda_{\theta^*,\xi} = \lambda_{\theta,\xi}$, then $\pi_{\theta^*} = \pi_\theta$. Hence, $\theta \in \Theta_{\min} := \arg\min_{\theta' \in \Theta} \Gamma(\theta')$ because $\Gamma(\theta') = \max_{\xi' \in \Xi} f(\lambda_{\theta',\xi'})$ can be seen as a function of $\pi_{\theta'}$. Therefore, $\theta^* = \theta$ which means $\mathcal{U}_{\theta,\xi} = \{\theta\}$ and $\mathcal{V}_{\theta,\xi} = \{\lambda_{\theta,\xi}\}$ are singletons. In this case, (P4) trivially holds.

Therefore, we focus on the case where $\lambda_{\theta^*,\xi} \neq \lambda_{\theta,\xi}$. Before proving (P4), we will prove the following statement.

(P5) There exists a constant $L' > 0$ such that $\|\theta^* - \theta\|_\infty \leq L'\|\lambda_{\theta^*,\xi} - \lambda_{\theta,\xi}\|_\infty$ for any $\theta \in \Theta = [-R,R]^{|\mathcal{S}| \times |\mathcal{A}|}$.

Define $\theta'^* \in \theta \in \mathbb{R}^{|\mathcal{S}| \times |\mathcal{A}|}$ such that $\theta'^*_{s,a} = \theta^*_{s,a} + \frac{1}{|\mathcal{A}|}\sum_{a'}(\theta_{s,a'} - \theta^*_{s,a'})$. Then it can be easily verified that

$$\pi_{\theta'^*,\xi} = \pi_{\theta^*,\xi}, \tag{80}$$

$$\sum_{a'} \theta'^*_{s,a'} = \sum_{a'} \theta_{s,a'}. \tag{81}$$

Note that $\pi_\theta(a|s) = \frac{\exp(\theta_{s,a})}{\sum_{a'}\exp(\theta_{s,a'})}$ and $\pi_{\theta'}(a|s) = \frac{\exp(\theta'_{s,a})}{\sum_{a'}\exp(\theta'_{s,a'})}$, so $\theta'_{s,a} - \theta_{s,a} = \log\pi_{\theta'}(a|s) - \log\pi_\theta(a|s) + u(s)$ where $u(s) := \log\left[\sum_{a'}\exp(\theta'_{s,a'})\right] - \log\left[\sum_{a'}\exp(\theta_{s,a'})\right]$. Then combining with Eq. (81), we obtain that $u(s) = \frac{1}{|\mathcal{A}|}\sum_{a'}[\log\pi_\theta(a'|s) - \log\pi_{\theta'}(a'|s)]$. Therefore,

$$|\theta'^*_{s,a} - \theta_{s,a}|$$

$$\leq |\log\pi_{\theta'^*}(a|s) - \log\pi_\theta(a|s)| + \frac{1}{|\mathcal{A}|}\sum_{a'}\left|\log\pi_\theta(a'|s) - \log\pi_{\theta'^*}(a'|s)\right|$$

$$\overset{(i)}{\leq} \pi_{\min}^{-1}|\pi_{\theta^*}(a|s) - \pi_\theta(a|s)| + \frac{\pi_{\min}^{-1}}{|\mathcal{A}|}\sum_{a'}\left|\pi_\theta(a'|s) - \pi_{\theta^*}(a'|s)\right|$$

$$\leq 2\pi_{\min}^{-1}\max_{a'}\left|\pi_{\theta^*}(a'|s) - \pi_\theta(a'|s)\right|$$

$$\leq 2\pi_{\min}^{-1}\max_{a'}\left|\frac{\lambda_{\theta^*,\xi}(s,a')}{\lambda_{\theta^*,\xi}(s)} - \frac{\lambda_{\theta,\xi}(s,a')}{\lambda_{\theta,\xi}(s)}\right|$$

$$= 2\pi_{\min}^{-1}\max_{a'}\left[\left|\frac{\lambda_{\theta^*,\xi}(s,a') - \lambda_{\theta,\xi}(s,a')}{\lambda_{\theta^*,\xi}(s)} + \frac{\lambda_{\theta,\xi}(s,a')}{\lambda_{\theta^*,\xi}(s)\lambda_{\theta,\xi}(s)}[\lambda_{\theta,\xi}(s) - \lambda_{\theta^*,\xi}(s)]\right|\right]$$

$$\overset{(ii)}{\leq} \frac{2}{|\mathcal{A}|\lambda_{\min}\pi_{\min}}\max_{a'}|\lambda_{\theta^*,\xi}(s,a') - \lambda_{\theta,\xi}(s,a')|$$

$$\quad + \frac{2}{|\mathcal{A}|^2\lambda_{\min}^2\pi_{\min}} \cdot \sum_{a'}|\lambda_{\theta^*,\xi}(s,a') - \lambda_{\theta,\xi}(s,a')|$$

$$\leq \frac{4\|\lambda_{\theta^*,\xi} - \lambda_{\theta,\xi}\|_\infty}{|\mathcal{A}|\lambda_{\min}^2\pi_{\min}}, \tag{82}$$

where (i) uses Eqs. (73) and (80) which imply that $\pi_\theta(a|s), \pi_{\theta^*}(a|s) = \pi_{\theta'}(a|s) \in [\pi_{\min}, 1]$ for $\theta, \theta^* \in \Theta$, (ii) uses $\lambda_{\theta^*,\xi}(s), \lambda_{\theta,\xi}(s) \geq |\mathcal{A}|\lambda_{\min}$ for $\theta, \theta^* \in \Theta$ as a result of Eq. (75).

Based on the definition of $\theta'^*$, we have $\max_{a'}\theta'^*_{s,a'} - \min_{a'}\theta'^*_{s,a'} = \max_{a'}\theta_{s,a'} - \min_{a'}\theta_{s,a'} \leq 2R$. Therefore, for each $s \in \mathcal{S}$, there are three cases: $-R \leq \min_{a'}\theta'^*_{s,a'} \leq \max_{a'}\theta'^*_{s,a'} \leq R$, $\max_{a'}\theta'^*_{s,a'} > R$ and $\min_{a'}\theta'^*_{s,a'} < -R$, and we can define $\theta'' \in \mathbb{R}^{|\mathcal{S}| \times |\mathcal{A}|}$ as follows

$$\theta''_{s,a} = \begin{cases} \theta'^*_{s,a}, & -R \leq \min_{a'}\theta'^*_{s,a'} \leq \max_{a'}\theta'^*_{s,a'} \leq R \\ \theta'^*_{s,a} - \max_{a'}\theta'^*_{s,a'} + R, & \max_{a'}\theta'^*_{s,a'} > R \\ \theta'^*_{s,a} - \min_{a'}\theta'^*_{s,a'} - R, & \min_{a'}\theta'^*_{s,a'} < -R \end{cases}.$$

It can be easily verified that the $\theta''$ defined above satisfies $\theta'' \in \Theta = [-R,R]^{|\mathcal{S}| \times |\mathcal{A}|}$ (since $\max_{a'}\theta'^*_{s,a'} - \min_{a'}\theta'^*_{s,a'} \leq 2R$) and $\pi_{\theta''} = \pi_{\theta'^*} = \pi_{\theta^*}$ (the second = comes from Eq. (80)). Therefore, $\theta'' \in \Theta_{\min}$ and thus

$$\|\theta^* - \theta\|_\infty \overset{(i)}{\leq} \|\theta'' - \theta\|_\infty \leq \|\theta'' - \theta'^*\|_\infty + \|\theta'^* - \theta\|_\infty, \tag{83}$$

where (i) uses $\theta'' \in \Theta_{\min}$ and $\theta^* \in \arg\min_{\theta' \in \Theta_{\min}} \|\theta' - \theta\|_\infty$. To further obtain the upper bound of $\|\theta'' - \theta'^*\|_\infty$ in the above inequality, we discuss the three aforementioned cases.

(Case I): When $-R \leq \min_{a'} \theta'^*_{s,a'} \leq \max_{a'} \theta'^*_{s,a'} \leq R$, we have $\theta''_{s,a} - \theta'^*_{s,a} = 0$.

(Case II): When $\max_{a'} \theta'^*_{s,a'} > R$, we have

$$0 < \theta'^*_{s,a} - \theta''_{s,a} = \max_{a'} \theta'^*_{s,a'} - R \overset{(i)}{\leq} \max_{a'} \theta'^*_{s,a'} - \max_{a'} \theta_{s,a'} \leq \|\theta'^* - \theta\|_\infty,$$

where (i) uses $\theta \in \Theta = [-R, R]^{|\mathcal{S}| \times |\mathcal{A}|}$.

(Case III): When $\min_{a'} \theta'^*_{s,a'} < -R$, we have

$$0 < \theta''_{s,a} - \theta'^*_{s,a} = -\min_{a'} \theta'^*_{s,a'} - R \overset{(i)}{\leq} \min_{a'} \theta_{s,a'} - \min_{a'} \theta'^*_{s,a'} \leq \|\theta'^* - \theta\|_\infty,$$

where (i) uses $\theta \in \Theta = [-R, R]^{|\mathcal{S}| \times |\mathcal{A}|}$.

Summarizing the above three cases, we obtain that $\|\theta'' - \theta'^*\|_\infty \leq \|\theta'^* - \theta\|_\infty$ and therefore Eq. (83) implies that

$$\|\theta^* - \theta\|_\infty \leq \|\theta'' - \theta'^*\|_\infty + \|\theta'^* - \theta\|_\infty \leq 2\|\theta'^* - \theta\|_\infty \overset{(i)}{\leq} \frac{8\|\lambda_{\theta^*,\xi} - \lambda_{\theta,\xi}\|_\infty}{|\mathcal{A}|\lambda_{\min}^2 \pi_{\min}} \overset{\text{def}}{=} L', \quad (84)$$

where (i) uses Eq. (82). This proves (P5).

We consider Eq. (76) as a function of $\delta$ and take its derivative as follows.

$$\left| \frac{\partial(\theta_{\theta,\xi}^{(\delta)})_{s,a}}{\partial \delta} \right|$$

$$= \left| \frac{\lambda_{\theta^*,\xi}(s,a) - \lambda_{\theta,\xi}(s,a)}{(1-\delta)\lambda_{\theta,\xi}(s,a) + \delta\lambda_{\theta^*,\xi}(s,a)} + \frac{\lambda_{\theta^*,\xi}(s,a)\exp(-\theta_{s,a}^*) - \lambda_{\theta,\xi}(s,a)\exp(-\theta_{s,a})}{(1-\delta)\lambda_{\theta,\xi}(s,a)\exp(-\theta_{s,a}) + \delta\lambda_{\theta^*,\xi}(s,a)\exp(-\theta_{s,a}^*)} \right|$$

$$\overset{(i)}{\leq} \frac{\|\lambda_{\theta^*,\xi} - \lambda_{\theta,\xi}\|_\infty}{\lambda_{\min}} + \frac{|\lambda_{\theta^*,\xi}(s,a)[\exp(-\theta_{s,a}^*) - \exp(-\theta_{s,a})] + \exp(-\theta_{s,a})[\lambda_{\theta^*,\xi}(s,a) - \lambda_{\theta,\xi}(s,a)]|}{\lambda_{\min}\exp(-R)}$$

$$\overset{(ii)}{\leq} \frac{\|\lambda_{\theta^*,\xi} - \lambda_{\theta,\xi}\|_\infty}{\lambda_{\min}} + \frac{\exp(R)|\theta_{s,a}^* - \theta_{s,a}| + \exp(R)|\lambda_{\theta^*,\xi}(s,a) - \lambda_{\theta,\xi}(s,a)|}{\lambda_{\min}\exp(-R)}$$

$$\overset{(iii)}{\leq} \frac{\|\lambda_{\theta^*,\xi} - \lambda_{\theta,\xi}\|_\infty}{\lambda_{\min}} + \exp(2R) \cdot \frac{L'\|\lambda_{\theta^*,\xi} - \lambda_{\theta,\xi}\|_\infty + \|\lambda_{\theta^*,\xi} - \lambda_{\theta,\xi}\|_\infty}{\lambda_{\min}}$$

$$\leq \frac{2(L'+1)\exp(2R)}{\lambda_{\min}}\|\lambda_{\theta^*,\xi} - \lambda_{\theta,\xi}\|_\infty \quad (85)$$

where (i) uses $\theta, \theta^* \in \Theta = [-R, R]^{|\mathcal{S}| \times |\mathcal{A}|}$ and Eq. (75), (ii) uses $\theta, \theta^* \in \Theta = [-R, R]^{|\mathcal{S}| \times |\mathcal{A}|}$, (iii) uses Eq. (84).

Therefore, for any $\delta, \delta' \in [0, 1]$, we have

$$\|\theta_{\theta,\xi}^{(\delta')} - \theta_{\theta,\xi}^{(\delta)}\|_\infty \leq \frac{2(L'+1)\exp(2R)}{\lambda_{\min}}\|\lambda_{\theta^*,\xi} - \lambda_{\theta,\xi}\|_\infty |\delta' - \delta|$$

$$= \frac{2(L'+1)\exp(2R)}{\lambda_{\min}}\|\lambda_{\theta,\xi}^{(\delta')} - \lambda_{\theta,\xi}^{(\delta)}\|_\infty,$$

which proves the statement (P4) and thus proves Assumption 8.

Assumption 4 can be proved in the same way simply by replacing $\theta^*$ with any $\theta^*(\xi) \in \arg\min_{\theta' \in \Theta} f(\lambda_{\theta',\xi})$.

## L Proof of Proposition 9

The estimated occupancy measure (40) is an unbiased estimator of the following truncated occupancy measure with truncation level $H_\lambda$.

$$\lambda_{\theta,\xi}^{(H_\lambda)}(s,a) \overset{\text{def}}{=} (1-\gamma) \sum_{t=0}^{H_\lambda - 1} \gamma^t \mathbb{P}_{\pi_\theta, p_\xi}(s_t = s, a_t = a | s_0 \sim \rho). \quad (86)$$

Denote $\widehat{\lambda}(\tau^{(\lambda)}) := \left[\widehat{\lambda}(\tau^{(\lambda)}; s, a)\right]_{s,a \in \mathcal{S} \times \mathcal{A}} \in \mathbb{R}^{|\mathcal{S}||\mathcal{A}|}$, $\lambda_{\theta,\xi}^{(H_\lambda)} := \left[\lambda_{\theta,\xi}^{(H_\lambda)}(s, a)\right]_{s,a \in \mathcal{S} \times \mathcal{A}} \in \mathbb{R}^{|\mathcal{S}||\mathcal{A}|}$, $\lambda_{\theta,\xi} := \left[\lambda_{\theta,\xi}(s, a)\right]_{s,a \in \mathcal{S} \times \mathcal{A}} \in \mathbb{R}^{|\mathcal{S}||\mathcal{A}|}$, Then the estimation error of occupancy measure has the following upper bound.

$$
\mathbb{E}_{\pi_\theta, p_\xi} \left\| \widehat{\lambda}(\tau^{(\lambda)}) - \lambda_{\theta,\xi} \right\|^2
$$

$$
\stackrel{(i)}{=} \mathrm{Var}_{\pi_\theta, p_\xi} \left[ \widehat{\lambda}(\tau^{(\lambda)}) \right] + \mathbb{E}_{\pi_\theta, p_\xi} \left\| \lambda_{\theta,\xi}^{(H_\lambda)} - \lambda_{\theta,\xi} \right\|^2
$$

$$
\stackrel{(ii)}{=} \frac{1}{m_\lambda} \mathrm{Var}\left[ \widehat{\lambda}_1(\tau_1^{(\lambda)}) \right] + \sum_{s,a} \left| (1 - \gamma) \sum_{t=H_\lambda}^{+\infty} \gamma^t \mathbb{P}_{\pi_\theta, p_\xi}(s_t = s, a_t = a | s_0 \sim \rho) \right|^2
$$

$$
\stackrel{(iii)}{\leq} \frac{1}{m_\lambda} \mathbb{E} \left\| \widehat{\lambda}_1(\tau_1^{(\lambda)}) \right\|^2 + \left[ (1 - \gamma) \sum_{t=H_\lambda}^{+\infty} \gamma^t \right] \sum_{s,a} \left[ (1 - \gamma) \sum_{t=H_\lambda}^{+\infty} \gamma^t \mathbb{P}_{\pi_\theta, p_\xi}(s_t = s, a_t = a | s_0 \sim \rho) \right]
$$

$$
\stackrel{(iv)}{\leq} \frac{1}{m_\lambda} + \gamma^{2H_\lambda}, \tag{87}
$$

where (i) uses $\mathbb{E}\|X\|^2 = \mathrm{Var}X + \|\mathbb{E}X\|^2$ for random vector $X := \widehat{\lambda}(\tau^{(\lambda)}) - \lambda_{\theta,\xi}$, (ii) uses Eqs. (1) and (86) and uses the fact that $\widehat{\lambda}$ defined by Eq. (40) is average among the $m_\lambda$ i.i.d. individual estimators $\widehat{\lambda}_i(\tau_i^{(\lambda)}; s, a) := (1 - \gamma) \sum_{h=0}^{H_\lambda - 1} \gamma^h \mathbb{1}\{s_{i,h}^{(\lambda)} = s, a_{i,h}^{(\lambda)} = a\}$ for $i = 1, \ldots, m_\lambda$, (iii) uses $\mathrm{Var}X \leq \mathbb{E}\|X\|^2$ for random vector $X := \widehat{\lambda}_1(\tau^{(\lambda)})$ and $\mathbb{P}_{\pi_\theta, p_\xi}(s_t = s, a_t = a | s_0 \sim \rho) \in [0, 1]$, and (iv) uses $0 \leq \|\widehat{\lambda}_1(\tau_i^{(\lambda)})\|^2 \leq \sum_{s,a} \widehat{\lambda}_1(\tau_i^{(\lambda)}; s, a) = 1$ and $\sum_{s,a} \mathbb{P}_{\pi_\theta, p_\xi}(s_t = s, a_t = a | s_0 \sim \rho) = 1$.

Define the cost function as $c := \nabla_\lambda f(\lambda_{\theta,\xi})$. The error of estimating $c$ by $\widehat{c} := \nabla_\lambda f[\widehat{\lambda}(\tau^{(\lambda)})]$ has the following upper bounds.

$$
\mathbb{E}_{\pi_\theta, p_\xi} \|\widehat{c} - c\|_\infty^2 = \mathbb{E}_{\pi_\theta, p_\xi} \|\nabla_\lambda f[\widehat{\lambda}(\tau^{(\lambda)})] - \nabla_\lambda f(\lambda_{\theta,\xi})\|_\infty^2
$$

$$
\stackrel{(i)}{\leq} L_\lambda^2 \mathbb{E}_{\pi_\theta, p_\xi} \|\widehat{\lambda}(\tau^{(\lambda)}) - \lambda_{\theta,\xi}\|^2
$$

$$
\stackrel{(ii)}{\leq} L_\lambda^2 \left( \frac{1}{m_\lambda} + \gamma^{2H_\lambda} \right) \tag{88}
$$

$$
\mathbb{E}_{\pi_\theta, p_\xi} \|\widehat{c} - c\|^2 \leq |\mathcal{S}||\mathcal{A}| \mathbb{E}_{\pi_\theta, p_\xi} \|\widehat{c} - c\|_\infty^2 \leq L_\lambda^2 |\mathcal{S}||\mathcal{A}| \left( \frac{1}{m_\lambda} + \gamma^{2H_\lambda} \right) \tag{89}
$$

where (i) uses Assumption 2 and (ii) uses Eq. (87). Also, $c$ and $\widehat{c}$ have the following norm bound based on Assumption 2.

$$
\max\left( \|c\|_\infty, \|\widehat{c}\|_\infty \right) \leq \ell_\lambda. \tag{90}
$$

Note that $g^{(\theta)}(\tau^{(\theta)}, \theta, \xi, c)$ defined by Eq. (41) (replace $\widehat{c}$ with $c$) is the average of the following $m_\theta$ i.i.d. individual stochastic gradients.

$$
g_i^{(\theta)}(\tau_i^{(\theta)}, \theta, \xi, c) = \sum_{t=0}^{H_\theta - 1} \gamma^t c(s_{i,t}^{(\theta)}, a_{i,t}^{(\theta)}) \sum_{h=0}^{t} \nabla_\theta \log \pi_\theta(a_{i,h}^{(\theta)} | s_{i,h}^{(\theta)}). \tag{91}
$$

Then it can be easily seen that $g_i^{(\theta)}(\tau_i^{(\theta)}, \theta, \xi, c)$ defined above is an unbiased estimator of the following truncated policy gradient.

$$
\nabla_\theta f^{(H_\theta)}(\lambda_{\theta,\xi}) = \mathbb{E}_{\pi_\theta, p_\xi} \left[ \sum_{t=0}^{H_\theta - 1} \gamma^t c(s_t, a_t) \sum_{h=0}^{t} \nabla_\theta \log \pi_\theta(a_h | s_h) \right]. \tag{92}
$$

Also, $g_i^{(\theta)}(\tau_i^{(\theta)}, \theta, \xi, c)$ can be bounded as follows by using Eq. (90) and Assumption 1.

$$
\|g_i^{(\theta)}(\tau_i^{(\theta)}, \theta, \xi, c)\| \leq \sum_{t=0}^{H_\theta - 1} \gamma^t |c(s_{i,t}^{(\theta)}, a_{i,t}^{(\theta)})| \sum_{h=0}^{t} \|\nabla_\theta \log \pi_\theta(a_{i,h}^{(\theta)} | s_{i,h}^{(\theta)})\|
$$

$$\leq \sum_{t=0}^{H_\theta-1}(t+1)\gamma^t\ell_\lambda\ell_{\pi_\theta} \leq \frac{\ell_\lambda\ell_{\pi_\theta}}{(1-\gamma)^2}. \tag{93}$$

Therefore, we can prove Eq. (43) as follows, and Eq. (44) can be proved using the same logic.

$$\mathbb{E}_{\pi_\theta,p_\xi}\|g^{(\theta)}(\tau^{(\theta)},\theta,\xi,\widehat{c}) - \nabla_\theta f(\lambda_{\theta,\xi})\|^2$$

$$\leq 3\mathbb{E}_{\pi_\theta,p_\xi}\|g^{(\theta)}(\tau^{(\theta)},\theta,\xi,\widehat{c}) - g^{(\theta)}(\tau^{(\theta)},\theta,\xi,c)\|^2$$

$$+ 3\mathbb{E}_{\pi_\theta,p_\xi}\|g^{(\theta)}(\tau^{(\theta)},\theta,\xi,c) - \nabla_\theta f^{(H_\theta)}(\lambda_{\theta,\xi})\|^2 + 3\|\nabla_\theta f^{(H_\theta)}(\lambda_{\theta,\xi}) - \nabla_\theta f(\lambda_{\theta,\xi})\|^2$$

$$\overset{(i)}{\leq} 3\mathbb{E}_{\pi_\theta,p_\xi}\Big\|\frac{1}{m_\theta}\sum_{i=1}^{m_\theta}\sum_{t=0}^{H_\theta-1}\gamma^t\big[\widehat{c}(s_{i,t}^{(\theta)},a_{i,t}^{(\theta)}) - c(s_{i,t}^{(\theta)},a_{i,t}^{(\theta)})\big]\sum_{h=0}^{t}\nabla_\theta\log\pi_\theta(a_{i,h}^{(\theta)}\mid s_{i,h}^{(\theta)})\Big\|^2$$

$$+ 3\mathrm{Var}_{\pi_\theta,p_\xi}[g^{(\theta)}(\tau^{(\theta)},\theta,\xi,c)] + 3\Big\|\mathbb{E}_{\pi_\theta,p_\xi}\sum_{t=H_\theta}^{+\infty}\gamma^t c(s_t,a_t)\sum_{h=0}^{t}\nabla_\theta\log\pi_\theta(a_h|s_h)\Big\|^2$$

$$\overset{(ii)}{\leq} 3\mathbb{E}_{\pi_\theta,p_\xi}\Big[\frac{1}{m_\theta}\sum_{i=1}^{m_\theta}\sum_{t=0}^{H_\theta-1}\gamma^t\|\widehat{c}-c\|_\infty(t+1)\ell_{\pi_\theta}\Big]^2$$

$$+ \frac{3}{m_\theta}\mathrm{Var}_{\pi_\theta,p_\xi}[g_1^{(\theta)}(\tau_1^{(\theta)},\theta,\xi,c)] + 3\Big(\sum_{t=H_\theta}^{+\infty}\gamma^t\ell_\lambda\ell_{\pi_\theta}(t+1)\Big)^2$$

$$\overset{(iii)}{\leq} 3L_\lambda^2|\mathcal{S}||\mathcal{A}|\Big(\frac{1}{m_\lambda}+\gamma^{2H_\lambda}\Big)\Big(\frac{\ell_{\pi_\theta}}{(1-\gamma)^2}\Big)^2 + \frac{3}{m_\theta}\Big(\frac{\ell_\lambda\ell_{\pi_\theta}}{(1-\gamma)^2}\Big)^2 + 3\Big(\frac{\ell_\lambda\ell_{\pi_\theta}\gamma^{H_\theta}[1+H_\theta(1-\gamma)]}{(1-\gamma)^2}\Big)^2$$

$$\leq \frac{3\ell_{\pi_\theta}^2}{(1-\gamma)^4}\Big[L_\lambda^2|\mathcal{S}||\mathcal{A}|\Big(\frac{1}{m_\lambda}+\gamma^{2H_\lambda}\Big) + \frac{\ell_\lambda^2}{m_\theta} + \ell_\lambda^2[1+H_\theta(1-\gamma)]^2\gamma^{2H_\theta}\Big],$$

where (i) uses Eqs. (8),(41),(92) and $\nabla_\theta f^{(H_\theta)}(\lambda_{\theta,\xi}) = \mathbb{E}_{\pi_\theta,p_\xi}g^{(\theta)}(\tau^{(\theta)},\theta,\xi,c)$, (ii) uses Eq. (90), Assumption 1 and $g^{(\theta)}(\tau^{(\theta)},\theta,\xi,c) = \frac{1}{m_\theta}\sum_{i=1}^{m_\theta}g_i^{(\theta)}(\tau_i^{(\theta)},\theta,\xi,c)$ where $\{g_i^{(\theta)}(\tau_i^{(\theta)},\theta,\xi,c)\}_{i=1}^{m_\theta}$ are independent, (iii) uses Eqs. (89) and (93).

## M  Proof of Theorem 1

The policy gradient (8) is proved by Eqs. (5) and (6) of [6]. We will next prove the transition gradient (9).

Under the transition kernel $p_\xi$ and policy $\pi_\theta$, the probability of obtaining a certain trajectory $\tau_t = \{s_h,a_h\}_{h=0}^t \cup \{s_{t+1}\}$ with initial state distribution $\rho$ can be expressed as follows.

$$\mathbb{P}_{\pi_\theta,p_\xi}(\tau_t) = \rho(s_0)\prod_{h=0}^{t}\big[\pi_\theta(a_h|s_h)p_\xi(s_{h+1}|s_h,a_h)\big].$$

Hence, the gradient of the log of the above probability can be computed as follows.

$$\nabla_\xi\log\mathbb{P}_{\pi_\theta,p_\xi}(\tau_t) = \sum_{h=0}^{t}\nabla_\xi\log p_\xi(s_{h+1}|s_h,a_h). \tag{94}$$

Denote $c := \nabla_\lambda f(\lambda_{\theta,\xi})$. Then the transition gradient (9) can be obtained as follows.

$$\nabla_\xi f(\lambda_{\theta,\xi}) = \nabla_\lambda f(\lambda_{\theta,\xi})^\top\nabla_\xi\lambda_{\theta,\xi}$$

$$= \sum_{s,a}c(s,a)\nabla_\xi\sum_{t=0}^{\infty}\gamma^t\mathbb{P}_{\pi_\theta,p_\xi}(s_t=s,a_t=a)$$

$$= \nabla_\xi\sum_{t=0}^{\infty}\int\gamma^t c(s_t,a_t)\mathbb{P}_{\pi_\theta,p_\xi}(\tau_t)\mathrm{d}\tau_t$$

$$= \sum_{t=0}^{\infty} \int \gamma^t c(s_t, a_t) \nabla_\xi \mathbb{P}_{\pi_\theta, p_\xi}(\tau_t) \mathrm{d}\tau_t$$

$$= \sum_{t=0}^{\infty} \int \gamma^t c(s_t, a_t) \left[ \frac{\nabla_\xi \mathbb{P}_{\pi_\theta, p_\xi}(\tau_t)}{\mathbb{P}_{\pi_\theta, p_\xi}(\tau_t)} \right] \cdot \mathbb{P}_{\pi_\theta, p_\xi}(\tau_t) \mathrm{d}\tau_t$$

$$= \sum_{t=0}^{\infty} \mathbb{E}_{\tau_t \sim \mathbb{P}_{\pi_\theta, p_\xi}} \left[ \gamma^t c(s_t, a_t) \nabla_\xi \log \mathbb{P}_{\pi_\theta, p_\xi}(\tau_t) \big| s_0 \sim \rho \right]$$

$$\overset{(i)}{=} \mathbb{E}_{\pi_\theta, p_\xi} \left[ \sum_{t=0}^{\infty} \gamma^t c(s_t, a_t) \sum_{h=0}^{t} \nabla_\xi \log p_\xi(s_{h+1}|s_h, a_h) \bigg| s_0 \sim \rho \right]$$

where (i) uses Eq. (94).

# N    Proof of Theorem 2

Based on Proposition 9, we define the following error terms $E_j^{(\xi)}$ and $E_j^{(\theta)}$ $(i, j \in \{1, 2, 3, 4\})$ to bound the estimation errors of the stochastic gradients in lines 6, 10, 17 and 21 of Algorithm 1 respectively.

$$E_j^{(\theta)} := \frac{3\ell_{\pi_\theta}^2}{(1-\gamma)^4} \left[ L_\lambda^2 |\mathcal{S}||\mathcal{A}| \left( \frac{1}{m_\lambda^{(j)}} + \gamma^{2H_\lambda^{(j)}} \right) + \frac{\ell_\lambda^2}{m_\theta^{(j)}} + \ell_\lambda^2 [1 + H_\theta^{(j)}(1-\gamma)]^2 \gamma^{2H_\theta^{(j)}} \right], \quad (95)$$

$$E_j^{(\xi)} := \frac{3\ell_{p_\xi}^2}{(1-\gamma)^4} \left[ L_\lambda^2 |\mathcal{S}||\mathcal{A}| \left( \frac{1}{m_\lambda^{(j)}} + \gamma^{2H_\lambda^{(j)}} \right) + \frac{\ell_\lambda^2}{m_\xi^{(j)}} + \ell_\lambda^2 [1 + H_\xi^{(j)}(1-\gamma)]^2 \gamma^{2H_\xi^{(j)}} \right]. \quad (96)$$

Then we prove Theorem 2 in the following procedures.

## N.1    Convergence Rate of Inner Update Step (14) of the First Original Phase

**Lemma 7.** *Suppose Assumptions 1-4 hold. Apply the projected stochastic gradient descent step* (14) *in Algorithm 1 to the policy optimization problem* $\min_{\theta \in \Theta} f(\lambda_{\theta, \xi_k})$ *with fixed* $\xi_k \in \Xi$. *Select stepsize* $\alpha = \frac{1}{2L_{\theta, \theta}}$ *and initialization* $\theta_{k,0} = \theta_0$. *Then the output* $\theta_k := \theta_{k,T}$ *globally converges at the following rate for any* $\delta \in [0, \bar{\delta}]$.

$$\mathbb{E}\left[ f(\lambda_{\theta_k, \xi_k}) - \min_{\theta \in \Theta} f(\lambda_{\theta, \xi_k}) \big| \xi_k \right]$$

$$\leq (1-\delta)^T \mathbb{E}\left[ f(\lambda_{\theta_0, \xi_k}) - \min_{\theta \in \Theta} f(\lambda_{\theta, \xi_k}) \big| \xi_k \right] + 4L_{\theta, \theta} \ell_{\lambda^{-1}}^2 \delta + \frac{E_1^{(\theta)}}{\delta L_{\theta, \theta}}, \quad (97)$$

*where* $E_1^{(\theta)}$ *is defined in Eq.* (95).

*Proof.* For any $\theta_{k,t} \in \Theta$ in the update rule (14), there exists at least one optimal policy $\theta_{k,t}^* \in \arg\min_{\theta' \in \Theta} f(\lambda_{\theta', \xi_k})$ such that for any $\delta \in [0, \bar{\delta}]$, there exists $\theta_{k,t}^{(\delta)} \in \Theta$ that satisfies $\lambda_{\theta_{k,t}^{(\delta)}, \xi_k} = (1-\delta)\lambda_{\theta_{k,t}, \xi_k} + \delta\lambda_{\theta_{k,t}^*, \xi_k}$. Since $f$ is convex, we have

$$f(\lambda_{\theta_{k,t}^{(\delta)}, \xi_k}) \leq (1-\delta)f(\lambda_{\theta_{k,t}, \xi_k}) + \delta f(\lambda_{\theta_{k,t}^*, \xi_k}) = (1-\delta)f(\lambda_{\theta_{k,t}, \xi_k}) + \delta \min_{\theta \in \Theta} f(\lambda_{\theta, \xi_k}). \quad (98)$$

Then, we analyze the optimization progress of the stochastic gradient descent step (14) as follows.

$$f(\lambda_{\theta_{k,t+1}, \xi_k})$$

$$\overset{(i)}{\leq} f(\lambda_{\theta_{k,t}, \xi_k}) + \nabla_\theta f(\lambda_{\theta_{k,t}, \xi_k})^\top (\theta_{k,t+1} - \theta_{k,t}) + \frac{L_{\theta, \theta}}{2} \|\theta_{k,t+1} - \theta_{k,t}\|^2$$

$$= f(\lambda_{\theta_{k,t}, \xi_k}) + \left[ \nabla_\theta f(\lambda_{\theta_{k,t}, \xi_k}) - g_{k,t}^{(\theta)} \right]^\top (\theta_{k,t+1} - \theta_{k,t})$$

$$+ (g_{k,t}^{(\theta)})^\top (\theta_{k,t+1} - \theta_{k,t}) + \frac{L_{\theta, \theta}}{2} \|\theta_{k,t+1} - \theta_{k,t}\|^2$$

$$\leq f(\lambda_{\theta_{k,t},\xi_k}) + \frac{1}{2L_{\theta,\theta}}\|\nabla_\theta f(\lambda_{\theta_{k,t},\xi_k}) - g_{k,t}^{(\theta)}\|^2 + \frac{L_{\theta,\theta}}{2}\|\theta_{k,t+1} - \theta_{k,t}\|^2$$

$$+ (g_{k,t}^{(\theta)})^\top(\theta_{k,t+1} - \theta_{k,t}) + \frac{L_{\theta,\theta}}{2}\|\theta_{k,t+1} - \theta_{k,t}\|^2$$

$$\leq f(\lambda_{\theta_{k,t},\xi_k}) + (g_{k,t}^{(\theta)})^\top(\theta_{k,t+1} - \theta_{k,t}) + L_{\theta,\theta}\|\theta_{k,t+1} - \theta_{k,t}\|^2$$

$$+ \frac{1}{2L_{\theta,\theta}}\|\nabla_\theta f(\lambda_{\theta_{k,t},\xi_k}) - g_{k,t}^{(\theta)}\|^2$$

$$\overset{(ii)}{\leq} f(\lambda_{\theta_{k,t},\xi_k}) + (g_{k,t}^{(\theta)})^\top(\theta_{k,t}^{(\delta)} - \theta_{k,t}) + L_{\theta,\theta}\|\theta_{k,t}^{(\delta)} - \theta_{k,t}\|^2$$

$$+ \frac{1}{2L_{\theta,\theta}}\|\nabla_\theta f(\lambda_{\theta_{k,t},\xi_k}) - g_{k,t}^{(\theta)}\|^2$$

$$\leq f(\lambda_{\theta_{k,t},\xi_k}) + \nabla_\theta f(\lambda_{\theta_{k,t},\xi_k})^\top(\theta_{k,t}^{(\delta)} - \theta_{k,t}) - \frac{L_{\theta,\theta}}{2}\|\theta_{k,t}^{(\delta)} - \theta_{k,t}\|^2$$

$$+ \left[g_{k,t}^{(\theta)} - \nabla_\theta f(\lambda_{\theta_{k,t},\xi_k})\right]^\top(\theta_{k,t}^{(\delta)} - \theta_{k,t}) + \frac{3L_{\theta,\theta}}{2}\|\theta_{k,t}^{(\delta)} - \theta_{k,t}\|^2$$

$$+ \frac{1}{2L_{\theta,\theta}}\|\nabla_\theta f(\lambda_{\theta_{k,t},\xi_k}) - g_{k,t}^{(\theta)}\|^2$$

$$\overset{(iii)}{\leq} f(\lambda_{\theta_{k,t}^{(\delta)},\xi_k}) + \frac{1}{2L_{\theta,\theta}}\|\nabla_\theta f(\lambda_{\theta_{k,t},\xi_k}) - g_{k,t}^{(\theta)}\|^2 + \frac{L_{\theta,\theta}}{2}\|\theta_{k,t}^{(\delta)} - \theta_{k,t}\|^2$$

$$+ \frac{3L_{\theta,\theta}}{2}\|\theta_{k,t}^{(\delta)} - \theta_{k,t}\|^2 + \frac{1}{2L_{\theta,\theta}}\|\nabla_\theta f(\lambda_{\theta_{k,t},\xi_k}) - g_{k,t}^{(\theta)}\|^2$$

$$\overset{(iv)}{\leq} (1-\delta)f(\lambda_{\theta_{k,t},\xi_k}) + \delta\min_{\theta\in\Theta} f(\lambda_{\theta,\xi_k}) + \frac{1}{L_{\theta,\theta}}\|\nabla_\theta f(\lambda_{\theta_{k,t},\xi_k}) - g_{k,t}^{(\theta)}\|^2 + 4L_{\theta,\theta}\ell_{\lambda^{-1}}^2\delta^2, \quad (99)$$

where (i) and (iii) use $L_{\theta,\theta}$-smoothness of $f(\lambda_{\cdot,\xi_k})$ based on Proposition 3, (ii) uses the update rule (14) with stepsize $\alpha = \frac{1}{2L_{\theta,\theta}}$ which implies that $\theta_{k,t+1} \in \arg\min_{\theta\in\Theta}\left[(g_{k,t}^{(\theta)})^\top(\theta - \theta_{k,t}) + L_{\theta,\theta}\|\theta - \theta_{k,t}\|^2\right]$, (iv) uses Eq. (98) and the following inequality.

$$\|\theta_{k,t}^{(\delta)} - \theta_{k,t}\| \overset{(i)}{\leq} \ell_{\lambda^{-1}}\|\lambda_{\theta_{k,t}^{(\delta)},\xi_k} - \lambda_{\theta_{k,t},\xi_k}\| = \ell_{\lambda^{-1}}\delta\|\lambda_{\theta_{k,t}^*,\xi_k} - \lambda_{\theta_{k,t},\xi_k}\| \overset{(ii)}{\leq} \sqrt{2}\ell_{\lambda^{-1}}\delta,$$

where (i) uses Assumption 4 and (ii) uses Lemma 4. Rearranging Eq. (99) and taking conditional expectation, we obtain that

$$\mathbb{E}\left[f(\lambda_{\theta_{k,t+1},\xi_k}) - \min_{\theta\in\Theta} f(\lambda_{\theta,\xi_k})\big|\xi_k\right]$$

$$\leq (1-\delta)\mathbb{E}\left[f(\lambda_{\theta_{k,t},\xi_k}) - \min_{\theta\in\Theta} f(\lambda_{\theta,\xi_k})\big|\xi_k\right] + \frac{1}{L_{\theta,\theta}}\mathbb{E}\left[\|\nabla_\theta f(\lambda_{\theta_{k,t},\xi_k}) - g_{k,t}^{(\theta)}\|^2\big|\xi_k\right] + 4L_{\theta,\theta}\ell_{\lambda^{-1}}^2\delta^2$$

$$\overset{(i)}{\leq} (1-\delta)\mathbb{E}\left[f(\lambda_{\theta_{k,t},\xi_k}) - \min_{\theta\in\Theta} f(\lambda_{\theta,\xi_k})\big|\xi_k\right] + 4L_{\theta,\theta}\ell_{\lambda^{-1}}^2\delta^2 + \frac{E_1^{(\theta)}}{L_{\theta,\theta}},$$

where (i) uses Eq. (43) in Proposition 9 and $E_1^{(\theta)}$ defined in Eq. (95) with $j = 1$. Then the convergence rate (97) can be proved by iterating the above inequality as follows. $\qquad\square$

## N.2 Convergence Rate of $\mathbb{E}[\|\nabla\widetilde{\Phi}(\widetilde{\xi})\|^2]$ from the First Original Phase

**Lemma 8.** *Implement the first original phase of Algorithm 1 with stepsizes $\alpha = \frac{1}{2L_{\theta,\theta}}$ and $\beta = \frac{1}{2L_{\xi,\xi}\sqrt{K}}$. The inner projected stochastic gradient descent step (14) is implemented up to precision $\epsilon_0 > 0$ as follows.*

$$\mathbb{E}\left[f(\lambda_{\theta_k,\xi_k}) - \min_{\theta\in\Theta} f(\lambda_{\theta,\xi_k})\big|\xi_k\right] \leq \epsilon_0. \tag{100}$$

*Then, the output $\widetilde{\xi}$ of the first original phase has the following convergence rate.*

$$\mathbb{E}\left[\|\nabla\widetilde{\Phi}(\widetilde{\xi})\|^2\right] \leq \frac{8f^* - 8\mathbb{E}\left[\Phi(\xi_0)\right]}{K\beta} + 10L_{\xi,\xi}\beta\ell_\xi^2 + 20L_{\xi,\xi}\epsilon_0 + 20E_2^{(\xi)}, \tag{101}$$

*where $E_2^{(\xi)}$ is defined in Eq. (96) with $j = 2$.*

*Proof.* For any fixed $\xi \in \Xi$, define the optimal policy parameter $\theta^*(\xi)$ and the optimal utility value $\Phi(\xi)$ as follows.

$$\theta^*(\xi) :\in \underset{\theta \in \Theta}{\arg\min}\, f(\lambda_{\theta,\xi}) \tag{102}$$

$$\Phi(\xi) := \min_{\theta \in \Theta} f(\lambda_{\theta,\xi}) = f(\lambda_{\theta^*(\xi),\xi}). \tag{103}$$

Since $f(\lambda_{\theta,\cdot})$ is $L_{\xi,\xi}$-smooth for any $\theta \in \Theta$ based on Proposition 3, for any $(\theta, \xi) \in \Theta \times \Xi$, $f(\lambda_{\theta,\xi'}) - \frac{L_{\xi,\xi}}{2}\|\xi' - \xi\|^2$ is a concave function of $\xi'$. As a result, $\Phi(\xi') - L_{\xi,\xi}\|\xi' - \xi\|^2$ is a $L_{\xi,\xi}$-strongly concave function of $\xi'$ and thus it has the following unique maximizer.

$$\xi^*(\xi) := \underset{\xi' \in \Xi}{\arg\max}\, \big[\Phi(\xi') - L_{\xi,\xi}\|\xi' - \xi\|^2\big] \tag{104}$$

Accordingly, we define the following Moreau envelope function (repeat Eq. (18)).

$$\widetilde{\Phi}(\xi) := \max_{\xi' \in \Xi} \big[\Phi(\xi') - L_{\xi,\xi}\|\xi' - \xi\|^2\big] = \Phi[\xi^*(\xi)] - L_{\xi,\xi}\|\xi^*(\xi) - \xi\|^2. \tag{105}$$

Based on Lemma 3.6 of [31], $\widetilde{\Phi}$ is $L_{\xi,\xi}$-smooth with

$$\nabla\widetilde{\Phi}(\xi) = 2L_{\xi,\xi}[\xi^*(\xi) - \xi] \tag{106}$$

Similar to Lemma D.3 of [31], we obtain the following ascent property of the above envelope function $\widetilde{\Phi}$ for any $k = 0, 1, \ldots, K - 1$.

$$\mathbb{E}\big[\widetilde{\Phi}(\xi_{k+1})\big|\xi_k\big]$$

$$\overset{(i)}{\geq} \mathbb{E}\big(\Phi[\xi^*(\xi_k)] - L_{\xi,\xi}\big\|\xi^*(\xi_k) - \mathrm{proj}_\Xi\big[\xi_k + \beta g_k^{(\xi)}\big]\big\|^2\big|\xi_k\big)$$

$$\overset{(ii)}{\geq} \mathbb{E}\big(\Phi[\xi^*(\xi_k)] - L_{\xi,\xi}(1 + \tau_k)\big\|\xi^*(\xi_k) - \mathrm{proj}_\Xi\big[\xi_k + \beta\nabla_\xi f(\lambda_{\theta_k,\xi_k})\big]\big\|^2$$
$$\quad - L_{\xi,\xi}(1 + \tau_k^{-1})\big\|\mathrm{proj}_\Xi\big[\xi_k + \beta\nabla_\xi f(\lambda_{\theta_k,\xi_k})\big] - \mathrm{proj}_\Xi\big[\xi_k + \beta g_k^{(\xi)}\big]\big\|^2\big|\xi_k\big)$$

$$\overset{(iii)}{\geq} \mathbb{E}\big(\Phi[\xi^*(\xi_k)] - L_{\xi,\xi}(1 + \tau_k)\big\|\xi^*(\xi_k) - \xi_k\big\|^2 - L_{\xi,\xi}\beta^2(1 + \tau_k)\big\|\nabla_\xi f(\lambda_{\theta_k,\xi_k})\big\|^2$$
$$\quad + 2L_{\xi,\xi}\beta(1 + \tau_k)\big\langle\xi^*(\xi_k) - \xi_k, \nabla_\xi f(\lambda_{\theta_k,\xi_k})\big\rangle$$
$$\quad - L_{\xi,\xi}\beta^2(1 + \tau_k^{-1})\big\|g_k^{(\xi)} - \nabla_\xi f(\lambda_{\theta_k,\xi_k})\big\|^2\big|\xi_k\big)$$

$$\overset{(iv)}{\geq} \Phi[\xi^*(\xi_k)] - L_{\xi,\xi}(1 + \tau_k)\big\|\xi^*(\xi_k) - \xi_k\big\|^2 - L_{\xi,\xi}\beta^2(1 + \tau_k)\ell_\xi^2$$
$$\quad + 2L_{\xi,\xi}\beta(1 + \tau_k)\mathbb{E}\Big[f(\lambda_{\theta_k,\xi^*(\xi_k)}) - f(\lambda_{\theta_k,\xi_k}) - \frac{L_{\xi,\xi}}{2}\|\xi^*(\xi_k) - \xi_k\|^2\Big|\xi_k\Big]$$
$$\quad - L_{\xi,\xi}\beta^2(1 + \tau_k^{-1})E_2^{(\xi)}$$

$$\overset{(v)}{\geq} \Phi[\xi^*(\xi_k)] - L_{\xi,\xi}(1 + \tau_k)\big\|\xi^*(\xi_k) - \xi_k\big\|^2 - L_{\xi,\xi}\beta^2(1 + \tau_k)\ell_\xi^2$$
$$\quad + 2L_{\xi,\xi}\beta(1 + \tau_k)\Big[\Phi[\xi^*(\xi_k)] - \Phi(\xi_k) - \epsilon_0 - \frac{L_{\xi,\xi}}{2}\|\xi^*(\xi_k) - \xi_k\|^2\Big] - L_{\xi,\xi}\beta^2(1 + \tau_k^{-1})E_2^{(\xi)}$$

$$\overset{(vi)}{\geq} \widetilde{\Phi}(\xi_k) - L_{\xi,\xi}\tau_k\big\|\xi^*(\xi_k) - \xi_k\big\|^2 - L_{\xi,\xi}\beta^2(1 + \tau_k)\ell_\xi^2$$
$$\quad + 2L_{\xi,\xi}\beta(1 + \tau_k)\Big[\frac{L_{\xi,\xi}}{2}\|\xi^*(\xi_k) - \xi_k\|^2 - \epsilon_0\Big] - L_{\xi,\xi}\beta^2(1 + \tau_k^{-1})E_2^{(\xi)}$$

$$\overset{(vii)}{\geq} \widetilde{\Phi}(\xi_k) + \frac{1}{4}\Big(\beta(1 + \tau_k) - \frac{\tau_k}{L_{\xi,\xi}}\Big)\big\|\nabla\widetilde{\Phi}(\xi_k)\big\|^2 - L_{\xi,\xi}\beta^2(1 + \tau_k)\ell_\xi^2$$
$$\quad - 2L_{\xi,\xi}\beta(1 + \tau_k)\epsilon_0 - L_{\xi,\xi}\beta^2(1 + \tau_k^{-1})E_2^{(\xi)}, \tag{107}$$

where (i) uses Eqs. (15) and (105), (ii) holds for any $\tau_k > 0$ whose value is to be determined later, (iii) uses contraction property of projection and $\xi^*(\xi_k) \in \Xi$, (iv) uses Propositions 3-9 and the error

term $E_2^{(\xi)}$ defined in Eq. (96), (v) uses Eqs. (100) and (103), (vi) uses Eqs. (104)-(105) which imply that $\widetilde{\Phi}(\xi_k) = \Phi[\xi^*(\xi_k)] - L_{\xi,\xi}\|\xi^*(\xi_k) - \xi_k\|^2 \geq \Phi(\xi_k)$, (vii) uses Eq. (106). Taking unconditional expectation of Eq. (107) and telescoping it over $k = 0, 1, \ldots, K-1$ with $\beta = \frac{1}{2L_{\xi,\xi}\sqrt{K}} \in \left[0, \frac{1}{2L_{\xi,\xi}}\right]$, $\tau_k \equiv \frac{\beta L_{\xi,\xi}}{2} \leq \frac{1}{4}$, we obtain that

$$
\frac{\beta}{8} \sum_{k=0}^{K-1} \mathbb{E}\big[\|\nabla\widetilde{\Phi}(\xi_k)\|^2\big]
$$

$$
\leq \mathbb{E}\big[\widetilde{\Phi}(\xi_K) - \widetilde{\Phi}(\xi_0)\big] + \frac{5K}{4}L_{\xi,\xi}\beta^2\ell_\xi^2 + \frac{5K}{2}L_{\xi,\xi}\beta\epsilon_0 + KL_{\xi,\xi}\beta^2\Big(1 + \frac{2}{\beta L_{\xi,\xi}}\Big)E_2^{(\xi)}
$$

$$
\overset{(i)}{\leq} f^* - \mathbb{E}\big[\Phi(\xi_0)\big] + \frac{5K}{4}L_{\xi,\xi}\beta^2\ell_\xi^2 + \frac{5K}{2}L_{\xi,\xi}\beta\epsilon_0 + \frac{5K\beta E_2^{(\xi)}}{2},
$$

where (i) uses the following range of $\widetilde{\Phi}(\xi)$ (defined in Eq. (105)) that holds for any $\xi \in \Xi$.

$$
\widetilde{\Phi}(\xi) \leq \max_{\xi'\in\Xi}\Phi(\xi') = \max_{\xi'\in\Xi}\min_{\theta\in\Theta} f(\lambda_{\theta,\xi'}) \leq \min_{\theta\in\Theta}\max_{\xi'\in\Xi} f(\lambda_{\theta,\xi'}) = f^* \tag{108}
$$

$$
\widetilde{\Phi}(\xi) \geq \Phi(\xi) \tag{109}
$$

As a result,

$$
\mathbb{E}\big[\|\nabla\widetilde{\Phi}(\widetilde{\xi})\|^2\big] = \frac{1}{K}\sum_{k=0}^{K-1}\mathbb{E}\big[\|\nabla\widetilde{\Phi}(\xi_k)\|^2\big]
$$

$$
\leq \frac{8f^* - 8\mathbb{E}\big[\Phi(\xi_0)\big]}{K\beta} + 10L_{\xi,\xi}\beta\ell_\xi^2 + 20L_{\xi,\xi}\epsilon_0 + 20E_2^{(\xi)}.
$$

### N.3  Convergence of the Inner Update Step (16) of the Second Corrected Phase

Next we focus on the second corrected phase which aims to solve the following minimax optimization problem (repeat Eq. (19)).

$$
\min_{\theta\in\Theta}\max_{\xi\in\Xi}\widetilde{f}(\theta,\xi) := f(\lambda_{\theta,\xi}) - L_{\xi,\xi}\|\xi - \widetilde{\xi}\|^2, \tag{110}
$$

where $\widetilde{\xi}$ is obtained from $\{\xi_k\}_{k=0}^{K-1}$ uniformly at random in the first original phase. Based on Proposition 3, it can be easily verified that $\widetilde{f}$ has the following smoothness properties and $\widetilde{f}(\theta, \cdot)$ is $L_{\xi,\xi}$-strongly concave.

$$
\|\nabla_\theta\widetilde{f}(\theta',\xi') - \nabla_\theta\widetilde{f}(\theta,\xi)\| \leq L_{\theta,\theta}\|\theta' - \theta\| + L_{\theta,\xi}\|\xi' - \xi\|, \tag{111}
$$

$$
\|\nabla_\xi\widetilde{f}(\theta',\xi') - \nabla_\xi\widetilde{f}(\theta,\xi)\| \leq L_{\xi,\theta}\|\theta' - \theta\| + 3L_{\xi,\xi}\|\xi' - \xi\|, \tag{112}
$$

Next, we will see the convergence rate of the projected stochastic gradient ascent steps (16) to the following optimal variable, which is unique due to strong concavity of $\widetilde{f}(\theta_k, \cdot)$.

$$
\xi_k^* := \arg\max_{\xi\in\Xi}\widetilde{f}(\theta_k,\xi). \tag{113}
$$

The optimization progress of each step of Eq. (16) for $k = K, K+1, \ldots, K+K'-1$ can be bounded as follows.

$$
\mathbb{E}\big[\|\xi_{k,t+1} - \xi_k^*\|^2\big|\theta_k\big]
$$

$$
\overset{(i)}{\leq} \mathbb{E}\big[\|\xi_{k,t} + a\big(g_{k,t}^{(\xi)} - 2L_{\xi,\xi}(\xi_{k,t} - \widetilde{\xi})\big) - \xi_k^*\|^2\big|\theta_k\big]
$$

$$
\overset{(ii)}{\leq} (1+c_k)\mathbb{E}\big[\|\xi_{k,t} + a\nabla_\xi\widetilde{f}(\theta_k,\xi_{k,t}) - \xi_k^*\|^2\big|\theta_k\big]
$$

$$
+ (1+c_k^{-1})\mathbb{E}\big[\|a\big(g_{k,t}^{(\xi)} - 2L_{\xi,\xi}(\xi_{k,t} - \widetilde{\xi}) - \nabla_\xi\widetilde{f}(\theta_k,\xi_{k,t})\big)\|^2\big|\theta_k\big]
$$

$$
\overset{(iii)}{=} (1+c_k)\mathbb{E}\big[\|\xi_{k,t} - \xi_k^*\|^2 + 2a\langle\nabla_\xi\widetilde{f}(\theta_k,\xi_{k,t}) - \nabla_\xi\widetilde{f}(\theta_k,\xi_k^*), \xi_{k,t} - \xi_k^*\rangle
$$

$$+ a^2 \|\nabla_\xi \widetilde{f}(\theta_k, \xi_{k,t}) - \nabla_\xi \widetilde{f}(\theta_k, \xi_k^*)\|^2 \big| \theta_k \big]$$

$$+ a^2 (1 + c_k^{-1}) \mathbb{E}\big[\|g_{k,t}^{(\xi)} - \nabla_\xi f(\lambda_{\theta_k, \xi_{k,t}})\|^2 \big| \theta_k\big]$$

$$\overset{(iv)}{\leq} (1 + c_k)(1 - 2L_{\xi,\xi} a + 9L_{\xi,\xi}^2 a^2) \mathbb{E}\big[\|\xi_{k,t} - \xi_k^*\|^2 \big| \theta_k\big] + a^2(1 + c_k^{-1}) E_3^{(\xi)}$$

$$\overset{(v)}{=} \frac{16}{17} \mathbb{E}\big[\|\xi_{k,t} - \xi_k^*\|^2 \big| \theta_k\big] + \frac{2E_3^{(\xi)}}{9L_{\xi,\xi}^2}, \tag{114}$$

where (i) uses Eq. (16), $\xi_k^* \in \Xi$ and contraction property of projection, (ii) holds for any $c_k > 0$ whose value will be assigned later, (iii) uses $\nabla_\xi \widetilde{f}(\theta_k, \xi_k^*) = 0$ and the definition of $\widetilde{f}$ in Eq. (110), (iv) uses Proposition 9, the error term $E_3^{(\xi)}$ defined in Eq. (96) as well as the $3L_{\xi,\xi}$-smoothness and $L_{\xi,\xi}$-strongly concavity of $\widetilde{f}(\theta_k, \cdot)$ (see Eq. (112)), and (v) uses $a = \frac{1}{9L_{\xi,\xi}}$ and $c_k = \frac{1}{17}$. Iterating the unconditional expectation of Eq. (114) over $t = 0, 1, \ldots, T' - 1$, we obtain that

$$\mathbb{E}\big[\|\xi_k - \xi_k^*\|^2 \big| \theta_k\big] = \mathbb{E}\big[\|\xi_{k,T'} - \xi_k^*\|^2 \big| \theta_k\big]$$

$$\leq \Big(\frac{16}{17}\Big)^{T'} \mathbb{E}\big[\|\xi_{k,0} - \xi_k^*\|^2 \big| \theta_k\big] + \frac{17(2E_2^{(\xi)})}{9L_{\xi,\xi}^2} \leq \Big(\frac{16}{17}\Big)^{T'} D_\Xi^2 + \frac{34 E_3^{(\xi)}}{9L_{\xi,\xi}^2}. \tag{115}$$

where the second $\leq$ denotes $D_\Xi := \sup_{\xi, \xi' \in \Xi} \|\xi' - \xi\|$ as the diameter of the compact set $\Xi$.

## N.4 Convergence Rate of $\mathbb{E}[\|G_b^{(\theta)}(\theta_{\widetilde{k}}, \xi_{\widetilde{k}})\|^2]$

Since $\widetilde{f}(\theta, \cdot)$ is strongly concave, it has unique maximizer $\widetilde{\xi}^*(\theta)$ and the corresponding function value $\widetilde{\Psi}(\theta)$ defined as follows.

$$\widetilde{\xi}^*(\theta) := \arg\max_{\xi \in \Xi} \widetilde{f}(\theta, \xi) \tag{116}$$

$$\widetilde{\Psi}(\theta) := \max_{\xi \in \Xi} \widetilde{f}(\theta, \xi), \tag{117}$$

Furthermore, since $\widetilde{f}(\theta, \cdot)$ is $L_{\xi,\xi}$-strongly concave and $\widetilde{f}$ has the smoothness properties (111) and (112), we can easily obtain that $\widetilde{\xi}^*(\theta)$ is $(L_{\xi,\theta}/L_{\xi,\xi})$-Lipschitz and $\widetilde{\Psi}$ is $\widetilde{L} := L_{\theta,\theta} + L_{\theta,\xi} L_{\xi,\theta}/L_{\xi,\xi}$-smooth with the following gradient, following the proof of Lemma 4.3 in [31]. [2]

$$\nabla \widetilde{\Psi}(\theta) = \nabla_1 \widetilde{f}[\theta, \widetilde{\xi}^*(\theta)] = \nabla_1 f(\lambda_{\theta, \widetilde{\xi}^*(\theta)}). \tag{118}$$

Note that for any $k = K, \ldots, K + K' - 1$, the projected stochastic gradient ascent step (17) satisfies

$$\|g_k^{(\theta)}\| = \frac{1}{b} \|\theta_k - (\theta_k - b g_k^{(\theta)})\|$$

$$\overset{(i)}{\geq} \frac{1}{b} \|\mathrm{proj}_\Theta(\theta_k - b g_k^{(\theta)}) - (\theta_k - b g_k^{(\theta)})\|$$

$$\overset{(ii)}{=} \frac{1}{b} \|\theta_{k+1} - \theta_k + b g_k^{(\theta)}\|,$$

where (i) uses $\theta_k \in \Theta$ and the definition of projection and (ii) uses the stochastic gradient descent step (17). The above inequality implies that

$$(g_k^{(\theta)})^\top (\theta_{k+1} - \theta_k) \leq -\frac{1}{2b} \|\theta_{k+1} - \theta_k\|^2. \tag{119}$$

Then, we analyze the optimization progress of the potential function (117) along the projected stochastic gradient descent step (17) as follows.

$$\mathbb{E}\widetilde{\Psi}(\theta_{k+1})$$

$$\overset{(i)}{\leq} \mathbb{E}\Big[\widetilde{\Psi}(\theta_k) + \nabla \widetilde{\Psi}(\theta_k)^\top (\theta_{k+1} - \theta_k) + \frac{\widetilde{L}}{2} \|\theta_{k+1} - \theta_k\|^2\Big]$$

---

[2] $\nabla_1 \widetilde{f}[\theta, \widetilde{\xi}^*(\theta)]$ and $\nabla_1 f(\lambda_{\theta, \widetilde{\xi}^*(\theta)})$ denote gradients with respect to only the first input argument $\theta$.

$$\overset{(ii)}{=}\mathbb{E}\Big[\widetilde{\Psi}(\theta_k) + \big\langle \nabla_1 f(\lambda_{\theta_k,\widetilde{\xi}^*(\theta_k)}) - g_k^{(\theta)}, \theta_{k+1} - \theta_k \big\rangle + (g_k^{(\theta)})^\top(\theta_{k+1} - \theta_k)$$

$$+ \frac{\widetilde{L}}{2}\|\theta_{k+1} - \theta_k\|^2\Big]$$

$$\overset{(iii)}{\leq}\mathbb{E}\Big[\widetilde{\Psi}(\theta_k) + \frac{1}{2\widetilde{L}}\big\|g_k^{(\theta)} - \nabla_1 f(\lambda_{\theta_k,\widetilde{\xi}^*(\theta_k)})\big\|^2 + \frac{\widetilde{L}}{2}\|\theta_{k+1} - \theta_k\|^2 - \frac{1}{2b}\|\theta_{k+1} - \theta_k\|^2$$

$$+ \frac{\widetilde{L}}{2}\|\theta_{k+1} - \theta_k\|^2\Big]$$

$$\leq\mathbb{E}\Big[\widetilde{\Psi}(\theta_k) + \frac{1}{\widetilde{L}}\big\|g_k^{(\theta)} - \nabla_\theta f(\lambda_{\theta_k,\xi_k})\big\|^2 + \frac{1}{\widetilde{L}}\big\|\nabla_\theta f(\lambda_{\theta_k,\xi_k}) - \nabla_1 f(\lambda_{\theta_k,\widetilde{\xi}^*(\theta_k)})\big\|^2$$

$$- \Big(\frac{1}{2b} - \widetilde{L}\Big)\|\theta_{k+1} - \theta_k\|^2\Big]$$

$$\overset{(iv)}{\leq}\mathbb{E}\widetilde{\Psi}(\theta_k) + \frac{E_4^{(\theta)}}{\widetilde{L}} + \frac{L_{\theta,\xi}^2}{\widetilde{L}}\mathbb{E}\big\|\xi_k - \widetilde{\xi}^*(\theta_k))\big\|^2 - \frac{1}{4b}\mathbb{E}\|\theta_{k+1} - \theta_k\|^2$$

$$\overset{(v)}{\leq}\mathbb{E}\widetilde{\Psi}(\theta_k) + \frac{E_4^{(\theta)}}{\widetilde{L}} + \frac{D_\Xi^2 L_{\theta,\xi}^2}{\widetilde{L}}\Big(\frac{16}{17}\Big)^{T'} + \frac{34 L_{\theta,\xi}^2 E_3^{(\xi)}}{9\widetilde{L}L_{\xi,\xi}^2} - \frac{1}{4b}\mathbb{E}\|\theta_{k+1} - \theta_k\|^2, \qquad (120)$$

where (i) uses the $\widetilde{L} := L_{\theta,\theta} + L_{\theta,\xi}L_{\xi,\theta}/L_{\xi,\xi}$-smoothness of $\widetilde{\Psi}$, (ii) uses Eq. (118), (iii) uses Cauchy-Schwartz inequality and Eq. (119), (iv) uses $b = \frac{1}{4\widetilde{L}}$, Propositions 3-9 and the error term $E_4^{(\theta)}$ defined by Eq. (95), (v) uses Eq. (115). Denote $G_b^{(\theta)}(\theta_k,\xi_k) = \frac{1}{b}\big[\theta_k - \mathrm{proj}_\Theta[\theta_k - b\nabla_\theta f(\lambda_{\theta_k,\xi_k})]\big]$ for $k = K, \dots, K + K' - 1$. Its norm can be bounded as follows.

$$\mathbb{E}\|G_b^{(\theta)}(\theta_k,\xi_k)\|^2$$

$$=\frac{1}{b^2}\mathbb{E}\|\theta_{k+1} - \theta_k + bG_b^{(\theta)}(\theta_k,\xi_k) - (\theta_{k+1} - \theta_k)\|^2$$

$$\leq\frac{2}{b^2}\mathbb{E}\|\theta_{k+1} - \theta_k + bG_b^{(\theta)}(\theta_k,\xi_k)\|^2 + \frac{2}{b^2}\mathbb{E}\| - (\theta_{k+1} - \theta_k)\|^2$$

$$\overset{(i)}{\leq}\frac{2}{b^2}\mathbb{E}\|\mathrm{proj}_\Theta[\theta_k - bg_k^{(\theta)}] - \mathrm{proj}_\Theta[\theta_k - b\nabla_\theta f(\lambda_{\theta_k,\xi_k})]\|^2$$

$$+ \frac{8}{b}\Big[\mathbb{E}[\widetilde{\Psi}(\theta_k) - \widetilde{\Psi}(\theta_{k+1})] + \frac{E_4^{(\theta)}}{\widetilde{L}} + \frac{D_\Xi^2 L_{\theta,\xi}^2}{\widetilde{L}}\Big(\frac{16}{17}\Big)^{T'} + \frac{34 L_{\theta,\xi}^2 E_3^{(\xi)}}{9\widetilde{L}L_{\xi,\xi}^2}\Big]$$

$$\overset{(ii)}{\leq}2\mathbb{E}\big\|g_k^{(\theta)} - \nabla_\theta f(\lambda_{\theta_k,\xi_k})\big\|^2$$

$$+ \frac{8}{b}\Big[\mathbb{E}[\widetilde{\Psi}(\theta_k) - \widetilde{\Psi}(\theta_{k+1})] + \frac{E_4^{(\theta)}}{\widetilde{L}} + \frac{D_\Xi^2 L_{\theta,\xi}^2}{\widetilde{L}}\Big(\frac{16}{17}\Big)^{T'} + \frac{34 L_{\theta,\xi}^2 E_3^{(\xi)}}{9\widetilde{L}L_{\xi,\xi}^2}\Big]$$

$$\overset{(iii)}{\leq}2E_4^{(\theta)} + \frac{8E_4^{(\theta)}}{b\widetilde{L}} + \frac{8}{b}\mathbb{E}[\widetilde{\Psi}(\theta_k) - \widetilde{\Psi}(\theta_{k+1})] + \frac{8D_\Xi^2 L_{\theta,\xi}^2}{b\widetilde{L}}\Big(\frac{16}{17}\Big)^{T'} + \frac{272 L_{\theta,\xi}^2 E_3^{(\xi)}}{9b\widetilde{L}L_{\xi,\xi}^2}, \qquad (121)$$

where (i) uses Eqs. (17) and (120) as well as $G_b^{(\theta)}(\theta_k,\xi_k) = \frac{1}{b}\big[\theta_k - \mathrm{proj}_\Theta[\theta_k - b\nabla_\theta f(\lambda_{\theta_k,\xi_k})]\big]$, (ii) uses Eq. (110), and (iii) uses Proposition 9 and the error term $E_4^{(\theta)}$ defined by Eq. (95). By rearranging the above inequality and averaging it over $k = K, K+1, \dots, K+K'-1$, we obtain the convergence rate of $\mathbb{E}[\|G_b^{(\theta)}(\theta_{\widetilde{k}},\xi_{\widetilde{k}})\|^2]$ as follows.

$$\mathbb{E}[\|G_b^{(\theta)}(\theta_{\widetilde{k}},\xi_{\widetilde{k}})\|^2] = \frac{1}{K'}\sum_{k=K}^{K+K'-1}\mathbb{E}\|G_b^{(\theta)}(\theta_k,\xi_k)\|^2$$

$$\leq 2E_4^{(\theta)} + \frac{8E_4^{(\theta)}}{b\widetilde{L}} + \frac{8}{bK'}\mathbb{E}[\widetilde{\Psi}(\theta_K) - \widetilde{\Psi}(\theta_{K+K'})] + \frac{8D_\Xi^2 L_{\theta,\xi}^2}{b\widetilde{L}}\Big(\frac{16}{17}\Big)^{T'} + \frac{272 L_{\theta,\xi}^2 E_3^{(\xi)}}{9b\widetilde{L}L_{\xi,\xi}^2}$$

$$\overset{(i)}{\leq} 34E_4^{(\theta)} + \frac{32\widetilde{L}}{K'}\mathbb{E}\big[\widetilde{f}(\theta_K, \widetilde{\xi}^*(\theta_K)) - \widetilde{f}(\theta_{K+K'}, \widetilde{\xi}^*(\theta_K))\big] + 32D_\Xi^2 L_{\theta,\xi}^2 \Big(\frac{16}{17}\Big)^{T'} + \frac{1088L_{\theta,\xi}^2 E_3^{(\xi)}}{9L_{\xi,\xi}^2}$$

$$\overset{(ii)}{=} 34E_4^{(\theta)} + \frac{32\widetilde{L}}{K'}\mathbb{E}\big[f(\lambda_{\theta_K, \widetilde{\xi}^*(\theta_K)}) - f(\lambda_{\theta_{K+K'}, \widetilde{\xi}^*(\theta_K)})\big] + 32D_\Xi^2 L_{\theta,\xi}^2 \Big(\frac{16}{17}\Big)^{T'} + \frac{1088L_{\theta,\xi}^2 E_3^{(\xi)}}{9L_{\xi,\xi}^2}$$

$$\overset{(iii)}{\leq} 34E_4^{(\theta)} + \frac{32\widetilde{L}}{K'}\big[\Gamma(\theta_K) - f^*\big] + 32D_\Xi^2 L_{\theta,\xi}^2 \Big(\frac{16}{17}\Big)^{T'} + \frac{1088L_{\theta,\xi}^2 E_3^{(\xi)}}{9L_{\xi,\xi}^2}, \tag{122}$$

where (i) uses Eqs. (116)-(117) and selects the stepsize $b = \frac{1}{4\widetilde{L}}$, (ii) uses Eq. (110), and (iii) uses $f^* := \min_{\theta\in\Theta}\max_{\xi\in\Xi} f(\lambda_{\theta,\xi})$ and $\Gamma(\theta) := \max_{\xi\in\Xi} f(\lambda_{\theta,\xi})$.

## N.5 Convergence Rate of $\mathbb{E}[\|G_a^{(\xi)}(\theta_{\widetilde{k}}, \xi_{\widetilde{k}})\|^2]$

Denote $\psi(\xi) := \min_{\theta\in\Theta}\widetilde{f}(\theta,\xi) = \Phi(\xi) - L_{\xi,\xi}\|\xi - \widetilde{\xi}\|^2$. Then, on one hand,

$$\psi\big[\xi^*(\widetilde{\xi})\big] - \psi(\xi_k) \overset{(i)}{=} \max_{\xi\in\Xi}\psi(\xi) - \psi(\xi_k)$$

$$= \max_{\xi\in\Xi}\min_{\theta\in\Theta}\widetilde{f}(\theta,\xi) - \min_{\theta\in\Theta}\widetilde{f}(\theta,\xi_k)$$

$$\leq \max_{\xi\in\Xi}\widetilde{f}(\theta_k,\xi) - \min_{\theta\in\Theta}\widetilde{f}(\theta,\xi_k)$$

$$\overset{(ii)}{=} \widetilde{f}(\theta_k,\xi_k^*) - \min_{\theta\in\Theta}\widetilde{f}(\theta,\xi_k)$$

$$\overset{(iii)}{\leq} \widetilde{f}(\theta_k,\xi_k) + \frac{3L_{\xi,\xi}}{2}\|\xi_k - \xi_k^*\|^2 - \min_{\theta\in\Theta}\widetilde{f}(\theta,\xi_k), \tag{123}$$

where (i) uses Eq. (104), (ii) uses Eq. (113), (iii) uses $\nabla_\xi\widetilde{f}(\theta_k,\xi_k^*) = 0$ at the optimal variable $\xi_k^*$ defined by Eq. (113) and $3L_{\xi,\xi}$-smoothness of $\widetilde{f}(\theta,\cdot)$ implied by Eq. (112). On the other hand, since $\widetilde{f}(\theta,\cdot)$ is $L_{\xi,\xi}$-strongly concave, $\psi$ is $L_{\xi,\xi}$-strongly concave. Hence,

$$\psi(\xi_k) \leq \psi\big[\xi^*(\widetilde{\xi})\big] + \nabla\psi\big[\xi^*(\widetilde{\xi})\big]^\top\big[\xi^*(\widetilde{\xi}) - \xi_k\big] - \frac{L_{\xi,\xi}}{2}\big\|\xi^*(\widetilde{\xi}) - \xi_k\big\|^2$$

$$\overset{(i)}{=} \psi\big[\xi^*(\widetilde{\xi})\big] - \frac{L_{\xi,\xi}}{2}\big\|\xi^*(\widetilde{\xi}) - \xi_k\big\|^2. \tag{124}$$

where (i) uses $\nabla\psi\big[\xi^*(\widetilde{\xi})\big] = 0$ at the unique optimizer $\xi^*(\widetilde{\xi}) = \arg\max_{\xi\in\Xi}\psi(\xi)$ (see Eq. (104)). Then, we have

$$\|G_a^{(\xi)}(\theta_k,\xi_k)\|^2$$

$$= \frac{1}{a^2}\big\|\text{proj}_\Xi\big(\xi_k + a\nabla_\xi f(\lambda_{\theta_k,\xi_k})\big) - \xi_k\big\|^2$$

$$\overset{(i)}{\leq} \frac{2}{a^2}\big\|\big[\text{proj}_\Xi\big(\xi_k + a\nabla_\xi\widetilde{f}(\theta_k,\xi_k)\big) - \xi_k\big] - \big[\text{proj}_\Xi\big(\xi_k^* + a\nabla_\xi\widetilde{f}(\theta_k,\xi_k^*)\big) - \xi_k^*\big]\big\|^2$$

$$\quad + \frac{2}{a^2}\big\|\text{proj}_\Xi\big(\xi_k + a\nabla_\xi f(\lambda_{\theta_k,\xi_k})\big) - \text{proj}_\Xi\big(\xi_k + a\nabla_\xi\widetilde{f}(\theta_k,\xi_k)\big)\big\|^2$$

$$\leq \frac{4}{a^2}\big\|\text{proj}_\Xi\big(\xi_k + a\nabla_\xi\widetilde{f}(\theta_k,\xi_k)\big) - \text{proj}_\Xi\big(\xi_k^* + a\nabla_\xi\widetilde{f}(\theta_k,\xi_k^*)\big)\big\|^2 + \frac{4}{a^2}\|\xi_k^* - \xi_k\|^2$$

$$\quad + \frac{2}{a^2}\big\|\text{proj}_\Xi\big(\xi_k + a\nabla_\xi f(\lambda_{\theta_k,\xi_k})\big) - \text{proj}_\Xi\big(\xi_k + a\nabla_\xi\widetilde{f}(\theta_k,\xi_k)\big)\big\|^2$$

$$\leq \frac{4}{a^2}\big\|\big(\xi_k + a\nabla_\xi\widetilde{f}(\theta_k,\xi_k)\big) - \big(\xi_k^* + a\nabla_\xi\widetilde{f}(\theta_k,\xi_k^*)\big)\big\|^2 + \frac{4}{a^2}\|\xi_k^* - \xi_k\|^2$$

$$\quad + 2\big\|\nabla_\xi f(\lambda_{\theta_k,\xi_k}) - \nabla_\xi\widetilde{f}(\theta_k,\xi_k)\big\|^2$$

$$\overset{(ii)}{\leq} 8\big\|\nabla_\xi\widetilde{f}(\theta_k,\xi_k) - \nabla_\xi\widetilde{f}(\theta_k,\xi_k^*)\big\|^2 + \frac{8}{a^2}\|\xi_k^* - \xi_k\|^2 + \frac{4}{a^2}\|\xi_k^* - \xi_k\|^2 + 2\big\|2L_{\xi,\xi}(\xi_k - \widetilde{\xi})\big\|^2$$

$$\overset{(iii)}{\leq} \left(72L_{\xi,\xi}^2 + \frac{12}{a^2}\right)\|\xi_k - \xi_k^*\|^2 + 16L_{\xi,\xi}^2\left[\|\xi_k - \xi^*(\widetilde{\xi})\|^2 + \|\xi^*(\widetilde{\xi}) - \widetilde{\xi}\|^2\right]$$

$$\overset{(iv)}{\leq} \left(72L_{\xi,\xi}^2 + \frac{12}{a^2}\right)\|\xi_k - \xi_k^*\|^2 + 32L_{\xi,\xi}\left(\widetilde{f}(\theta_k, \xi_k) + \frac{3L_{\xi,\xi}}{2}\|\xi_k - \xi_k^*\|^2 - \min_{\theta\in\Theta}\widetilde{f}(\theta, \xi_k)\right) + 4\|\nabla\widetilde{\Phi}(\widetilde{\xi})\|^2$$

$$\overset{(v)}{=} \left(120L_{\xi,\xi}^2 + \frac{12}{a^2}\right)\|\xi_k - \xi_k^*\|^2 + 32L_{\xi,\xi}\left[f(\lambda_{\theta_k,\xi_k}) - \min_{\theta\in\Theta}f(\lambda_{\theta,\xi_k})\right] + 4\|\nabla\widetilde{\Phi}(\widetilde{\xi})\|^2, \tag{125}$$

where (i) uses $\mathrm{proj}_\Xi\left(\xi_k^* + a\nabla_\xi\widetilde{f}(\theta_k, \xi_k^*)\right) - \xi_k^* = 0$ based on Eq. (113), (ii) uses the definition of $\widetilde{f}$ given by Eq. (110), (iii) uses $3L_{\xi,\xi}$-smoothness of $\widetilde{f}(\theta_k, \cdot)$ based on Eq. (112), (iv) uses Eqs. (106), (123) and (124), (v) uses Eq. (110). Taking expectation of the above Eq. (125) and averaging it over $k = K, \ldots, K + K' - 1$, we obtain the convergence rate of $\mathbb{E}[\|G_a^{(\xi)}(\theta_{\widetilde{k}}, \xi_{\widetilde{k}})\|^2]$ as follows.

$$\mathbb{E}[\|G_a^{(\xi)}(\theta_{\widetilde{k}}, \xi_{\widetilde{k}})\|^2]$$

$$\leq \frac{1}{K'}\left(120L_{\xi,\xi}^2 + \frac{12}{a^2}\right)\sum_{k=K}^{K+K'-1}\mathbb{E}\|\xi_k - \xi_k^*\|^2 + 4\mathbb{E}\|\nabla\widetilde{\Phi}(\widetilde{\xi})\|^2$$

$$+ \frac{32L_{\xi,\xi}}{K'}\sum_{k=K}^{K+K'-1}\mathbb{E}\left[f(\lambda_{\theta_k,\xi_k}) - \min_{\theta\in\Theta}f(\lambda_{\theta,\xi_k})\right]$$

$$\overset{(i)}{\leq} \left(120L_{\xi,\xi}^2 + \frac{12}{a^2}\right)\left[\left(\frac{16}{17}\right)^{T'}D_\Xi^2 + \frac{34E_2^{(\xi)}}{9L_{\xi,\xi}^2}\right]$$

$$+ 4\left(\frac{8f^* - 8\mathbb{E}\left[\Phi(\xi_0)\right]}{K\beta} + 10L_{\xi,\xi}\beta\ell_\xi^2 + 20L_{\xi,\xi}\epsilon_0 + 20E_2^{(\xi)}\right)$$

$$+ 32L_{\xi,\xi}\left[\sqrt{2}\ell_{\lambda^{-1}}(bL_{\theta,\theta} + 1) + b\ell_\theta\right]\mathbb{E}\left[\|G_b^{(\theta)}(\theta, \xi)\|\right]$$

$$\overset{(ii)}{\leq} \left(120L_{\xi,\xi}^2 + \frac{12}{a^2}\right)\left[\left(\frac{16}{17}\right)^{T'}D_\Xi^2 + \frac{34E_2^{(\xi)}}{9L_{\xi,\xi}^2}\right] + \frac{32f^* - 32\mathbb{E}\left[\Phi(\xi_0)\right]}{K\beta} + 40L_{\xi,\xi}\beta\ell_\xi^2$$

$$+ 80L_{\xi,\xi}\epsilon_0 + 80E_2^{(\xi)} + 32L_{\xi,\xi}\left[\sqrt{2}\ell_{\lambda^{-1}}(bL_{\theta,\theta} + 1) + b\ell_\theta\right]$$

$$\left[34E_4^{(\theta)} + \frac{32\widetilde{L}}{K'}\left[\Gamma(\theta_K) - f^*\right] + 32D_\Xi^2 L_{\theta,\xi}^2\left(\frac{16}{17}\right)^{T'} + \frac{1088L_{\theta,\xi}^2 E_3^{(\xi)}}{9L_{\xi,\xi}^2}\right]^{1/2} \tag{126}$$

where (i) uses Eqs. (20), (101) and (115), (ii) uses Eq. (122), (iii) uses $\ell_\theta, \ell_\xi = \mathcal{O}[(1-\gamma)^{-2}]$, $L_{\theta,\xi}, L_{\xi,\xi}, \widetilde{L} = \mathcal{O}[(1-\gamma)^{-3}]$ based on Proposition 3 and selects stepsizes $a = \frac{1}{9L_{\xi,\xi}} = \mathcal{O}[(1-\gamma)^3]$, $b = \frac{1}{4\widetilde{L}} = \mathcal{O}[(1-\gamma)^3]$, $\beta = \frac{1}{2L_{\xi,\xi}\sqrt{K}} = \mathcal{O}[K^{-1/2}(1-\gamma)^3]$.

## N.6  Substituting Hyperparameters

Denote $\delta = \min\left[\bar{\delta}, \frac{\epsilon^2}{5760L_{\xi,\xi}L_{\theta,\theta}\ell_{\lambda^{-1}}^2}, \frac{1}{2}\right] = \mathcal{O}[(1-\gamma)^6\epsilon^2]$. Then we select the following hyperparameter values.

$$K = 36\epsilon^{-4}\left\{64L_{\xi,\xi}\left[f^* - \mathbb{E}[\Phi(\xi_0)]\right] + 20\ell_\xi^2\right\}^2 = \mathcal{O}[(1-\gamma)^{-8}\epsilon^{-4}] \tag{127}$$

$$T = \frac{1}{3\delta}\log\left\{1440L_{\xi,\xi}\epsilon^{-2}\mathbb{E}\left[\Gamma(\theta_0) - \min_{\theta\in\Theta, \xi\in\Xi}f(\lambda_{\theta,\xi})\right]\right\} = \mathcal{O}\left[\frac{\log[(1-\gamma)^{-1}\epsilon^{-1}]}{(1-\gamma)^6\epsilon^2}\right] \tag{128}$$

$$K' = \frac{294912\widetilde{L}L_{\xi,\xi}^2}{\widetilde{L}^2\epsilon^4}\left[\Gamma(\theta_K) - f^*\right]\left[\sqrt{2}\ell_{\lambda^{-1}}\left(L_{\theta,\theta} + 4\widetilde{L}\right) + \ell_\theta\right]^2 = \mathcal{O}[(1-\gamma)^{-9}\epsilon^{-4}] \tag{129}$$

$$T' = 33\log\left\{544L_{\xi,\xi}D_\Xi L_{\theta,\xi}\widetilde{L}^{-1}\epsilon^{-2}\left[\sqrt{2}\ell_{\lambda^{-1}}\left(L_{\theta,\theta} + 4\widetilde{L}\right) + \ell_\theta\right]\right\} = \mathcal{O}\left(\log[(1-\gamma)^{-1}\epsilon^{-1}]\right) \tag{130}$$

$$\alpha = \frac{1}{2L_{\theta,\theta}} \tag{131}$$

$$\beta = \frac{1}{2L_{\xi,\xi}\sqrt{K}} \tag{132}$$

$$a = \frac{1}{9L_{\xi,\xi}}, \tag{133}$$

$$b = \frac{1}{4\widetilde{L}}, \tag{134}$$

$$m_\theta^{(1)} = \frac{17280 L_{\xi,\xi} \ell_{\pi_\theta}^2 \ell_\lambda^2}{L_{\theta,\theta} \delta \epsilon^2 (1-\gamma)^4} = \mathcal{O}[(1-\gamma)^{-10} \epsilon^{-4}], \tag{135}$$

$$m_\lambda^{(1)} = \frac{17280 L_{\xi,\xi} \ell_{\pi_\theta}^2 L_\lambda^2 |\mathcal{S}||\mathcal{A}|}{L_{\theta,\theta} \delta \epsilon^2 (1-\gamma)^4} = \mathcal{O}[(1-\gamma)^{-10} \epsilon^{-4}], \tag{136}$$

$$H_\lambda^{(1)} = \frac{1}{2 \log(\gamma^{-1})} \log \left[ \frac{17280 L_{\xi,\xi} \ell_{\pi_\theta}^2 L_\lambda^2 |\mathcal{S}||\mathcal{A}|}{L_{\theta,\theta} \delta \epsilon^2 (1-\gamma)^4} \right] = \mathcal{O}\left[ \frac{\log[(1-\gamma)^{-1} \epsilon^{-1}]}{1-\gamma} \right] \tag{137}$$

$$H_\theta^{(1)} = \frac{4}{1-\gamma} \log \left( \frac{2}{1-\gamma} \right) + \frac{1}{\log(\gamma^{-1})} \log \left[ \frac{51840 L_{\xi,\xi} \ell_{\pi_\theta}^2 \ell_\lambda^2}{L_{\theta,\theta} \delta \epsilon^2 (1-\gamma)^4} \right] = \mathcal{O}\left[ \frac{\log[(1-\gamma)^{-1} \epsilon^{-1}]}{1-\gamma} \right] \tag{138}$$

$$m_\xi^{(2)} = \frac{148512 \ell_\lambda^2 \ell_{p_\xi}^2}{\epsilon^2 (1-\gamma)^4} = \mathcal{O}[(1-\gamma)^{-4} \epsilon^{-2}], \tag{139}$$

$$m_\lambda^{(2)} = \frac{148512 L_\lambda^2 \ell_{p_\xi}^2 |\mathcal{S}||\mathcal{A}|}{\epsilon^2 (1-\gamma)^4} = \mathcal{O}[(1-\gamma)^{-4} \epsilon^{-2}], \tag{140}$$

$$H_\lambda^{(2)} = \frac{1}{2 \log(\gamma^{-1})} \left[ \frac{148512 L_\lambda^2 \ell_{p_\xi}^2 |\mathcal{S}||\mathcal{A}|}{\epsilon^2 (1-\gamma)^4} \right] = \mathcal{O}\left[ \frac{\log[(1-\gamma)^{-1} \epsilon^{-1}]}{1-\gamma} \right] \tag{141}$$

$$H_\xi^{(2)} = \frac{4}{1-\gamma} \log \left( \frac{2}{1-\gamma} \right) + \frac{1}{\log(\gamma^{-1})} \log \left[ \frac{297024 \ell_\lambda^2 \ell_{p_\xi}^2}{\epsilon^2 (1-\gamma)^4} \right] = \mathcal{O}\left[ \frac{\log[(1-\gamma)^{-1} \epsilon^{-1}]}{1-\gamma} \right] \tag{142}$$

$$m_\xi^{(3)} = \frac{13369344 L_{\theta,\xi}^2 \ell_{p_\xi}^2 \ell_\lambda^2 [\sqrt{2} \ell_{\lambda^{-1}}(L_{\theta,\theta} + 4\widetilde{L}) + \ell_\theta]^2}{\widetilde{L}^2 \epsilon^4 (1-\gamma)^4} = \mathcal{O}[(1-\gamma)^{-10} \epsilon^{-4}], \tag{143}$$

$$m_\lambda^{(3)} = \frac{13369344 L_{\theta,\xi}^2 \ell_{p_\xi}^2 L_\lambda^2 |\mathcal{S}||\mathcal{A}| [\sqrt{2} \ell_{\lambda^{-1}}(L_{\theta,\theta} + 4\widetilde{L}) + \ell_\theta]^2}{\widetilde{L}^2 \epsilon^4 (1-\gamma)^4} = \mathcal{O}[(1-\gamma)^{-10} \epsilon^{-4}], \tag{144}$$

$$H_\lambda^{(3)} = \frac{1}{2 \log(\gamma^{-1})} \left[ \frac{13369344 L_{\theta,\xi}^2 \ell_{p_\xi}^2 L_\lambda^2 |\mathcal{S}||\mathcal{A}| [\sqrt{2} \ell_{\lambda^{-1}}(L_{\theta,\theta} + 4\widetilde{L}) + \ell_\theta]^2}{\widetilde{L}^2 \epsilon^4 (1-\gamma)^4} \right]$$
$$= \mathcal{O}\left[ \frac{\log[(1-\gamma)^{-1} \epsilon^{-1}]}{1-\gamma} \right] \tag{145}$$

$$H_\xi^{(3)} = \frac{4}{1-\gamma} \log \left( \frac{2}{1-\gamma} \right) + \frac{1}{\log(\gamma^{-1})} \log \left[ \frac{26738688 L_{\theta,\xi}^2 \ell_{p_\xi}^2 \ell_\lambda^2 [\sqrt{2} \ell_{\lambda^{-1}}(L_{\theta,\theta} + 4\widetilde{L}) + \ell_\theta]^2}{\widetilde{L}^2 \epsilon^4 (1-\gamma)^4} \right]$$
$$= \mathcal{O}\left[ \frac{\log[(1-\gamma)^{-1} \epsilon^{-1}]}{1-\gamma} \right] \tag{146}$$

$$m_\theta^{(4)} = \frac{3760128 L_{\xi,\xi}^2 \ell_{\pi_\theta}^2 \ell_\lambda^2 [\sqrt{2} \ell_{\lambda^{-1}}(L_{\theta,\theta} + 4\widetilde{L}) + \ell_\theta]^2}{\widetilde{L}^2 \epsilon^4 (1-\gamma)^4} = \mathcal{O}[(1-\gamma)^{-10} \epsilon^{-4}], \tag{147}$$

$$m_\lambda^{(4)} = \frac{3760128 L_{\xi,\xi}^2 \ell_{\pi_\theta}^2 L_\lambda^2 |\mathcal{S}||\mathcal{A}| [\sqrt{2} \ell_{\lambda^{-1}}(L_{\theta,\theta} + 4\widetilde{L}) + \ell_\theta]^2}{\widetilde{L}^2 \epsilon^4 (1-\gamma)^4} = \mathcal{O}[(1-\gamma)^{-10} \epsilon^{-4}], \tag{148}$$

$$H_\lambda^{(4)} = \frac{1}{2 \log(\gamma^{-1})} \left[ \frac{3760128 L_{\xi,\xi}^2 \ell_{\pi_\theta}^2 L_\lambda^2 |\mathcal{S}||\mathcal{A}| [\sqrt{2} \ell_{\lambda^{-1}}(L_{\theta,\theta} + 4\widetilde{L}) + \ell_\theta]^2}{\widetilde{L}^2 \epsilon^4 (1-\gamma)^4} \right]$$
$$= \mathcal{O}\left[ \frac{\log[(1-\gamma)^{-1} \epsilon^{-1}]}{1-\gamma} \right] \tag{149}$$

$$H_\theta^{(4)} = \frac{4}{1-\gamma} \log \left( \frac{2}{1-\gamma} \right) + \frac{1}{\log(\gamma^{-1})} \log \left[ \frac{7520256 L_{\xi,\xi}^2 \ell_{\pi_\theta}^2 \ell_\lambda^2 [\sqrt{2} \ell_{\lambda^{-1}}(L_{\theta,\theta} + 4\widetilde{L}) + \ell_\theta]^2}{\widetilde{L}^2 \epsilon^4 (1-\gamma)^4} \right]$$

$$=\mathcal{O}\Big[\frac{\log[(1-\gamma)^{-1}\epsilon^{-1}]}{1-\gamma}\Big] \tag{150}$$

Substituting the above hyperparameter choices into Eqs. (95) and (96), we have

$$E_1^{(\theta)} := \frac{3\ell_{\pi_\theta}^2}{(1-\gamma)^4}\Big[L_\lambda^2|\mathcal{S}||\mathcal{A}|\Big(\frac{1}{m_\lambda^{(1)}}+\gamma^{2H_\lambda^{(1)}}\Big)+\frac{\ell_\lambda^2}{m_\theta^{(1)}}+\ell_\lambda^2[1+H_\theta^{(1)}(1-\gamma)]^2\gamma^{2H_\theta^{(1)}}\Big]$$

$$\leq \frac{L_{\theta,\theta}\delta\epsilon^2}{1440 L_{\xi,\xi}}, \tag{151}$$

$$E_2^{(\xi)} := \frac{3\ell_{p_\xi}^2}{(1-\gamma)^4}\Big[L_\lambda^2|\mathcal{S}||\mathcal{A}|\Big(\frac{1}{m_\lambda^{(2)}}+\gamma^{2H_\lambda^{(2)}}\Big)+\frac{\ell_\lambda^2}{m_\xi^{(2)}}+\ell_\lambda^2[1+H_\xi^{(2)}(1-\gamma)]^2\gamma^{2H_\xi^{(2)}}\Big]$$

$$\leq \frac{\epsilon^2}{12376}, \tag{152}$$

$$E_3^{(\xi)} := \frac{3\ell_{p_\xi}^2}{(1-\gamma)^4}\Big[L_\lambda^2|\mathcal{S}||\mathcal{A}|\Big(\frac{1}{m_\lambda^{(3)}}+\gamma^{2H_\lambda^{(3)}}\Big)+\frac{\ell_\lambda^2}{m_\xi^{(3)}}+\ell_\lambda^2[1+H_\xi^{(3)}(1-\gamma)]^2\gamma^{2H_\xi^{(3)}}\Big]$$

$$\leq \frac{\widetilde{L}^2\epsilon^4}{1114112 L_{\theta,\xi}^2\big[\sqrt{2}\ell_{\lambda^{-1}}\big(L_{\theta,\theta}+4\widetilde{L}\big)+\ell_\theta\big]^2}, \tag{153}$$

$$E_4^{(\theta)} := \frac{3\ell_{\pi_\theta}^2}{(1-\gamma)^4}\Big[L_\lambda^2|\mathcal{S}||\mathcal{A}|\Big(\frac{1}{m_\lambda^{(4)}}+\gamma^{2H_\lambda^{(4)}}\Big)+\frac{\ell_\lambda^2}{m_\theta^{(4)}}+\ell_\lambda^2[1+H_\theta^{(4)}(1-\gamma)]^2\gamma^{2H_\theta^{(4)}}\Big]$$

$$\leq \frac{\widetilde{L}^2\epsilon^4}{313344 L_{\xi,\xi}^2[\sqrt{2}\ell_{\lambda^{-1}}(L_{\theta,\theta}+4\widetilde{L})+\ell_\theta]^2}, \tag{154}$$

where we used $\big[1+H_\theta^{(k)}(1-\gamma)\big]^2\gamma^{H_\theta^{(k)}}\leq 2$ for $H_\theta^{(k)}\geq\frac{4}{1-\gamma}\log\big(\frac{2}{1-\gamma}\big)$ $(k=1,2,3,4)$.

Lemma 7 implies that for any fixed $\xi\in\Xi$, $\theta_{T'}$ obtained from the update rule (14) satisfies

$$\mathbb{E}\big[f(\lambda_{\theta_{T'},\xi})-\min_{\theta\in\Theta}f(\lambda_{\theta,\xi})\big]$$

$$\overset{(i)}{\leq}(1-\delta)^T\mathbb{E}\big[f(\lambda_{\theta_0,\xi_k})-\min_{\theta\in\Theta}f(\lambda_{\theta,\xi_k})\big]+4L_{\theta,\theta}\ell_{\lambda^{-1}}^2\delta+\frac{E_1^{(\theta)}}{\delta L_{\theta,\theta}}$$

$$\overset{(ii)}{\leq}\mathbb{E}\Big[\Gamma(\theta_0)-\min_{\theta\in\Theta,\xi\in\Xi}f(\lambda_{\theta,\xi})\Big]\exp\Big\{\log(1-\delta)\cdot\frac{1}{\delta}\log\Big[1440L_{\xi,\xi}\epsilon^{-2}\mathbb{E}\Big[\Gamma(\theta_0)-\min_{\theta\in\Theta,\xi\in\Xi}f(\lambda_{\theta,\xi})\Big]\Big]\Big\}$$

$$+\frac{\epsilon^2}{1440L_{\xi,\xi}}+\frac{\epsilon^2}{1440L_{\xi,\xi}}$$

$$\overset{(iii)}{\leq}\frac{\epsilon^2}{480L_{\xi,\xi}}.$$

where (i) uses Lemma 7, (ii) uses Eqs. (128), (151) and $\delta\leq\frac{\epsilon^2}{5760L_{\xi,\xi}L_{\theta,\theta}\ell_{\lambda^{-1}}^2}$, (iii) uses $\log(1-\delta)\leq-\delta$ for $\delta\in[0,1/2]$. Hence, the above inequality implies that $\epsilon_0$ defined by Lemma 8 satisfies $\epsilon_0\leq\frac{\epsilon^2}{480L_{\xi,\xi}}$.

As a result, we can prove that $\mathbb{E}\big[\|G_a^{(\xi)}(\theta_{\widetilde{k}},\xi_{\widetilde{k}})\|^2\big]\leq\epsilon^2$ and $\mathbb{E}\big[\|G_b^{(\theta)}(\theta_{\widetilde{k}},\xi_{\widetilde{k}})\|^2\big]\leq\epsilon^2$ by substituting $\epsilon_0\leq\frac{\epsilon^2}{480L_{\xi,\xi}}$ and Eqs. (127)-(154) into the convergence rates (122) and (126). The number of samples required by Algorithm 1 is

$$KT(m_\lambda^{(1)}H_\lambda^{(1)}+m_\theta^{(1)}H_\theta^{(1)})+K(m_\lambda^{(2)}H_\lambda^{(2)}+m_\xi^{(2)}H_\xi^{(2)})$$

$$+K'T'(m_\lambda^{(3)}H_\lambda^{(3)}+m_\xi^{(3)}H_\xi^{(3)})+K'(m_\lambda^{(4)}H_\lambda^{(4)}+m_\theta^{(4)}H_\theta^{(4)})$$

$$=\mathcal{O}[(1-\gamma)^{-8}\epsilon^{-4}]\mathcal{O}\Big[\frac{\log[(1-\gamma)^{-1}\epsilon^{-1}]}{(1-\gamma)^6\epsilon^2}\Big]\mathcal{O}[(1-\gamma)^{-10}\epsilon^{-4}]\mathcal{O}\Big[\frac{\log[(1-\gamma)^{-1}\epsilon^{-1}]}{1-\gamma}\Big]$$

$$+ \mathcal{O}[(1-\gamma)^{-8}\epsilon^{-4}]\mathcal{O}[(1-\gamma)^{-4}\epsilon^{-2}]\mathcal{O}\Big[\frac{\log[(1-\gamma)^{-1}\epsilon^{-1}]}{1-\gamma}\Big]$$

$$+ \mathcal{O}[(1-\gamma)^{-9}\epsilon^{-4}]\mathcal{O}\big(\log[(1-\gamma)^{-1}\epsilon^{-1}]\big)\mathcal{O}[(1-\gamma)^{-10}\epsilon^{-4}]\mathcal{O}\Big[\frac{\log[(1-\gamma)^{-1}\epsilon^{-1}]}{1-\gamma}\Big]$$

$$+ \mathcal{O}[(1-\gamma)^{-9}\epsilon^{-4}]\mathcal{O}[(1-\gamma)^{-10}\epsilon^{-4}]\mathcal{O}\Big[\frac{\log[(1-\gamma)^{-1}\epsilon^{-1}]}{1-\gamma}\Big]$$

$$= \mathcal{O}\Big[\frac{\log^2[(1-\gamma)^{-1}\epsilon^{-1}]}{(1-\gamma)^{25}\epsilon^{10}}\Big].$$

$\square$

## O   Proof of Theorem 3

Note that $\sigma_k = 2\beta_k\ell_\theta$, so by the definition of $\Xi_k := \{\xi \in V(\Xi) : f(\lambda_{\theta_k,\xi}) \geq \max_{\xi'\in V(\Xi)} f(\lambda_{\theta_k,\xi'}) - 2\beta_k\ell_\theta\}$ we have

$$f(\lambda_{\theta_k,\xi}) < \max_{\xi'\in V(\Xi)} f(\lambda_{\theta_k,\xi'}) - 2\beta_k\ell_\theta, \quad \forall\xi \in V(\Xi)/\Xi_k. \tag{155}$$

Hence,

$$\max_{\xi'\in\Xi} f(\lambda_{\theta_k,\xi'}) - f(\lambda_{\theta_{k+1},\xi}) \overset{(i)}{\geq} \max_{\xi'\in\Xi} f(\lambda_{\theta_k,\xi'}) - f(\lambda_{\theta_k,\xi}) - 2\ell_\theta\|\theta_{k+1} - \theta_k\|$$

$$\overset{(ii)}{>} 2\beta_k\ell_\theta - 2\ell_\theta\beta_k = 0, \quad \forall\xi \in V(\Xi)/\Xi_k \tag{156}$$

where (i) uses Eq. (10) in Proposition 3 which implies that $f(\lambda_{\cdot,\xi})$ is $\ell_\theta$-Lipschitz continuous, (ii) uses Eq. (155) and the update rule (22) with $\|d_k\| = 1$. Eqs. (155) and (156) respectively imply the following two equations.

$$\Gamma(\theta_k) = \max_{\xi'\in\Xi_k} f(\lambda_{\theta_k,\xi'}) \geq f(\lambda_{\theta_k,\xi_{k+1}^*}), \tag{157}$$

$$\Gamma(\theta_{k+1}) = \max_{\xi'\in\Xi_k} f(\lambda_{\theta_{k+1},\xi'}) = f(\lambda_{\theta_{k+1},\xi_{k+1}^*}), \tag{158}$$

where $\xi_{k+1}^* \in \arg\max_{\xi'\in\Xi_k} f(\lambda_{\theta_{k+1},\xi'})$.

Based on Proposition 7, there exists a unit descent direction $\widetilde{d}_k$ ($\|\widetilde{d}_k\| = 1$) such that

$$0 < f(\lambda_{\theta_k,\xi}) - \min_{\theta'\in\Theta}\Gamma(\theta') \leq f(\lambda_{\theta_k,\xi}) - f(\lambda_{\theta^*,\xi}) \leq \big[-\sqrt{2}\ell_{\lambda^{-1}}\nabla_\theta f(\lambda_{\theta_k,\xi})^\top\widetilde{d}_k\big]_+, \forall\xi \in \Xi_k \tag{159}$$

Then we have

$$\Gamma(\theta_k) - \min_{\theta'\in\Theta}\Gamma(\theta') \overset{(i)}{=} \max_{\xi'\in V(\Xi)} f(\lambda_{\theta_k,\xi'}) - \min_{\theta'\in\Theta}\Gamma(\theta')$$

$$\overset{(ii)}{\leq} \min_{\xi\in\Xi_k} f(\lambda_{\theta_k,\xi}) - \min_{\theta'\in\Theta}\Gamma(\theta') + 2\beta_k\ell_\theta$$

$$\overset{(iii)}{\leq} \min_{\xi\in\Xi_k} \big[-\sqrt{2}\ell_{\lambda^{-1}}\nabla_\theta f(\lambda_{\theta_k,\xi})^\top\widetilde{d}_k\big]_+ + 2\beta_k\ell_\theta$$

$$= \big[-\sqrt{2}\ell_{\lambda^{-1}}A_k(\widetilde{d}_k)\big]_+ + 2\beta_k\ell_\theta$$

$$\overset{(iv)}{\leq} \big[\sqrt{2}\ell_{\lambda^{-1}}[\epsilon_k - A_k(d_k')]\big]_+ + 2\beta_k\ell_\theta, \tag{160}$$

where (i) uses Proposition 6, (ii) uses $\Xi_k := \{\xi \in V(\Xi) : f(\lambda_{\theta_k,\xi}) \geq \max_{\xi'\in V(\Xi)} f(\lambda_{\theta_k,\xi'}) - 2\beta_k\ell_\theta\}$, (iii) uses Eq. (159), (iv) uses $A_k(\widetilde{d}_k) \geq \min_{d\in B_1} A_k(d) \geq A_k(d_k') - \epsilon_k$ based on line 6 of Algorithm 2.

## O.1 Analyze the $k$-th Iteration

**(Case 1):** If $A_k(d'_k) \geq 0$, then Eq. (160) implies that

$$\Gamma(\theta_k) - \min_{\theta' \in \Theta} \Gamma(\theta') \leq \sqrt{2}\ell_{\lambda^{-1}}\epsilon_k + 2\beta_k\ell_\theta. \tag{161}$$

Hence, by $\ell_\theta$-Lipschitz continuity of $\Gamma(\cdot) := \max_{\xi \in \Xi} f(\lambda_{\cdot,\xi})$ (based on Proposition 3), we have

$$\Gamma(\theta_{k+1}) - \min_{\theta' \in \Theta} \Gamma(\theta') \leq \Gamma(\theta_k) - \min_{\theta' \in \Theta} \Gamma(\theta') + \ell_\theta\|\theta_{k+1} - \theta_k\| \overset{(i)}{\leq} \sqrt{2}\ell_{\lambda^{-1}}\epsilon_k + 3\beta_k\ell_\theta, \tag{162}$$

where (i) uses Eq. (161) and the update rule (155) with $\|d_k\| = 1$.

**(Case 2):** If $A_k(d'_k) < 0$, then since $\|d'_k\| \leq 1$, $d_k = d'_k/\|d'_k\|$ satisfies $A_k(d_k) \leq A_k(d'_k) < 0$. Hence, Eq. (160) implies that

$$\Gamma(\theta_k) - \min_{\theta' \in \Theta} \Gamma(\theta') \leq \sqrt{2}\ell_{\lambda^{-1}}[\epsilon_k - A_k(d_k)] + 2\beta_k\ell_\theta. \tag{163}$$

As a result, we bound the one-step optimization progress of $\Gamma(\theta_k)$ as follows.

$$
\begin{aligned}
\Gamma(\theta_{k+1}) - \Gamma(\theta_k) &\overset{(i)}{\leq} f(\lambda_{\theta_{k+1}}, \xi^*_{k+1}) - f(\lambda_{\theta_k}, \xi^*_{k+1}) \\
&\overset{(ii)}{\leq} \nabla_\theta f(\lambda_{\theta_k}, \xi^*_{k+1})^\top(\theta_{k+1} - \theta_k) + \frac{L_{\theta,\theta}}{2}\|\theta_{k+1} - \theta_k\|^2 \\
&\overset{(iii)}{=} \beta_k \nabla_\theta f(\lambda_{\theta_k}, \xi^*_{k+1})^\top d_k + \frac{L_{\theta,\theta}}{2}\beta_k^2 \\
&\overset{(iv)}{\leq} \beta_k A_k(d_k) + \frac{L_{\theta,\theta}}{2}\beta_k^2 \\
&\overset{(v)}{\leq} \beta_k\left(\epsilon_k + \frac{L_{\theta,\theta}\beta_k}{2} + \frac{\sqrt{2}\beta_k\ell_\theta}{\ell_{\lambda^{-1}}}\right) - \frac{\beta_k}{\sqrt{2}\ell_{\lambda^{-1}}}\left[\Gamma(\theta_k) - \min_{\theta' \in \Theta}\Gamma(\theta')\right],
\end{aligned}
$$

where (i) uses Eqs. (157) and (158), (ii) uses $L_{\theta,\theta}$-smoothness of $f(\lambda_{\cdot,\xi})$ based on Proposition 3, (iii) uses the update rule (22) with $\|d_k\| = 1$, (iv) uses $\xi^*_{k+1} \in \Xi_k$ and the definition of $A_k$ in Eq. (23), (v) uses Eq. (163). Rearranging the above inequality yields that

$$
\begin{aligned}
\Gamma(\theta_{k+1}) &- \min_{\theta' \in \Theta}\Gamma(\theta') \\
&\leq \left(1 - \frac{\beta_k}{\sqrt{2}\ell_{\lambda^{-1}}}\right)\left[\Gamma(\theta_k) - \min_{\theta' \in \Theta}\Gamma(\theta')\right] + \beta_k\left(\epsilon_k + \frac{L_{\theta,\theta}\beta_k}{2} + \frac{\sqrt{2}\beta_k\ell_\theta}{\ell_{\lambda^{-1}}}\right) \\
&\overset{(i)}{=} \frac{k}{k+2}\left[\Gamma(\theta_k) - \min_{\theta' \in \Theta}\Gamma(\theta')\right] + \frac{2\sqrt{2}\ell_{\lambda^{-1}}\epsilon_k}{k+2} + \frac{8\ell_{\lambda^{-1}}^2}{(k+2)^2}\left(\frac{L_{\theta,\theta}}{2} + \frac{\sqrt{2}\ell_\theta}{\ell_{\lambda^{-1}}}\right),
\end{aligned} \tag{164}
$$

where (i) uses $\beta_k = \frac{2\sqrt{2}\ell_{\lambda^{-1}}}{k+2}$.

## O.2 Obtain the Convergence Rate (25)

**(Case 1):** If $A_k(d'_k) < 0$ for all $k = 0, 1, \ldots, K-1$, then we iterate Eq. (164) over $k = 0, 1, \ldots, K-1$ as follows.

$$
\begin{aligned}
\Gamma(\theta_K) &- \min_{\theta' \in \Theta}\Gamma(\theta') \\
&\leq \sum_{k=0}^{K}\frac{k(k+1)}{K(K+1)}\left[\frac{2\sqrt{2}\ell_{\lambda^{-1}}\epsilon_k}{k+1} + \frac{8\ell_{\lambda^{-1}}^2}{(k+1)^2}\left(\frac{L_{\theta,\theta}}{2} + \frac{\sqrt{2}\ell_\theta}{\ell_{\lambda^{-1}}}\right)\right] \\
&\leq \frac{2\sqrt{2}\ell_{\lambda^{-1}}}{K(K+1)}\sum_{k=1}^{K}(k\epsilon_k) + \frac{4\ell_{\lambda^{-1}}}{K+1}(\ell_{\lambda^{-1}}L_{\theta,\theta} + 2\sqrt{2}\ell_\theta) \\
&\leq \sqrt{2}\ell_{\lambda^{-1}}\max_{1 \leq k \leq K}\epsilon_k + \frac{4\ell_{\lambda^{-1}}}{K+1}(\ell_{\lambda^{-1}}L_{\theta,\theta} + 2\sqrt{2}\ell_\theta). 
\end{aligned} \tag{165}
$$

**(Case 2):** If $A_{K-1}(d_{K-1}) \geq 0$, then Eq. (162) holds for $k = K - 1$, i.e.,

$$\Gamma(\theta_K) - \min_{\theta' \in \Theta} \Gamma(\theta') \leq \sqrt{2}\ell_{\lambda^{-1}}\epsilon_{K-1} + 3\beta_{K-1}\ell_\theta \leq \sqrt{2}\ell_{\lambda^{-1}}\epsilon_{K-1} + \frac{6\sqrt{2}\ell_\theta\ell_{\lambda^{-1}}}{K+1}. \tag{166}$$

**(Case 3):** If $A_{K'-1}(d_{K'-1}) \geq 0$ for some $K' \in \{1, \ldots, K-1\}$ while $A_k(d_k) < 0$ for all $k = K', \ldots, K-1$, then we iterate Eq. (164) over $k = K', \ldots, K-1$ as follows.

$$\Gamma(\theta_K) - \min_{\theta' \in \Theta} \Gamma(\theta')$$

$$\leq \sum_{k=K'+1}^{K} \frac{k(k+1)}{K(K+1)}\left[\frac{2\sqrt{2}\ell_{\lambda^{-1}}\epsilon_k}{k+1} + \frac{8\ell_{\lambda^{-1}}^2}{(k+1)^2}\left(\frac{L_{\theta,\theta}}{2} + \frac{\sqrt{2}\ell_\theta}{\ell_{\lambda^{-1}}}\right)\right]$$

$$+ \frac{K'(K'+1)}{K(K+1)}[\Gamma(\theta_{K'}) - \min_{\theta' \in \Theta}\Gamma(\theta')]$$

$$\overset{(i)}{\leq} \frac{2\sqrt{2}\ell_{\lambda^{-1}}}{K(K+1)}\sum_{k=K'+1}^{K}(k\epsilon_k) + \frac{4\ell_{\lambda^{-1}}(K-K')}{K(K+1)}(\ell_{\lambda^{-1}}L_{\theta,\theta} + 2\sqrt{2}\ell_\theta)$$

$$+ \frac{K'(K'+1)}{K(K+1)}(\sqrt{2}\ell_{\lambda^{-1}}\epsilon_{K'} + 2\beta_{K'}\ell_\theta)$$

$$\overset{(ii)}{=} \frac{2\sqrt{2}\ell_{\lambda^{-1}}}{K(K+1)}\sum_{k=K'+1}^{K}(k\epsilon_k) + \frac{4\ell_{\lambda^{-1}}(K-K')}{K(K+1)}(\ell_{\lambda^{-1}}L_{\theta,\theta} + 2\sqrt{2}\ell_\theta)$$

$$+ \frac{K'(K'+1)}{K(K+1)}\left(\sqrt{2}\ell_{\lambda^{-1}}\epsilon_{K'} + \frac{4\sqrt{2}\ell_\theta\ell_{\lambda^{-1}}}{K'+2}\right)$$

$$\leq \sqrt{2}\ell_{\lambda^{-1}}\max_{K'+1\leq k\leq K}\epsilon_k + \frac{4\ell_{\lambda^{-1}}}{K+1}(\ell_{\lambda^{-1}}L_{\theta,\theta} + 2\sqrt{2}\ell_\theta), \tag{167}$$

where (i) applies Eq. (162) to $k = K'$ and (ii) uses $\beta_k = \frac{2\sqrt{2}\ell_{\lambda^{-1}}}{k+2}$.

In sum, the convergence rate (25) holds in all the above three cases.

## P   Proof of Corollary 1

Note that $A_k(d) := \max_{\xi \in \Xi_k}\left[\nabla_\theta f(\lambda_{\theta_k,\xi})^\top d\right]$ defined by Eq. (23) is convex and $\ell_\theta$-Lipschitz continuous. Hence, the best direction $d_k \in \arg\max_{d \in \{d_{k,t}:0\leq t\leq T_k\}} A_k(d)$ obtained from the subgradient method (28) converge at the following rate [11].

$$A_k(d_k) - \min_{d \in B_1} A_k(d) \overset{(i)}{\leq} \frac{\|d_{k,0} - d_k^*\|^2 + \ell_\theta^2 T\alpha^2}{2T\alpha} \overset{(ii)}{\leq} \frac{4+4}{24\ell_{\lambda^{-1}}\epsilon^{-1}} = \frac{\epsilon}{3\ell_{\lambda^{-1}}},$$

where (i) denotes $d_k^* \in \arg\max_{d \in B_1} A_k(d)$, (ii) uses $T = \frac{36\ell_{\lambda^{-1}}^2\ell_\theta^2}{\epsilon^2} = \mathcal{O}[(1-\gamma)^{-4}\epsilon^{-2}]$, $\alpha = \frac{\epsilon}{3\ell_{\lambda^{-1}}\ell_\theta^2}$ and $\|d_{k,0}\|, \|d_k^*\| \leq 1$. Hence, the accuracy $\epsilon_k = \frac{\epsilon}{3\ell_{\lambda^{-1}}}$ is achieved in Algorithm 2. Substituting $\epsilon_k = \frac{\epsilon}{3\ell_{\lambda^{-1}}}$, $K = \frac{8\ell_{\lambda^{-1}}}{\epsilon}(\ell_{\lambda^{-1}}L_{\theta,\theta} + 2\sqrt{2}\ell_\theta) = \mathcal{O}[(1-\gamma)^{-3}\epsilon^{-1}]$ into the convergence rate (25), we obtain that

$$\Gamma(\theta_K) - \min_{\theta' \in \Theta}\Gamma(\theta') \leq \sqrt{2}\ell_{\lambda^{-1}}\max_{1\leq k\leq K}\epsilon_k + \frac{4\ell_{\lambda^{-1}}}{K+1}(\ell_{\lambda^{-1}}L_{\theta,\theta} + 2\sqrt{2}\ell_\theta) \leq \epsilon.$$

The above $\epsilon$ accuracy above requires $K|V(\Xi)| = \mathcal{O}[|V(\Xi)|(1-\gamma)^{-3}\epsilon^{-1}]$ evaluations to $\lambda_{\theta_k,\xi}$, $f(\lambda_{\theta_k,\xi})$ and $\nabla_\theta f(\lambda_{\theta_k,\xi})$, $KT = \mathcal{O}[(1-\gamma)^{-7}\epsilon^{-3}]$ subgradient updates (28), and $K = \mathcal{O}[(1-\gamma)^{-3}\epsilon^{-1}]$ gradient descent updates to the policy gradient descent updates (22).

