# OpenReview forum: "Robust Reinforcement Learning with General Utility"
_NeurIPS.cc/2024/Conference — NeurIPS 2024 poster_

### Official Review · Reviewer_7fYr · 2024-07-03

**Soundness:** 1
**Presentation:** 1
**Contribution:** 2
**Rating:** 7
**Confidence:** 2

**Summary:**

The authors combine the topics of robust RL with general non-linear utility. They use policy gradient formulae from general utility RL and combine those with gradient algorithms for minimax problems due to Lin et al. The authors claim to have a convergence theory for the sample based algorithm that ensures convergence to a stationary point (convergence of gradient).

**Strengths:**

The topic is interesting and it is very reasonable to combine the different setups in the way the authors propose. The algorithm is clever, the results look (mostly) reasonable, a very extensive theoretical study is carried out.

**Weaknesses:**

It seems to me the authors did not understand the Landau O notation, at least the way it is used does not make much sense. The O notation is only reasonable (and only defined for) in a limiting sense where typically one limiting variable is chosen. For a fixed number there is no sense in the O notation, it could allow to multiply that number by any real which of course is non-sense! Looking at Theorem 2 the problem becomes visible. The statement as is does not make sense: The accuracy $\epsilon$ is fixed. Then a number of parameters of the algorithm are fixed as K=O(...), T=O(...), K'=O(...). What does that mean, O in what variable? Please compare with your Corollary 1, that statement makes sense. I went into the proofs to see if this is only imprecisely formulated, but it got even worse. The proof is full of estimates of the type $A\leq O(...)$ for a number $A$ and, even worse, $O(...)\leq A$. What is that supposed to be? The proofs are way too long (and this is not the referee's job) to check if the proofs can be saved or if there are O-constants that build up and spoil the proof. I am not too skeptical, the theorems looks somewhat reasonable. On the other hand a lot can happen if all constants are swept under the carpet, I cannot confirm correctness of the results. The analysis should be completely reworked and might make the article interesting for the next top tier conference.

Here are some further points:
- There is no clear motivation given for the article. As a Mathematician I agree the question is interesting, for a leading ML conference there should be some clearer motivation. The example is extremely artificial and not convincing at all.
- It is not particularly pleasant to read the article. I believe there are way too many repetitive (and imprecise) citations. One clear citation for the general utility policy gradient theorem should be enough, most citations given do not include such a variant. To me the article is too technical, a little bit of story telling would be very much appreciated. Perhaps the authors might want to think of splitting the article in two.
- There should be examples for the assumptions. For instance, what standard class of policies satisfy the policy assumptions? This is certainly standard for PG methods but almost all articles discuss some examples (softmax, linear softmax, ...).
- The complexity "result" of Theorem 2 is pretty extreme. For the usual $\gamma=0.99$ and $\epsilon=0.01$ the powers make it essentially infinite. I do not criticise the result, that's what the Maths gives. Still, the authors should point out that this is a theoretical estimate with no practical implication. Similarly, the theoretical iteration numbers and batch sizes are ridiculously large. This is a theoretical worst case estimate (perfectly fine with me) but it should be mentioned that in practice those numbers would not be used. If I am not wrong those numbers are not used in their own simulation example (which should also be mentioned).

**Questions:**

- It might be interesting to think about unbiased gradient estimators for general utility RL. Such have been found in the past years for linear RL, I guess something similar works in the non-linear case.
- Assumption 1, assumption for p. That assumption is somewhat crucial. I do not have a feeling, there is no discussion, if (or not) that assumption is strong. In what situation is the assumption satisfied? Is there any interesting situation in which the assumption is satisfied?
- Assumption 4 looks brutal to me. Is there any way to verify that assumption? Is there any example for which it can be verified? Perhaps that was addressed in past work, but some discussion is needed.
- The example: If I am not wrong the choices of parameters do not fit to the theory. Do they? If not, that should at least be discussed. How about the assumptions, can they be checked in that example? If not, I fear the article might prove something on the empty set.

**Limitations:**

As mentioned above I cannot confirm that I agree with the theoretical results. The statement of Theorem 2 seems incorrect and the proofs continue in the same fashion. Since the article is of only theoretical nature I do not think it can be published in the current form at the top conference.

Nonetheless, I believe the research direction is very reasonable and the authors have made very important and very non-trivial research progress! In my view the article needs a full major revision which is not in the scope of a conference revision.

---

> ### Author Rebuttal · Authors · 2024-08-07
>
> Thank you very much for reviewing our manuscript and providing valuable feedback. Below is a response to the review questions/comments. We will revise the manuscript accordingly after the review discussion period. Please let us know if further clarifications are needed.
>
> **Q1:** The weakness in the O notation requires reworking the long proof.
>
> **A:** Thank you for your suggestion. Similar to theoretical results in [1,2], our Theorem 2 uses the $\mathcal{O}$ notation to stress the dependence on the large positive constants like $(1-\gamma)^{-1}$ and $\epsilon^{-1}$. For example, $\mathcal{O}[(1-\gamma)^{-8}\epsilon^{-4}]$ is defined as $C(1-\gamma)^{-8}\epsilon^{-4}$ where the constant $C$ **does not depend on** $\gamma$ and $\epsilon$. This definition fits the following limiting sense.
> $$
>     \frac{\mathcal{O}[(1-\gamma)^{-8}\epsilon^{-4}]}{(1-\gamma)^{-8}\epsilon^{-4}}=\frac{C(1-\gamma)^{-8}\epsilon^{-4}}{(1-\gamma)^{-8}\epsilon^{-4}}
>    \to C {\rm ~ as ~} \gamma\to 1^-, \epsilon\to 0^+.
> $$
>
> The proof of Theorem 2 only uses the $\mathcal{O}$ notations to simplify the convergence rates at the end of Appendices N.4-N.5 and to derive the sample complexity in Appendix N.6. **These $\mathcal{O}$ notations take about only 1 page in total, and will be changed to explicit expressions in the revision within 3 hours, following the procedure below:**
>
> Step 1: At the end of Appendices N.4-N.5, by substituting Eqs. (93-94) into Eqs. (120) and (125), we can obtain **O-free convergence rates** of $\mathbb{E}\big[\|G_{b}^{(\theta)}(\theta_{\widetilde{k}},\xi_{\widetilde{k}})\|^2\big]$ and $\mathbb{E}\big[\|G_{a}^{(\xi)}(\theta_{\widetilde{k}},\xi_{\widetilde{k}})\|^2\big]$ respectively.
>
> Step 2: In Appendix N.6, the hyperparameters with $\mathcal{O}$-notations will be changed to **O-free hyperparameters** as follows.
> $$m_{\lambda}^{(1)}=m_{\theta}^{(1)}=m_{\lambda}^{(3)}=m_{\xi}^{(3)}=m_{\lambda}^{(4)}=m_{\theta}^{(4)}=c_1(1-\gamma)^{-10}\epsilon^{-4},
> m_{\lambda}^{(2)}=m_{\xi}^{(2)}=c_2(1-\gamma)^{-4}\epsilon^{-2},$$ $$H_{\lambda}^{(1)}=H_{\theta}^{(1)}=H_{\lambda}^{(2)}=H_{\xi}^{(2)}=H_{\lambda}^{(3)}=H_{\xi}^{(3)}=H_{\lambda}^{(4)}= H_{\theta}^{(4)}=\frac{c_3\log[(1-\gamma)^{-1}\epsilon^{-1}]}{1-\gamma},$$ $$K=c_4(1-\gamma)^{-8}\epsilon^{-4}, K'=c_5(1-\gamma)^{-9}\epsilon^{-4}], T'=c_6\log[(1-\gamma)^{-1}\epsilon^{-1}],$$
> where the constants $c_1,c_2,c_3,c_4,c_5,c_6>0$ **do not depend on** $\gamma$, $\epsilon$ and will be selected in Step 3.
>
> Step 3: In Appendix A.6, substitute the **O-free hyperparameters** (from Step 2) into the **O-free convergence rates** (from step 1). Then all the inequalities like $A\le \mathcal{O}(...)$ and $\mathcal{O}(...)\le A$ will no longer contain $\mathcal{O}$ notations and can be proved with sufficiently small constants $c_1,c_2,c_3,c_4,c_5,c_6>0$.
>
> Step 4: In Appendix A.6, substitute the **O-free hyperparameters** (from Step 2) into the sample complexity.
>
> **About other $\mathcal{O}$ notations:** All $\mathcal{O}$ notations in the main text follow the above definition and are used to present either results or intuitive proof sketch, which is typical in ML optimization works. Appendices D, E, P and Q containing $\mathcal{O}$ notations will be removed since these sections focus on concave utilities and other utilities that satisfy weak Minty variational inequality that lack application examples.
>
> [1] Barakat, Anas, Ilyas Fatkhullin, and Niao He. "Reinforcement learning with general utilities: Simpler variance reduction and large state-action space." International Conference on Machine Learning. PMLR, 2023.
>
> [2] Zhang, Junyu, et al. "On the convergence and sample efficiency of variance-reduced policy gradient method." Advances in Neural Information Processing Systems 34 (2021): 2228-2240.
>
> **Q2:** There is no clear motivation given for the article.
>
> **A:** Thank you for pointing out the motivation issue. We have elaborated our motivation with application examples in our global response. We will add these examples to our revised paper.
>
> **Q3:** There are way too many repetitive (and imprecise) citations. One clear citation for the general utility policy gradient theorem should be enough, most citations given do not include such a variant.
>
> **A:**  Thank you for your suggestion. Existing policy gradient theorems only give $\nabla_{\theta}f(\lambda_{\theta,\xi})$ while we also need transition kernel gradient $\nabla_{\xi}f(\lambda_{\theta,\xi})$, so we present both in Theorem 1 instead of citing policy gradient theorems. We guess you may misunderstand that the 5 citations right before Theorem 1 are used to support Theorem 1. However, we have clearly stated that they are used as examples of linear utility functions, not policy gradient theorem.
>
> What other citations do you think are repetitive and imprecise?
>
> **Q4:** To me the article is too technical, a little bit of story telling would be very much appreciated. Perhaps the authors might want to think of splitting the article in two.
>
> **A:** Thank you for your suggestions. In the revised main text, we will add stories including our motivation with application examples (elaborated in our global response) and examples for our assumptions (elaborated in our answer to your Q5), and move the propositions to the appendices to make more main-text space for story telling. To shorten the article, we will remove the parts of concave utilities and other utilities that satisfy weak Minty variational inequality since they lack application examples, so that our theory mainly consists of the two convergence theorems, for gradient and global convergence respectively.

---

> > ### Comment · Reviewer_7fYr · 2024-08-07
> > **Thanks & further comments**
> >
> > Thanks for going through your proof again!
> > I strongly disagree on the opinion that sloppy work of others justifies sloppy work of oneself. Good ML papers have clean statements, clean proofs, and only use the O-notation in text or tables for easy comparison. I am happy the way you used the O-notation is much stronger than what the O-notation really is. Please work properly to keep the ML community sound. It might be useful to familiarise yourself with standard notation, https://en.wikipedia.org/wiki/Big_O_notation. Otherwise sooner or later this will result in trouble once good and bad use of standard notation is mixed.
> >
> > The way you formulate in the rebuttal the constants would likely depend on $\epsilon$ as you suggest to chose the constants in the last step where you aim to achieve $\epsilon^2$ upper bound. I checked your proof and it seems ok, you can already choose the constants in step 3 without dependence on the target $\epsilon^2$. I think there are more constants to chose carefully as you say so a careful modification touches more than 1 page, but that's not the point. I am fine with your improvement and will improve the score. There were enough reviewers that hopefully checked these details as well.
> >
> > Q3: As an example, have a look at the text below Assumption 3. Nobody is going to read that, the overview is extremely wide. Similarly the intro to robust RL at line 97. I would suggest to replace this paragraph by text and collect some citations without names to safe a lot of space. I certainly do not decrease/increase my score for these matters of taste.

---

> > > ### Author Response · Authors · 2024-08-08
> > > **Authors' response to Q3 (2)**
> > >
> > > Q3: We replaced the named citations in these paragraphs with numbered citations as follows.
> > >
> > > Intro to robust RL at line 97:
> > >
> > > Robust RL is designed to learn a policy that is robust to perturbation of environmental factors. Usually robust RL is NP-hard [43], but becomes tractable for ambiguity sets that is (s, a)-rectangular [33, 18, 43, 41, 28, 54] or s-rectangular [43, 40, 22, 25]. Methods to solve robust RL include value iteration [33, 18, 43, 14, 24], policy iteration [18, 4, 23] and policy gradient [28, 41, 54, 40, 25, 16, 27].
> > >
> > > Under Assumption 3:
> > >
> > > Robust RL with convex utility subsumes three important special cases, the commonly used convex RL problem [52 , 49 , 6] with fixed $\xi$, the standard robust RL [33, 18, 43, 40] with linear utility function $f$, and the standard RL [39] with linear utility function f and fixed $\xi$. Convex RL can be applied to maximum-entropy exploration [17, 52, 49, 12, 6], constrained RL [52, 49, 12] and demonstration learning [52, 49].

---

> ### Author Response · Authors · 2024-08-07
> **Author Response to Reviewer 7fYr (2)**
>
> **Q5:** There should be examples for the assumptions. For instance, what standard class of policies satisfy the policy assumptions? This is certainly standard for PG methods but almost all articles discuss some examples (softmax, linear softmax, ...).
>
> **A:** Thank you for your suggestion. In the revised paper, we will add the following practical examples for the assumptions.
>
> Assumption 1 covers popular policy parameterizations including softmax policy $\pi_{\theta}(a|s)=\frac{\exp(\theta_{s,a})}{\sum_{a'}\exp(\theta_{s,a'})}$ [1] and log-linear policy $\pi_{\theta}(a|s)=\frac{\exp(\theta\cdot\phi_{s,a})}{\sum_{a'}\exp(\theta\cdot\phi_{s,a'})}$ [1], as well as popular transition kernels including direct parameterization $p_{\xi}(s'|s,a)=\xi _ {s,a,s'}$ [2,3] and linear parameterization $p_{\xi}(s'|s,a)=\xi\cdot\phi_{s,a,s'}$ [4,5] when $\Xi$ is located away from 0, i.e., $\inf_{s,a,s',\xi\in\Xi}p_{\xi}(s'|s,a)\ge p_{\min}$ for a constant $p_{\min}>0$. Assumptions 2-3 cover the three convex utility examples in Section 2 of [6], including MDP with Constraints, pure exploration, and learning to mimic a demonstration. Assumptions 4 and 8 are similar and cover direct policy parameterization $\pi_{\theta}(a|s)=\theta _ {s,a}$, since it has been proved that direct policy parameterization satisfies Assumption 4.1 of [6] which implies Assumptions 4 and 8. Assumptions 5-7 cover $s$-rectangular $L_1$ and $L_{\infty}$ ambiguity sets (defined as $\Xi=\{\xi:\|\xi(s,:,:)-\xi^0(s,:,:)\| _ p\le \alpha_s\}$ for $p\in\{1,\infty\}$ respectively using direct kernel parameterization $p_{\xi}(s'|s,a)=\xi_{s,a,s'}$), which are very popular in robust RL [7,8].
>
> [1] Agarwal, Alekh, et al. "On the theory of policy gradient methods: Optimality, approximation, and distribution shift." Journal of Machine Learning Research 22.98 (2021): 1-76.
>
> [2] Kumar, Navdeep, et al. "Policy gradient for rectangular robust markov decision processes." Advances in Neural Information Processing Systems (2023).
>
> [3] Behzadian, Bahram, Marek Petrik, and Chin Pang Ho. "Fast Algorithms for $L_\infty$-constrained S-rectangular Robust MDPs." Advances in Neural Information Processing Systems (2021).
>
> [4] Ayoub, Alex, et al. "Model-based reinforcement learning with value-targeted regression." International Conference on Machine Learning. PMLR, 2020.
>
> [5] Zhang, Junkai, Weitong Zhang, and Quanquan Gu. "Optimal horizon-free reward-free exploration for linear mixture mdps." International Conference on Machine Learning. PMLR, 2023.
>
> [6] Zhang, Junyu, et al. "Variational policy gradient method for reinforcement learning with general utilities." Advances in Neural Information Processing Systems 33 (2020): 4572-4583.
>
> [7] Hu, Xuemin, et al. "Long and Short-Term Constraints Driven Safe Reinforcement Learning for Autonomous Driving." ArXiv:2403.18209 (2024).
>
> [8] Khairy, Sami, et al. "Constrained deep reinforcement learning for energy sustainable multi-UAV based random access IoT networks with NOMA." IEEE Journal on Selected Areas in Communications 39.4 (2020): 1101-1115.
>
> **Q6:** The authors should point out that this is a theoretical estimate with no practical implication. Similarly, the theoretical iteration numbers and batch sizes are ridiculously large. This is a theoretical worst case estimate (perfectly fine with me) but it should be mentioned that in practice those numbers would not be used. If I am not wrong those numbers are not used in their own simulation example (which should also be mentioned).
>
> **A:** Thank you for your suggestion. Yes, you are right. These theoretical and conservatively large hyperparameter choices are not necessarily needed in practical implementation, so the hyperparameters in our simulation are obtained by fine-tuning not theory. We will add this explanation in the revision.
>
> **Q7:** It might be interesting to think about unbiased gradient estimators for general utility RL. Such have been found in the past years for linear RL, I guess something similar works in the non-linear case.
>
> **A:** Thank you for your suggestion. All the robust RL with general utility works we found either only provide true gradients or provide gradient estimators with bias which goes to 0 as the truncation level $H\to+\infty$. We select such a gradient estimator with sufficiently large $H$ to bound the bias.
>
> **Q8:** About Assumption 1 for p, there is no discussion, if (or not) that assumption is strong. In what situation is the assumption satisfied? Is there any interesting situation in which the assumption is satisfied?
>
> **A:** Thank you for your suggestion. See our answer to your Q5.
>
> **Q9:** Assumption 4 looks brutal to me. Is there any way to verify that assumption? Is there any example for which it can be verified? Perhaps that was addressed in past work, but some discussion is needed.
>
> **A:** Good question and suggestion. See our answer to your Q5.

---

> > ### Comment · Reviewer_7fYr · 2024-08-07
> > **Answer 2**
> >
> > Q5: I fear you must be more specific. Which policies/MDPs do satisfy ALL assumptions of your Thm 2, 3, Prop 3, 4, 5. Not only for the review process, that must be integral part of the paper to justify the statements are not empty. I am satisfied if I get one precise example as an answer.
> >
> > Q7: Not for this revision but for future work you might want to have a loot (for instance at)
> >
> > Kaiqing Zhang, Alec Koppel, Hao Zhu, and Tamer Ba¸sar. “Global convergence of policy gradient methods to (almost) locally optimal policies”. SIAM Journal on Control and Optimization, 2020.
> >
> > It is so obvious how to obtain unbiased gradient estimators that it is amazing how many people just do truncation.

---

> > > ### Author Response · Authors · 2024-08-08
> > > **The 1 example for Reviewer 7fYr's Q5**
> > >
> > > The combination of the following policy, transition kernel, ambiguity set and utility function satisfy all the assumptions (Assumptions 1-8) of Theorems 2, 3 and Proposition 3, 4, 5.
> > >
> > > $\bullet$ Softmax policy: $\pi _ {\theta}(a|s)=\frac{\exp(\theta _ {s,a})}{\sum _ {a'}\exp(\theta _ {s,a'})}, \theta\in\Theta$, where the range $\Theta\subseteq[-R,R]^{|\mathcal{S}|\times|\mathcal{A}|}$ for some constant $R>0$ to prevent $\pi _ {\theta}(a|s)$ from approaching 0.
> > >
> > > $\bullet$ Directly parameterized transition kernel: $p _ {\xi}(s'|s,a)=\xi _ {s,a,s'}$.
> > >
> > > $\bullet$ s-rectangular $L^1$ ambiguity set: $\Xi\overset{\rm def}{=}${$\xi:||\xi(s,:,:)-\xi^0(s,:,:)||_1\le \alpha_s, \forall s$}, where the fixed nominal kernel $\xi^0$ satisfies $\xi^0(s,a,s')>\alpha_s, \forall s,a,s'$ to prevent $\xi(s,a,s')$ from approaching 0.
> > >
> > > $\bullet$ Convex utility function for pure exploration [1]: $f(\lambda)=\sum_s \lambda(s)\log\lambda(s)$ where $\lambda(s)\overset{\rm def}{=}\sum_a\lambda(s,a)$. The inital state distribution $\rho$ satisfies $\rho(s)>0, \forall s$ to prevent $\lambda(s)$ from approaching 0.
> > >
> > > [1] Zhang, Junyu, et al. "Variational policy gradient method for reinforcement learning with general utilities." Advances in Neural Information Processing Systems 33 (2020): 4572-4583.

---

> > > > ### Comment · Reviewer_7fYr · 2024-08-08
> > > > **Thanks, I will raise my score.**
> > > >
> > > > Thank you very much for the clarification, my major concerns (flaws in the proofs, statement of theorem, possible empty result) have been resolved I will raise my score to 7.
> > > >
> > > > One last comment. When you rewrite the proof for the constants, it is perhaps possible to explicitly keep track of all constants that appear in the algorithmic choices needed for Theorem 2? I had the impression when checking if your modification looks right. If so you should do it (as it is the theorem could not be used even if I ignore the huge powers) but I leave it to you.

---

> > > > > ### Author Response · Authors · 2024-08-08
> > > > > **Thank Reviewer 7fYr very much**
> > > > >
> > > > > Thank you very much, Reviewer 7fYr.
> > > > >
> > > > > Yes, It is possible to explicitly keep track of all constants that appear in the algorithmic choices needed for Theorem 2, i.e., $c_1,\cdots, c_6$ in our initial response. We are conducting the change.
> > > > >
> > > > > Best regards,
> > > > > Authors

---

> ### Author Response · Authors · 2024-08-07
> **Author Response to Reviewer 7fYr (3)**
>
> **Q10:** The example: If I am not wrong the choices of parameters do not fit to the theory. Do they? If not, that should at least be discussed. How about the assumptions, can they be checked in that example?
>
> **A:** Good question and suggestion. Our simulation does not exactly follow the theoretical hyperparameter choices and assumptions. We will explain this in our revision.
>
> **About Summary:** Our work not only provides Algorithm 1 with gradient convergence results, but also Algorithm 2 with **global** convergence results.

---

### Official Review · Reviewer_nGSn · 2024-07-10

**Soundness:** 3
**Presentation:** 3
**Contribution:** 4
**Rating:** 7
**Confidence:** 3

**Summary:**

In this submission, the authors tackle a new problem: how to train a policy with a general utility when one needs to be robust to some uncertainty in the environment. More precisely, they are looking for the solution of min-max problem: the minimization over a set of parametric policies of the worst case over a set of possible transition probabilities of a (convex) functional of the occupancy measure. They propose a policy gradient type algorithm for which they prove that the gradient goes to 0. They also introduce a more complex algorithm for a specific classical uncertainty set (s-regular polyhedral ambiguity set) that is provably convergent toward the global optimal solution.

**Strengths:**

- The authors attack a novel problem (the combination of general utility and robustness) for which they construct algorithms with theoretical guarantees.

- The content is very technical, but the writing makes it understandable. I really liked the way the authors stress their specific contribution.

- The proof seem correct, although I did not check all of them extensively.

- The results are supported by some numerical experiments).

**Weaknesses:**

- The nonconvex case is only handled in the Appendix. I liked the fact that they have much more in their pocket than the content of the main article. The nonconvex case could nevertheless have been mentioned only in the conclusion as it is not really part of the main article.

- The numerical experiments should be part of the main article as they add a lot to the theoretical results.

- The results are very technical and some explanations (for example for the main theorems) could be beneficial to the readers.

**Questions:**

Can the author explain if the control on the size of the gradient with respect to \xi corresponds to a convergence for the functional itself or just that the algorithm is going to slow down?


Typos and misc:
- 134: $\quad$ before $,$
- 309: $\epsilon_k$ only defined in Algorithm

**Limitations:**

The complexity of their algorithms is quite high, as it is mentioned, and the convergences are local or in a specific setting, but this makes sense for a first paper in a topic.

---

> ### Author Rebuttal · Authors · 2024-08-04
>
> Thank you very much for reviewing our manuscript and providing valuable feedback. Below is a response to the review questions/comments. We will revise the manuscript accordingly after the review discussion period. Please let us know if further clarifications are needed.
>
> **Q1:** The nonconvex case is only handled in the Appendix. I liked the fact that they have much more in their pocket than the content of the main article. The nonconvex case could nevertheless have been mentioned only in the conclusion as it is not really part of the main article.
>
> **A:** Thank you for your suggestion and appreciation. Based on all the reviewers' feedback, we will remove concave utility and other utilities that satisfy weak Minty variational inequality, since they lack application examples.
>
> **Q2:** The numerical experiments should be part of the main article as they add a lot to the theoretical results.
>
> **A:** Thank you for your suggestion and appreciation. We originally planned to put numerical experiments in the main article. Later, we found that after fitting some other necessary parts into the main article, including problem formulation, algorithms and theoretical results for both gradient and global convergence, the 9-page limit did not allow experiments. Therefore, in line 194 in the main article, we refer the experiments to Appendix A.
>
> **Q3:** The results are very technical and some explanations (for example for the main theorems) could be beneficial to the readers.
>
> **A:** Thank you for your suggestion.
>
> Theorem 2 provides the first sample complexity result to achieve $\epsilon$-stationtary point of the objective function $f(\lambda_{\theta,\xi})$ for robust RL with general utility. Specifically, we select sufficiently large batchsizes and truncation numbers to ensure that the stochastic gradients are close to the true gradients, and select sufficiently many iterations to ensure optimization progress. These theoretical and conservatively large hyperparameter choices are not necessarily needed in practical implementation.
>
> Theorem 3 provides the first global convergence rate for robust RL with general utility. The global convergence error consists of sublinear convergence rate $\mathcal{O}(1/K)$ with $K$ iterations, and $\max_{1\le k\le K}\epsilon_k$ which denotes the convergence error of the subroutine for maximizing the concave function $A_k$.
>
> **Q4:** Can the author explain if the control on the size of the gradient with respect to $\xi$ corresponds to a convergence for the functional itself or just that the algorithm is going to slow down?
>
> **A:** Good question. We select sufficiently large batchsizes and truncation numbers to ensure that the estimated stochastic gradients are close to the true gradients $\nabla_{\theta} f(\lambda_{\theta,\xi})$ and $\nabla_{\xi} f(\lambda_{\theta,\xi})$.
>
> **Q5:** "134: \quad before ,"
>
> **A:** Thank you for pointing this out. We will add spaces before the commas.
>
> **Q6:** 309: $\epsilon_k$ only defined in Algorithm.
>
> **A:** Thank you for pointing this out. We will modify line 309 to "with $\beta_k=\frac{2\sqrt{2}\ell_{\lambda}}{k+2}$, $\sigma_k=\frac{4\sqrt{2}\ell_{\theta}\ell_{\lambda}}{k+2}$ and any $\epsilon_k>0$", since the convergence rate (20) holds for any $\epsilon_k>0$.

---

> > ### Comment · Reviewer_nGSn · 2024-08-12
> >
> > Thanks for the authors' response. It addressed most of my concerns, and I believe the paper meets the acceptance threshold.

---

> > > ### Author Response · Authors · 2024-08-12
> > > **Thank Reviewer nGSn for your appreciation**
> > >
> > > Thank Reviewer nGSn for your appreciation.
> > >
> > > Best regards,
> > >
> > > Authors

---

### Official Review · Reviewer_joNT · 2024-07-13

**Soundness:** 3
**Presentation:** 2
**Contribution:** 3
**Rating:** 5
**Confidence:** 3

**Summary:**

This paper aims to incorporate robustness in reinforcement learning environments by allowing the transition kernel to be within a polyhedral s-rectangular uncertainty set to handle general utility functions. It applies a stochastic gradient descent with (gradient sampling subroutines) algorithm designed for global convergence and presents numerical experiments to showcase its convergence properties.

**Strengths:**

The methodology behind robustifying the transition kernel is well thought out. The theory is supported by proofs that are generally clear to follow. The convergence behavior is validated by numerical simulations.

**Weaknesses:**

- Modeling motivation: The paper lacks a comparative analysis to demonstrate superiority over existing non-robust methods in handling scenarios like constrained and risk-averse RL. To solidify its contributions, it should include empirical or theoretical evidence comparing its performance against established methods, especially in real-world scenarios that involve distributional shifts.

- Computational costs. Detailed evaluations of the computational overhead comparing to non-robust approaches needs to be presented to demonstrate the price of robustness and usefulness of the modeling.

- Scalability. The robustness in transition kernels is modeled through s-rectangular uncertainty with direct parametrization of the transition kernels, which appears to be less advantageous for scalability and generalization in complex or continuous environments. The paper should discuss potential adaptations or extensions of the method that can handle larger state spaces without compromising theoretical integrity.

- Hyperparameters. The algorithm requires managing an extensive set of hyperparameters, complicating its practical implementation. The paper should propose methods to reduce hyperparameter complexity, such as adaptive tuning or simplified parameter settings, to enhance usability and accessibility.

**Questions:**

- Comparison with Barakat et al. 2023: It would be helpful if the authors could provide a more detailed comparative analysis of their approach with the work by Barakat et al., especially in terms of algorithmic differences and performance in continuous state-action spaces.
- $\ell_{\lambda^{-1}}$: please provide an explicit example of this constant and its dependence on |S| and |A|
- Line 230: Inversible -> Invertible

**Limitations:**

The authors touched upon the limitations of theoretical complexity

---

> ### Author Rebuttal · Authors · 2024-08-07
>
> Thank you very much for reviewing our manuscript and providing valuable feedback. Below is a response to the review questions/comments. We will revise the manuscript accordingly after the review discussion period. Please let us know if further clarifications are needed.
>
> **Q1:** Modeling motivation: The paper lacks a comparative analysis to demonstrate superiority over existing non-robust methods in handling scenarios like constrained and risk-averse RL. To solidify its contributions, it should include empirical or theoretical evidence comparing its performance against established methods, especially in real-world scenarios that involve distributional shifts.
>
> **A:** Thank you for your suggestion. Our work is fundamentally different from the works on *non-robust* RL with general utility, not only in methodology, but also in the objective function and convergence measures. Specifically, *non-robust* RL with general utility aims to solve $\min_{\theta\in\Theta}f(\lambda_{\theta,\xi_0})$ under a fixed environment $\xi_0\in\Xi$, while our proposed *robust* RL with general utility aims to solve $\min_{\theta\in\Theta}\max_{\xi\in\Xi}f(\lambda_{\theta,\xi})$ under the worst possible test environment $\xi\in\Xi$. As a result, our convergence measures are also very different, so the performance is not directly comparable. We will stress these fundamental differences in our revised paper.
>
> **Q2:** Computational costs. Detailed evaluations of the computational overhead comparing to non-robust approaches needs to be presented to demonstrate the price of robustness and usefulness of the modeling.
>
> **A:** Thank you for your suggestion. As mentioned in our answer to Q1, the computational overhead is not directly comparable, since the non-robust approaches and our robust approaches have different objectives and different convergence measures.
>
> **Q3:** Scalability. The robustness in transition kernels is modeled through s-rectangular uncertainty with direct parameterization of the transition kernels, which appears to be less advantageous for scalability and generalization in complex or continuous environments. The paper should discuss potential adaptations or extensions of the method that can handle larger state spaces without compromising theoretical integrity.
>
> **A:** Thank you for your suggestion. To extend to large state spaces, we can adopt linear occupancy measure $\lambda_H^{\pi_{\theta}}(s,a)\approx \omega_{\theta}\cdot\phi(s,a)$ (Barakat et al. 2023) and linear transition kernel parameterization $p_{\xi}(s'|s,a)=\xi\cdot\phi_{s,a,s'}$ [1,2], with parameters $\omega_{\theta}$ and $\xi$ of scalable dimensionality. To extend to Robust RL with continuous state space, we can adopt Gaussian policy (Barakat et al. 2023) and transition kernel $s_{t+1}=f(s_t,a_t)+\omega_t$ with Gaussian noise $\omega_t\in\mathbb{R}^d$ [3]. We will add these potential extensions to our conclusion section.
>
> [1] Ayoub, Alex, et al. "Model-based reinforcement learning with value-targeted regression." International Conference on Machine Learning. PMLR, 2020.
>
> [2] Zhang, Junkai, Weitong Zhang, and Quanquan Gu. "Optimal horizon-free reward-free exploration for linear mixture mdps." International Conference on Machine Learning. PMLR, 2023.
>
> [3] Ramesh, Shyam Sundhar, et al. "Distributionally robust model-based reinforcement learning with large state spaces." International Conference on Artificial Intelligence and Statistics. PMLR, 2024.
>
> **Q4:** Hyperparameters. The algorithm requires managing an extensive set of hyperparameters, complicating its practical implementation. The paper should propose methods to reduce the complexity of hyperparameters, such as adaptive tuning or simplified parameter settings, to improve usability and accessibility.
>
> **A:** Thank you for your suggestion. Based on Theorem 2, we can reduce the hyperparameters in Algorithm 1 by replacing $m_{\lambda}^{(1)}, m_{\theta}^{(1)}, m_{\lambda}^{(3)}, m_{\theta}^{(3)}, m_{\lambda}^{(4)}, m_{\theta}^{(4)}$ with $m^{(1)}$, replacing $m_{\lambda}^{(2)}, m_{\theta}^{(2)}$ with $m^{(2)}$, and replacing $\{H_{\lambda}^{(k)},H_{\theta}^{(k)}\}_{k=1}^4$ with $H$. Based on Theorem 3 and Corollary 1, we can reduce the hyperparameters in Algorithm 2 by replacing $\sigma_k$, $\epsilon_k$ and $\beta_k$ with $\frac{\sigma}{k+2}$, $\epsilon$ and $\frac{\beta}{k+2}$ respectively.
>
> **Q5:** Comparison with Barakat et al. 2023: It would be helpful if the authors could provide a more detailed comparative analysis of their approach with the work by Barakat et al., especially in terms of algorithmic differences and performance in continuous state-action spaces.
>
> **A:** Thank you for your suggestion. As mentioned in our answer to Q1, the most fundamental difference between our work and (Barakat et al. 2023) is that we aim at different objectives and convergence measures. As a result, the biggest difference in algorithms is that (Barakat et al. 2023) only updates policy parameter $\theta$ under fixed transition kernel parameter $\xi_0\in\Xi$ while we update both $\theta$ and $\xi\in\Xi$. The performance is not directly comparable also due to the differences in objectives and convergence measures.
>
> **Q6:** $\ell_{\lambda^{-1}}$: please provide an explicit example of this constant and its dependence on |S| and |A|.
>
> **A:** Thank you for your suggestion. It has been proved in [4] that for direct policy parameterization $\pi_{\theta}(a|s)=\theta_{s,a}$, $\ell_{\lambda^{-1}}=\frac{2}{\min_s\rho(s)}\ge \frac{2}{|S|}$ where $\rho$ is the initial state distribution.
>
> [4] Zhang, Junyu, et al. "Variational policy gradient method for reinforcement learning with general utilities." Advances in Neural Information Processing Systems 33 (2020): 4572-4583.
>
> **Q7:** Line 230: Inversible -> Invertible.
>
> **A:** Thank you for pointing out this typo. We will correct that.

---

> > ### Comment · Reviewer_joNT · 2024-08-09
> >
> > Thank you for your response.
> >
> > I understand that robust and non-robust approaches have different objective functions in the optimization formulation. Still, one driving motivation for using robust approaches is to achieve better performance in the face of the sim-to-real gap because the robust approach avoids overfitting (see e.g. introduction and fig. 5 of [1] for static stochastic programming setting, and fig. 1 of [2] for offline RL setting).
> >
> > I also agree that robust constrained RL and robust entropy-regularized RL can be viewed as a special case of robust RL with general utility. What remains unclear to me is e.g. the performance of existing robust constrained RL approaches against your robust RL with general utility algorithms on the same test problems. Eventually, we want to solve specific cases and I would argue that it is worthwhile to show that your more general algorithm provides superior results than more tailored algorithms.
> >
> > [1] Mohajerin Esfahani, Peyman, and Daniel Kuhn. "Data-driven distributionally robust optimization using the Wasserstein metric: Performance guarantees and tractable reformulations." Mathematical Programming 171.1 (2018): 115-166.
> >
> > [2] Shi, Laixi, and Yuejie Chi. "Distributionally robust model-based offline reinforcement learning with near-optimal sample complexity." Journal of Machine Learning Research 25.200 (2024): 1-91.

---

> > > ### Author Response · Authors · 2024-08-09
> > > **Complexity Result Comparison for Reviewer joNT**
> > >
> > > We compare our complexity results for robust RL with general utility vs. the state-of-the-art complexity results for *robust constrained RL (RC-RL)* and *entropy regularized robust RL (ER-RL)* as follows. For fair comparison, we focus on policy gradient methods and leave aside the value iteration [1-2] and policy iteration [2] methods of ER-RL, since they directly leverage the Bellman equation for linear utility function $f$ that does not hold for general utility function $f$.
> > >
> > > (1) To achieve an $\epsilon$-global optimal point for non-stochastic setting, our Algorithm 2 takes $KT=\mathcal{O}(\epsilon^{-3})$ iterations based on our Corollary 1, while ER-RL takes $\mathcal{O}(\epsilon^{-3}\log\epsilon^{-1})$ iterations [3], and there is no global convergence result yet for RC-RL to our knowledge.
> > >
> > > (2) To achieve an $\epsilon$-stationary point for stochastic setting, our Algorithm 1 uses $\mathcal{O}(\epsilon^{-10})$ samples, while to our knowledge, ER-RL has no gradient convergence results, and the only available complexity result for RC-RL is $\mathcal{O}(\epsilon^{-14})$ in Remark 1 at the end of page 27 of [4].
> > >
> > > **As a result, our complexity results either outperform those of ER-RL and RC-RL, or even fill in their blanks in complexity results.**
> > >
> > > [1] Mankowitz, D. J., Levine, N., Jeong, R., Abdolmaleki, A., Springenberg, J. T., Shi, Y., Kay, J., Hester, T., Mann, T., and Riedmiller, M. (2019). Robust reinforcement learning for continuous control with model misspecification. In Proceedings of the International Conference on Learning Representations (ICLR).
> > >
> > > [2] Mai, T. and Jaillet, P. (2021). Robust entropy-regularized markov decision processes. ArXiv:2112.15364.
> > >
> > > [3] Chen, Z. and Huang, H. (2024). Accelerated Policy Gradient for s-rectangular Robust MDPs with Large State Spaces. In Proceedings of the International Conference on Machine Learning (ICML).
> > >
> > > [4] Wang, Y., Miao F., and Zou, S. (2022). Robust constrained reinforcement learning. ArXiv:2209.06866.

---

> > > > ### Comment · Reviewer_joNT · 2024-08-10
> > > >
> > > > Thank you for your response, I have increased my score accordingly.

---

> > > > > ### Author Response · Authors · 2024-08-10
> > > > > **Thank Reviewer joNT very much**
> > > > >
> > > > > Hello, Reviewer joNT,
> > > > >
> > > > > Thank you very much.
> > > > > Best regards,
> > > > >
> > > > > Authors

---

> ### Author Response · Authors · 2024-08-07
> **About Reviewer joNT's summary**
>
> The properties mentioned in Reviewer joNT's summary belong to two algorithms respectively.
>
> Algorithm 1: stochastic gradient descent with gradient sampling subroutines, general uncertainty set, gradient convergence.
>
> Algorithm 2: polyhedral s-rectangular uncertainty set, global convergence.

---

### Official Review · Reviewer_iNNQ · 2024-07-13

**Soundness:** 3
**Presentation:** 2
**Contribution:** 2
**Rating:** 6
**Confidence:** 4

**Summary:**

The paper studies the problem of robust RL with general utility function, which is looking at maximizing a general (possibly non-convex) utility function with the worst-case possible transition kernels in an ambiguity set. The paper provides convergence analysis for a wide range of utility functions and ambiguity set.

**Strengths:**

The technical contribution mainly comes from the following convergence analysis:

1. With convex utility, a two-phase projected stochastic gradient descent ascent algorithm is shown to converge to stationary point.
2. WIth convex utility and an s-rectangular polyhedral ambiguity set, global convergence is proven.
3. The gradient convergence is also proven for concave utility and utility that satisfies weak Minty variational inequality.

**Weaknesses:**

My main concern is about the motivation, interpretation of the results, along with potential tightness results. Please see question section.

**Questions:**

1. I think the authors might do a better job motivating and introducing the problem at the introduciton section. I'm confused about the exact definition of robust RL with general utility until the end of section 2. Is it possible to provide intuitive explanations at the beginning?

2. it might be easier to understand the 4 different settings if the authors can motivate with good examples in practice (convex utility, concave utility, polyhedral ambiguity set and weak Minty variational inequality).

3. There have been a lot of assumptions in the paper before the convergence results. Can authors comment on how practical those assumptions are? And under such assumptions, can the authors comment on the tightness of the results in terms of sample complexity / iteration complexity?

---

> ### Author Rebuttal · Authors · 2024-08-06
>
> Thank you very much for reviewing our manuscript and providing valuable feedback. Below is a response to the review questions/comments. We will revise the manuscript accordingly after the review discussion period. Please let us know if further clarifications are needed.
>
> **Q1:** I think the authors might do a better job motivating and introducing the problem at the introduction section. I'm confused about the exact definition of robust RL with general utility until the end of section 2. Is it possible to provide intuitive explanations at the beginning?
>
> **A:** Thank you for your suggestion. In the **revised introduction**, we will briefly compare the definitions of the existing *RL with general utility problem* and our proposed *robust RL with general utility problem*. Specifically, the existing *RL with general utility problem* is formulated as $\min_{\theta} f(\lambda_{\theta,\xi_0})$ where the agent aims to select its policy parameter $\theta$ to minimize the cost-related utility function $f$, under a **fixed transition kernel parameter** $\xi_0$. In contrast, *our robust RL with general utility problem* is formulated as $\min_{\theta} \max_{\xi\in\Xi}f(\lambda_{\theta,\xi})$ where the agent aims to select a robust optimal policy $\theta$ that minimizes the utility **under the worst possible transition kernel parameters** $\xi\in\Xi$. We will refer the readers to Section 2 for more details. To motivate our research, we will also briefly summarize the useful special cases and application examples of our proposed problem listed in our global response.
>
> **Q2:** It might be easier to understand the 4 different settings if the authors can motivate with good examples in practice (convex utility, concave utility, polyhedral ambiguity set and weak Minty variational inequality).
>
> **A:** Thank you for your suggestion.
>
> Based on all the reviewers' feedback, we will remove concave utility and other utilities that satisfy weak Minty variational inequality, since they lack application examples.
>
> We have listed two popular examples of polyhedral ambiguity set, $L_1$ and $L_{\infty}$ ambiguity sets in lines 48-49 in the introduction, and defined these sets right after Assumption 7. To have stronger motivation, we will also add the explanation that these ambiguity sets are popular in robust reinforcement learning, an important special case of our proposed robust RL with general utility.
>
> We will add the following examples of convex utilities [1] to our revised paper.
>
> (1) MDP with Constraints or Barriers, which has safety-critical applications including healthcare [2], autonomous driving [3] and unmanned aerial vehicle [4].
>
> (2) Pure exploration, which can be applied to explore an environment that lacks reward signals [5].
>
> (3) Learning to mimic a demonstration, which is used to help the agent mimic the expert demonstration [1].

---

> ### Author Response · Authors · 2024-08-06
> **Author Response to Reviewer iNNQ (2)**
>
> **Q3:** There have been a lot of assumptions in the paper before the convergence results. Can authors comment on how practical those assumptions are? And under such assumptions, can the authors comment on the tightness of the results in terms of sample complexity / iteration complexity?
>
> **A:** Thank you for your suggestion. In the revised paper, we will add the following practical examples for the assumptions.
>
> Assumption 1 covers popular policy parameterizations including softmax policy $\pi_{\theta}(a|s)=\frac{\exp(\theta_{s,a})}{\sum_{a'}\exp(\theta_{s,a'})}$ [6] and log-linear policy $\pi_{\theta}(a|s)=\frac{\exp(\theta\cdot\phi_{s,a})}{\sum_{a'}\exp(\theta\cdot\phi_{s,a'})}$ [6], as well as popular transition kernels including direct parameterization $p_{\xi}(s'|s,a)=\xi_{s,a,s'}$ [7,8] and linear parameterization $p_{\xi}(s'|s,a)=\xi\cdot\phi_{s,a,s'}$ [9,10] when $\Xi$ is located away from 0, i.e., $\inf_{s,a,s',\xi\in\Xi}p_{\xi}(s'|s,a)\ge p_{\min}$ for a constant $p_{\min}>0$. Assumptions 2-3 cover all the three convex utilities listed in our answer to your Q2. Assumptions 4 and 8 are similar and cover direct policy parameterization $\pi_{\theta}(a|s)=\theta_{s,a}$ (Actually, it has also been proved to satisfy Assumption 4.1 of [1] which implies Assumptions 4 and 8). Assumptions 5-7 cover $s$-rectangular $L_1$ and $L_{\infty}$ ambiguity sets (defined as $\Xi=\{\xi:\|\xi(s,:,:)-\xi^0(s,:,:)\| _ p\le \alpha_s\}$ for $p\in\{1,\infty\}$ respectively using direct kernel parameterization $p_{\xi}(s'|s,a)=\xi_{s,a,s'}$), which are very popular in robust RL [3,4].
>
> We are not sure about the tightness of our results because the complexity lower bounds are unknown for our proposed robust RL with general utility. However, as mentioned in our conclusion, our gradient complexities may still be improved in the future, compared with the state-of-the-art gradient complexities (Lin et al., 2020; Pethick et al., 2023) for minimax optimization.
>
> [1] Zhang, Junyu, et al. "Variational policy gradient method for reinforcement learning with general utilities." Advances in Neural Information Processing Systems 33 (2020): 4572-4583.
>
> [2] Corsi, Davide, et al. "Constrained reinforcement learning and formal verification for safe colonoscopy navigation." 2023 IEEE/RSJ International Conference on Intelligent Robots and Systems (IROS). IEEE, 2023.
>
> [3] Hu, Xuemin, et al. "Long and Short-Term Constraints Driven Safe Reinforcement Learning for Autonomous Driving." ArXiv:2403.18209 (2024).
>
> [4] Khairy, Sami, et al. "Constrained deep reinforcement learning for energy sustainable multi-UAV based random access IoT networks with NOMA." IEEE Journal on Selected Areas in Communications 39.4 (2020): 1101-1115.
>
> [5] Hazan, Elad, et al. "Provably efficient maximum entropy exploration." International Conference on Machine Learning. PMLR, 2019.
>
> [6] Agarwal, Alekh, et al. "On the theory of policy gradient methods: Optimality, approximation, and distribution shift." Journal of Machine Learning Research 22.98 (2021): 1-76.
>
> [7] Kumar, Navdeep, et al. "Policy gradient for rectangular robust markov decision processes." Advances in Neural Information Processing Systems (2023).
>
> [8] Behzadian, Bahram, Marek Petrik, and Chin Pang Ho. "Fast Algorithms for $L_\infty$-constrained S-rectangular Robust MDPs." Advances in Neural Information Processing Systems (2021).
>
> [9] Ayoub, Alex, et al. "Model-based reinforcement learning with value-targeted regression." International Conference on Machine Learning. PMLR, 2020.
>
> [10] Zhang, Junkai, Weitong Zhang, and Quanquan Gu. "Optimal horizon-free reward-free exploration for linear mixture mdps." International Conference on Machine Learning. PMLR, 2023.

---

> > ### Author Response · Authors · 2024-08-09
> > **A better answer to Reviewer iNNQ's Q3**
> >
> > Dear Reviewer iNNQ,
> >
> > Our above answer to your Q3 gives examples for each assumption. The combination of the following policy, transition kernel, ambiguity set and utility function satisfy all the assumptions (Assumptions 1-8).
> >
> > $\bullet$ Softmax policy: $\pi _ {\theta}(a|s)=\frac{\exp(\theta _ {s,a})}{\sum _ {a'}\exp(\theta _ {s,a'})}, \theta\in\Theta$, where the range $\Theta\subseteq[-R,R]^{|\mathcal{S}|\times|\mathcal{A}|}$ for some constant $R>0$ to prevent $\pi _ {\theta}(a|s)$ from approaching 0.
> >
> > $\bullet$ Directly parameterized transition kernel: $p _ {\xi}(s'|s,a)=\xi _ {s,a,s'}$.
> >
> > $\bullet$ s-rectangular $L^1$ ambiguity set: $\Xi\overset{\rm def}{=}${$\xi:||\xi(s,:,:)-\xi^0(s,:,:)||_1\le \alpha_s, \forall s$}, where the fixed nominal kernel $\xi^0$ satisfies $\xi^0(s,a,s')>\alpha_s, \forall s,a,s'$ to prevent $\xi(s,a,s')$ from approaching 0.
> >
> > $\bullet$ Convex utility function for pure exploration [1]: $f(\lambda)=\sum_s \lambda(s)\log\lambda(s)$ where $\lambda(s)\overset{\rm def}{=}\sum_a\lambda(s,a)$. The inital state distribution $\rho$ satisfies $\rho(s)>0, \forall s$ to prevent $\lambda(s)$ from approaching 0.
> >
> > [1] Zhang, Junyu, et al. "Variational policy gradient method for reinforcement learning with general utilities." Advances in Neural Information Processing Systems 33 (2020): 4572-4583.

---

> > > ### Comment · Reviewer_iNNQ · 2024-08-11
> > > **Thanks**
> > >
> > > Thank you for your response. I’ll keep my score.

---

### Author Rebuttal · Authors · 2024-08-06

**About Motivation**

Thank the reviewers for bringing the motivation issue to our attention. **Our revision will include the following special cases and application examples of our proposed problem. First, our proposed *robust RL with general utility* can be applied to improve the policy robustness for the following useful special cases of the existing *RL with general utility* as well as their application examples [1].**

(1) MDP with Constraints or Barriers, which has **safety-critical applications including healthcare [2], autonomous driving [3] and unmanned aerial vehicle [4].**

(2) Pure exploration, which can be **applied to explore environment that is lack of reward signals [5].**

(3) Learning to mimic a demonstration, which is **used to help the agent mimic the expert demonstration [1].**

**Second, our proposed problem also covers the following useful *robust* special cases and their application examples.**

**(4) Robust Constrained RL [6-8]:**
In **safety critical applications such as healthcare and unmanned aerial vehicle**, it is important for an intelligent agent to constrain its behavior to a safe range while optimizing the performance. However, in practice, the test environment often differs from the training environment which may degrade performance, and may even violate the safety constraints. For example, a safe and effective treatment for a patient may be fatal for another patient. A drone may run out of battery, leading to a crash in the test environment [8].

Robust constrained RL has been proposed to guarantee safety in all possible test environments while  optimizing the robust performance. In robust constrained RL, there are two cost functions $c^{(0)}, c^{(1)}$ relating to performance and safety respectively. Denote $\lambda_{\theta,\xi}$ as the occupancy measure under the policy parameter $\theta\in\Theta$ and the environment parameter $\xi\in\Xi$. Define value functions $V_{\theta,\xi}^{(0)}, V_{\theta,\xi}^{(1)}$ and robust value functions $V_{\theta}^{(0)}, V_{\theta}^{(1)}$ as follows.

$$
    V_{\theta,\xi}^{(k)}\stackrel{\text { def }}{=}\langle c^{(k)},\lambda_{\theta,\xi}\rangle=\sum_{s,a}c^{(k)}(s,a)\lambda_{\theta,\xi}(s,a),  \quad\quad
    V_{\theta}^{(k)}\stackrel{\text { def }}{=}\max_{\xi\in\Xi} V_{\theta,\xi}^{(k)}, ~ k=0,1.
$$
Robust constrained RL is formulated as the following constrained policy optimization problem.
$$
    \min _ {\theta\in\Theta} V _ {\theta}^{(0)}, ~ {\rm s.t.} ~ V _ {\theta}^{(1)}\le \tau.
$$
where $\tau\in\mathbb{R}$ is the safety threshold.  It can be easily verified that the above robust constrained RL problem is a special case of our proposed  problem $\min_{\theta\in\Theta}\max_{\xi\in\Xi}f(\lambda_{\theta,\xi})$ with the following utility function $f$.
$$
    f(\lambda)=
        \langle c^{(0)},\lambda\rangle {\rm ~ if ~ }\langle c^{(1)},\lambda\rangle\le \tau {\rm ~ and ~ }
        f(\lambda)=+\infty{\rm ~ if ~ }\langle c^{(1)},\lambda\rangle>\tau.
$$

**(5) Entropy Regularized Robust RL [9-10]:** Entropy regularized robust RL has been applied to **imitation learning and inverse reinforcement learning which help agents learn from human demonstration [10]**. Entropy regularized robust RL is formulated as the following minimax optimization problem.
\begin{align}
    \min_{\theta\in\Theta}\max_{\xi\in\Xi} \sum_{s,a}\big[\lambda_{\theta,\xi}(s,a)c(s,a)\big]-\mu\sum_s\big[\lambda_{\theta,\xi}(s)\mathcal{H}[\pi_{\theta}(\cdot|s)]\big],
\end{align}
where $c$ is the cost function, $\lambda_{\theta,\xi}(s)=\sum_a\lambda_{\theta,\xi}(s,a)$ is the state occupancy measure, and $\mathcal{H}[\pi_{\theta}(\cdot|s)]=-\sum_{a}\pi_{\theta}(a|s)\log\pi_{\theta}(a|s)$ is the entropy regularizer (with coefficient $\mu>0$) which encourages the agent to explore more states and actions and helps to prevent early convergence to sub-optimal policies.

It can be easily verified that the above entropy regularized robust RL problem is a special case of our proposed problem $\min_{\theta\in\Theta}\max_{\xi\in\Xi}f(\lambda_{\theta,\xi})$ with the following utility function $f$.
\begin{align}
    f(\lambda)=\sum_{s,a}\lambda(s,a)\Big[c(s,a)+\mu\log\frac{\lambda(s,a)}{\sum_{a'}\lambda(s,a')}\Big]
\end{align}

[1] Zhang, Junyu, et al. "Variational policy gradient method for reinforcement learning with general utilities." Advances in Neural Information Processing Systems 33 (2020): 4572-4583.

[2] Corsi, Davide, et al. "Constrained reinforcement learning and formal verification for safe colonoscopy navigation." 2023 IEEE/RSJ International Conference on Intelligent Robots and Systems (IROS). IEEE, 2023.

[3] Hu, Xuemin, et al. "Long and Short-Term Constraints Driven Safe Reinforcement Learning for Autonomous Driving." ArXiv:2403.18209 (2024).

[4] Khairy, Sami, et al. "Constrained deep reinforcement learning for energy sustainable multi-UAV based random access IoT networks with NOMA." IEEE Journal on Selected Areas in Communications 39.4 (2020): 1101-1115.

[5] Hazan, Elad, et al. "Provably efficient maximum entropy exploration." International Conference on Machine Learning. PMLR, 2019.

[6] Russel, Reazul Hasan, Mouhacine Benosman, and Jeroen Van Baar. "Robust Constrained-MDPs: Soft-constrained Robust Policy Optimization Under Model Uncertainty." ArXiv:2010.04870 (2020).

[7] Sun, Zhongchang, et al. "Constrained Reinforcement Learning Under Model Mismatch." ArXiv:2405.01327 (2024).

[8] Wang, Yue, Fei Miao, and Shaofeng Zou. "Robust Constrained Reinforcement Learning." ArXiv:2209.06866 (2022).

[9] Mankowitz, D. J., Levine, N., Jeong, R., Abdolmaleki, A., Springenberg, J. T., Shi, Y., Kay, J., Hester, T., Mann, T., and Riedmiller, M. (2019). Robust reinforcement learning for continuous control with model misspecification. In Proceedings of the International Conference on Learning Representations (ICLR).

[10] Mai, T. and Jaillet, P. (2021). Robust entropy-regularized markov decision processes. ArXiv:2112.15364.

---

### Decision · Program_Chairs · 2024-09-25

**Decision:**

Accept (poster)

**Comment:**

This paper studies concave utility reinforcement learning where the policy belongs to a parameterized uncertainty set. A novel stochastic gradient descent-ascent algorithm for policy optimization is proposed, theoretically analyzed, and experimentally validated. The reviewers are nearly unanimous of its technical merit and innovations. I concur with these assessments and thus recommend this paper be accepted.